# Stochastic Optimal Control for Continuous-Time fMRI Representation Learning

**Joonhyeong Park**[1*], **Byoungwoo Park**[1*], **Chang-Bae Bang**[2], **Jungwon Choi**[1],
**Hyungjin Chung**[1,3], **Byung-Hoon Kim**[2†], **Juho Lee**[1†]
KAIST[1], Yonsei University[2], EverEx[3]
{clearclouds, bw.park, jungwon.choi, juholee}@kaist.ac.kr
hyungjin.chg@gmail.com, {changbae.bang, egyptdj}@yonsei.ac.kr

## Abstract

Learning robust representations from functional magnetic resonance imaging (fMRI) is fundamentally challenged by the temporal irregularity and noise inherent in data from heterogeneous sources. Existing self-supervised learning (SSL) methods often discard critical temporal information by discretizing or averaging fMRI signals. To address this, we introduce a novel framework that reframes SSL as a Stochastic Optimal Control (SOC) problem. Our approach models brain activity as continuous-time latent dynamics, learning a robust representation of brain dynamics by optimizing a control policy that is agnostic to the temporal irregularity. This SOC framework naturally unifies masked autoencoding (MAE) and joint-embedding prediction (JEPA) to extract compact, control-derived representations. Furthermore, a simulation-free inference strategy ensures computational efficiency and scalability for large-scale fMRI datasets. Our model demonstrates state-of-the-art performance across diverse downstream applications, highlighting the potential of the SOC-based continuous-time representation learning framework.

## 1 Introduction

Blood-oxygen-level-dependent (BOLD) signals, captured by fMRI, reflect localized brain activity, providing an indirect measure of underlying brain dynamics (Ogawa et al., 1990; Heeger & Ress, 2002). These signals imply fundamental brain dynamics essential for understanding cognitive functions, behavior, and clinical characteristics (Park & Friston, 2013). While supervised learning models tailored for specific tasks like diagnosis prediction have shown promise (Kawahara et al., 2017; Li et al., 2021; Kan et al., 2022), their dependence on labeled data limits flexibility and generalizability. This motivates the development of methods that leverage vast amounts of unlabeled fMRI data to learn rich, transferable brain representations (Abraham et al., 2017; Yamashita et al., 2020; Zhang & Metaxas, 2024). Building on its success in fields such as computer vision and natural language processing (Devlin et al., 2019; He et al., 2020; Chen et al., 2020; LeCun, 2022; He et al., 2022), the SSL framework has recently been applied to fMRI analysis (Caro et al., 2024; Dong et al., 2024; Yang et al., 2024).

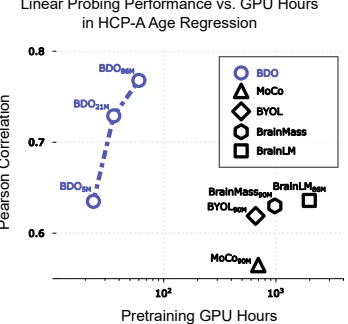

Figure 1: Our BDO outperforms other self-supervised approaches, demonstrating superior efficiency.

Despite their success at capturing global brain representations, existing SSL methods often compromise the fidelity of fMRI temporal dynamics by treating the time-series data as segmented patches or static connectivity graphs as described in Figure 2. Since fMRI signals are noisy and continuously evolve over time, it is desirable for models to incorporate temporal inductive biases that naturally reflect the true nature of fMRI data to fully exploit these dynamics. However, in SSL scenarios that must integrate heterogeneous fMRI datasets, temporal modeling becomes even more complex. A key difficulty arises from variability in acquisition protocols, especially in repetition time (TR), the

---

[*]Equal contribution
[†]Equal advising.

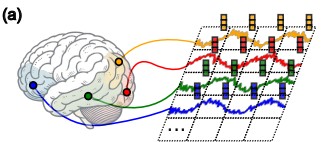 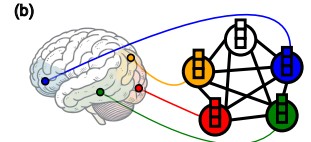 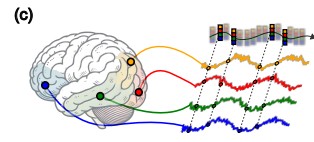

Figure 2: Conceptual illustration of SSL approaches for fMRI data. (a) Image-based approach (Caro et al., 2024; Dong et al., 2024), where time-series data for each region-of-interest (ROI) is divided into fixed-size windows. (b) Graph-based approach (Yang et al., 2024), where static graphs are constructed to represent functional connectivity between ROIs. Both approaches discard high-resolution temporal dynamics during data preprocessing. (c) Our approach, leveraging continuous-time latent dynamics, directly captures the evolution of brain activity over time, making it robust to varying TRs across heterogeneous datasets and preserving fine-grained details.

interval between successive fMRI measurements. For example, the UK Biobank (UKB) uses a TR of 0.735-second (Alfaro-Almagro et al., 2018), whereas the Autism Brain Imaging Data Exchange (ABIDE) reports TRs ranging from 1.5 to 3.0-second (Di Martino et al., 2014). Integrating such datasets poses challenges in aligning heterogeneous time scales into a common time axis, resulting in the irregularity as described in Figure 3. Therefore, such irregularity highlights the need for a unified framework that can model temporal dynamics across heterogeneous datasets with diverse TRs.

Continuous-time dynamical modeling has become a natural solution for irregularly sampled data in finance (Black & Scholes, 1973), climate (Menne et al., 2016), and healthcare (Goldberger et al., 2000). Neural differential equations (Chen et al., 2018; Rubanova et al., 2019; Li et al., 2020; Kidger et al., 2020; Zeng et al., 2023) and continuous–discrete state-space methods (Schirmer et al., 2022; Ansari et al., 2023; Park et al., 2024) learn continuous dynamics in a compact latent space rather than on the noisy, high-dimensional

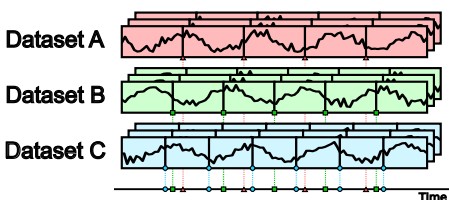

Figure 3: Diverse TRs induce multi-scale irregularities on a unified time axis.

observation space. While integrating these latent trajectories requires costly numerical solvers, recent simulation-free and amortized inference techniques remove this bottleneck, making continuous-time modeling practical for large-scale, high-dimensional datasets (Park et al., 2024) such as fMRI.

Building on these advances, we propose a novel SSL framework, **Brain Dynamics with Optimal control (BDO)**, designed to explicitly model continuous latent dynamics while accounting for the heterogeneous data acquisition protocols of multi-site fMRI studies. In BDO, continuous latent state evolution is modeled as a stochastic differential equation (SDE) (Oksendal, 1992), which encodes multi-scale brain activity onto a single continuous real-time axis. Crucially, we fundamentally reframe fMRI representation learning as a SOC problem (Fleming & Soner, 2006; Carmona, 2016). By viewing the control signal as the driving force that aligns latent dynamics with observed brain activity, we reinterpret the *optimal control policy* as a learned latent encoder. This formulation naturally yields a unified objective that integrates two complementary SSL schemes, the masked autoencoder (MAE) (He et al., 2022) and the joint embedding predictive architecture (JEPA) (LeCun, 2022; Assran et al., 2023), to learn robust representations. Furthermore, to enhance computational efficiency, we employ a *simulation-free* inference strategy based on locally linear approximations of the SDE. The learned control-based latent representations serve as compact and highly transferable representations for a wide range of downstream applications. We summarize our contributions as follows:

- We explicitly model the temporal structure of fMRI time-series via continuous latent dynamical modeling, thereby ensuring robustness to heterogeneous data acquisition protocols.
- Building on the SOC formulation, we incorporate MAE and JEPA into a single SSL framework yielding compact, transferable control-based features.
- A simulation-free inference scheme eliminates costly numerical solvers, enabling scalable and efficient training on large, high-dimensional fMRI datasets as shown in Figure 1.
- Our experiments empirically show that the learned control-based features outperform baselines across diverse downstream tasks while retaining high computational efficiency and scalability.

## 2 RELATED WORK

**Task-specific Models for fMRI.** Supervised deep learning has driven fMRI analysis for diagnosis and trait inference. Convolutional approaches like BrainNetCNN (Kawahara et al., 2017), graph-based models like BrainGNN (Li et al., 2021), and transformer-based BrainNetTF (Kan et al., 2022) have achieved strong performance by capturing ROI interactions or functional modules. However, these task-specific models rely heavily on labeled data and fail to learn generalizable representations.

To address this, SSL has emerged as a powerful paradigm designed to overcome the reliance on labeled data. By exploiting the abundance of unlabeled fMRI scans, SSL aims to distill task-agnostic representations that generalize across a wide spectrum of downstream fMRI analyses.

**Self-Supervised Learning for fMRI.** Building on the success in vision and language domains (He et al., 2020; Grill et al., 2020; He et al., 2022; Assran et al., 2023), existing SSL approaches for fMRI generally follow two main routes as illustrated in Figure 2: image-based and graph-based approaches.

Image-based models, such as BrainLM (Caro et al., 2024) and Brain-JEPA (Dong et al., 2024), treat fMRI time-series as image-like fixed grids of spatiotemporal patches. However, this rigid discretization creates a fundamental *temporal mismatch* when applying models to heterogeneous datasets with varying TRs. For instance, even when artificial downsampling is employed to approximate a common temporal resolution, it inevitably fails to perfectly synchronize the diverse time scales of multi-site data, forcing the model to learn inconsistent dynamics from patches representing disparate durations.

Conversely, graph-based approaches, including BrainMass (Yang et al., 2024), compress time-series into static functional connectivity graphs, inherently discarding fine-grained temporal dynamics. While recent dynamic graph methods (Choi et al., 2024) utilize sliding windows, they still average out intra-window fluctuations and struggle to align diverse sampling rates across datasets.

To address these issues, we adopt continuous-time latent dynamical modeling. While the concurrent work BrainHarmonix (Dong et al., 2025) introduces a multimodal framework unifying structure and function, its approach to fMRI attempts to mitigate TR heterogeneity via adaptive patch resizing, fundamentally remaining a discrete patch-based method. In contrast, by viewing each fMRI scan as noisy observations of a continuous SDE, our model naturally aligns data with different TRs on a shared continuous real-time axis, handles variable sequence lengths without resampling, and retains the fine-grained temporal dynamics that patch-based or static-graph approaches discard.

**Continuous Dynamical Models.** Continuous-time dynamical models have been developed to capture irregular time-series dynamics and uncertainty. Deterministic approaches like Neural and Latent ODEs parameterize smooth trajectories via neural networks (Chen et al., 2018; Rubanova et al., 2019), while GRU-ODE-B incorporates Bayesian updates for uncertainty (De Brouwer et al., 2019). Stochastic Latent-SDEs introduce variational inference schemes for SDE-driven latent trajectories (Li et al., 2020; Hasan et al., 2021; Zeng et al., 2023). Classical continuous–discrete state-space models generalize discrete transitions to SDE-governed updates (Jazwinski, 2007), inspiring neural CD-SSM variants that leverage locally linear approximations (Schirmer et al., 2022; Ansari et al., 2023).

Although these frameworks excel at modeling irregular time-series, they rely on costly numerical solvers, hindering scalability for high-dimensional fMRI. Recently, simulation-free approaches have been proposed to bypass expensive integration (Bartosh et al., 2025; Park et al., 2024). Inspired by these simulation-free paradigms, we introduce a scalable SSL framework with a closed-form objective, capturing fMRI dynamics and uncertainty without expensive numerical solvers. [1]

## 3 METHOD

In this section, we introduce BDO, a novel SSL framework for modeling fMRI time-series data using continuous latent dynamics. We formulate representation learning as a SOC problem, allowing us to unify two complementary SSL schemes, MAE and JEPA, into a single training objective, in a scalable and simulation-free manner. Proofs and detailed derivations are provided in Appendix B.

---

[1]We refer readers to Section A for an additional related work section including detailed backgrounds on self-supervised learning and stochastic optimal control foundations.

### 3.1 Modeling Continuous Dynamics in Latent Space for fMRI

**Latent Dynamics.** Here, we briefly introduce the core assumptions that guide our method. Directly working with raw fMRI time-series data, which we call $\mathcal{Y} := \{\mathbf{y}_{t_1}, \cdots, \mathbf{y}_{t_k}\}$ (where each data point $\mathbf{y}_{t_i} \in \mathbb{R}^n$) with time stamps $\mathcal{T} = \{t_1, \cdots, t_k\}$[2] is challenging due to their high-dimensional nature and inherent noise. Instead, we introduce a *latent space*, where the essential structure of the data is assumed to lie in a lower-dimensional space of dimension $d < n$. This approach enables more efficient modeling by reducing the dimensionality of the data (Fraccaro et al., 2017; Rubanova et al., 2019; Li et al., 2020; Kidger et al., 2020; Zeng et al., 2023; Ansari et al., 2023; Park et al., 2024).

Based on this, we assume that the complex signals we see in brain scans are produced by some underlying hidden dynamics that unfold continuously over time. We model these continuous latent states $\mathbf{X}_t \in \mathbb{R}^d$ as being governed by an Itô diffusion process (Oksendal, 1992):

$$d\mathbf{X}_t = f(t, \mathbf{X}_t)dt + \sigma(t)d\mathbf{W}_t, \quad (1)$$

where $f(t, \cdot) : \mathbb{R}^d \to \mathbb{R}^d$ is the drift, $\sigma(t) \in \mathbb{R}$ is the diffusion coefficient, and $\mathbf{W}_t \in \mathbb{R}^d$ is a standard Wiener process. The stochastic process $\mathbf{X}_{[0,T]} := \{\mathbf{X}_t\}_{t \in [0,T]}$ will serve as the *prior*

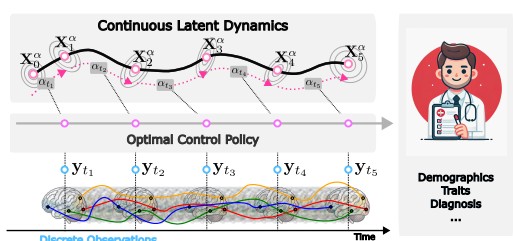

Figure 4: Conceptual illustration of our proposed **Brain Dynamics with Optimal control (BDO)**.

understanding of the system before we look at the actual fMRI data $\mathcal{Y}$. Because fMRI data are too complex, it is hard to make strong initial assumptions. So, a common starting point is to simplify the prior to $d\mathbf{X}_t = \sigma(t)d\mathbf{W}_t$. This means that we initially assume the underlying changes are purely random, acknowledging the difficulty in modeling the intricate nature of fMRI signals beforehand.

To learn meaningful representations from fMRI time-series, our goal is to infer the latent dynamics that generate the observed complex signals $\mathcal{Y}$. A successful latent dynamics model should serve as the representation itself, providing a compressed and powerful summary of the brain's activity. Here, we define these dynamics as the *posterior* stochastic process, $\mathbf{X}_t^\star$, which provides the best possible explanation for the fMRI data $\mathcal{Y}$ that we have actually observed. The posterior states $\mathbf{X}_t^\star$ are also governed by an Itô diffusion process (Li et al., 2020; Park et al., 2024):

$$d\mathbf{X}_t^\star = [f(t, \mathbf{X}_t^\star) + \sigma(t)\alpha^\star(t, \mathbf{X}_t^\star; \mathcal{Y})]dt + \sigma(t)d\mathbf{W}_t, \quad (2)$$

where $\alpha^\star(t, \cdot, \cdot) : \mathbb{R}^d \times \mathbb{R}^{n \times k} \to \mathbb{R}^d$ is introduced as part of the drift term. Compared to the prior process in (1), the newly added term $\alpha^\star$ steers our initial understanding of the hidden dynamics. It achieves this by incorporating the information from the actual fMRI signals $\mathcal{Y}$ we collected, pushing the latent dynamics towards a path that better aligns with the observed fMRI data.

**Problem setup.** The core idea is that the prior process is aligned with the observed data $\mathcal{Y}$ through the term $\alpha^\star$, which plays an exclusive role in incorporating observed information into the latent states, thereby enabling the direct extraction of a meaningful representation from $\alpha^\star$. Once an *optimal* profile of $\alpha^\star$ is determined for the given set of data $\mathcal{Y}$, this representation can then be employed for downstream tasks as illustrated in Figure 4. Indeed, $\alpha^\star$ is referred to as the *optimal policy*, which can be developed under stochastic optimal control (SOC) theory (Fleming & Soner, 2006; Carmona, 2016; Park et al., 2024). Drawing on the SOC theory, we propose our method specifically designed to model the fMRI time-series data, with a focus on SSL to extract meaningful representations.

**Stochastic Optimal Control.** SOC provides a robust theoretical foundation for the alignment process by combining optimization and probability theory to determine optimal control strategies for dynamical systems (Fleming & Soner, 2006; Carmona, 2016). To estimate the *optimal* latent trajectory in (2) which is aligned with the observed data $\mathcal{Y}$, we introduce a *controlled* SDE:

$$d\mathbf{X}_t^\theta = \left[f(t, \mathbf{X}_t^\theta) + \sigma(t)\alpha^\theta(t, \mathbf{X}_t^\theta; \mathcal{Y})\right]dt + \sigma(t)d\mathbf{W}_t, \quad (3)$$

where $\alpha^\theta(t, \cdot, \cdot) : \mathbb{R}^d \times \mathbb{R}^{n \times k} \to \mathbb{R}^d$ represents the parameterized (with a neural network) control policy we aim to optimize and $\mathbf{X}_{[0,T]}^\theta$ represent the parameterized latent states controlled by $\alpha^\theta$. In

---

[2] The time stamps are ordered as $0 < t_1 < \cdots < t_k < T$ and the intervals between consecutive time stamps can be arbitrary.

our formulation, the control policy $\alpha^\theta$ acts as an *encoder* by mapping the observed data $\mathcal{Y}$ to the latent space. We parameterize this encoder $\alpha^\theta$ as a transformer (Vaswani et al., 2017), recognized for its effectiveness in processing sequential data such as time-series observations. The objective is to approximate the optimal control policy $\alpha^\star$ by $\alpha^{\theta \to \theta^\star}$, thereby aligning $\mathbf{X}^\star_{[0,T]} \equiv \mathbf{X}^{\theta^\star}_{[0,T]}$. This optimization is related to the variational inference framework (Chen et al., 2018), and the solution is structured as follows (Theodorou, 2015; Kappen & Ruiz, 2016; Li et al., 2020; Park et al., 2024):

**Proposition 3.1** (Evidence lower bound). *Let us consider the following optimal control problem:*

$$\mathcal{J}(\alpha^\theta, \mathcal{Y}) = \mathbb{E}_{\mathbf{X}^\alpha \sim (3)} \left[ \int_0^T \frac{1}{2} \left\| \alpha^\theta(t, \mathbf{X}^\theta_t; \mathcal{Y}) \right\|^2 dt - \sum_{t \in \mathcal{T}} \log g_\psi(\mathbf{y}_t | \mathbf{X}^\theta_t) \right], \tag{4}$$

*where $g_\psi(\mathbf{y}|\mathbf{x}) = \mathcal{N}(\mathbf{y}|\mathbf{D}_\psi(\mathbf{x}), \sigma_\mathbf{D}^2 \mathbf{I})$ is parameterized (Gaussian) likelihood function with a non-linear decoder network $\mathbf{D}_\psi : \mathbb{R}^d \to \mathbb{R}^n$. The negation of the objective function $\mathcal{J}(\alpha^\theta, \mathcal{Y})$ coincides with the evidence lower bound (ELBO) of the variational inference for the posterior dynamics (2):*

$$ELBO(\theta, \psi) \geq -\mathcal{J}(\alpha^\theta, \mathcal{Y}), \tag{5}$$

*where the equality holds at the optimal control policy $\alpha^\star = \arg\min_\theta \mathcal{J}(\alpha^\theta, \mathcal{Y})$.*

With this parameterized encoder-decoder, $(\alpha^\theta, \mathbf{D}_\psi)$, Proposition 3.1 shows that the optimization problem with the cost function specified in (4) is equivalent to performing variational inference within an *autoencoder framework* (Kingma & Welling, 2014).

Moreover, when related to reinforcement learning (RL) literature, it aligns with continuous-time RL with entropy regularization (Todorov, 2006), where the integral term $\frac{1}{2} \left\| \alpha^\theta(t, \mathbf{X}^\theta_t; \mathcal{Y}) \right\|^2$ enforces regularization to maintain proximity to the prior process (1) and the negative log-likelihood, $-\log g_\psi(\mathbf{y}|\mathbf{x}) :=$ $\frac{1}{2\sigma_\mathbf{D}^2} \left\| \mathbf{y} - \mathbf{D}_\psi(\mathbf{x}) \right\|^2$, acts as a reward for posterior process to explain the observed fMRI time-series.

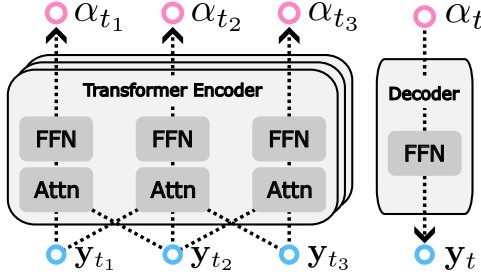

Figure 5: **(Left)** Encoder network architecture **(Right)** Decoder network architecture.

Once the optimal policy $\alpha^\star$ is obtained by solving the SOC problem, we can sample from the posterior states $\mathbf{X}^{\theta^\star}_t$ by simulating the *optimally controlled* SDE (3) via a numerical solver (Li et al., 2020).

**Universal Feature from Optimal Control Policy.** After obtaining the optimal policy $\alpha^\star$, we can derive a *universal feature* $\mathbb{A}$. This feature is generated by aggregating the control signals from the optimal policy across a relevant set of time points or intervals $\mathcal{T}$, using an aggregation function $f$, which gives $\mathbb{A} = f(\{\alpha^\star_t\}_{t \in \mathcal{T}})$. The aggregated feature $\mathbb{A}$ is designed to serve as a robust and generalizable representation for various downstream tasks. The rationale behind its potential as a universal feature is that $\mathbb{A}$, derived from the optimal policy $\alpha^\star$, encapsulates the essential data-driven adjustments required to accurately model the observed fMRI time-series $\mathcal{Y}$, thereby capturing salient and transferable characteristics of the underlying neural dynamics. The process of aggregating the control signals $\{\alpha^\star_t\}_{t \in \mathcal{T}}$ to extract the universal feature $\mathbb{A}$ is described in detail in Appendix D.4.

### 3.2 REPRESENTATION LEARNING WITH STOCHASTIC OPTIMAL CONTROL

Extracting robust and transferable features from fMRI time-series is crucial for downstream clinical and research applications. However, conventional approaches to representation learning often struggle to adequately capture the intricate underlying dynamics and subtle individual variations present in fMRI signals. This frequently results in features with limited generalizability across diverse tasks.

To overcome these challenges, we propose a novel SOC framework which integrates two complementary SSL schemes. First, we adopt a masked autoencoder (MAE) framework (He et al., 2022). In this framework, portions of the input fMRI time-series $\mathcal{Y}$ are masked, and the model is tasked with reconstructing the missing target $\mathcal{Y}_{\texttt{tar}}$ from the unmasked context $\mathcal{Y}_{\texttt{ctx}}$. This process compels the encoder $\alpha^\theta$ to learn rich and meaningful latent representations that capture the underlying structure and temporal dependencies. Second, to mitigate the risk of representation collapse—a phenomenon where learned features become trivial or non-discriminative—we introduce an auxiliary variable $\tilde{\alpha}_t$

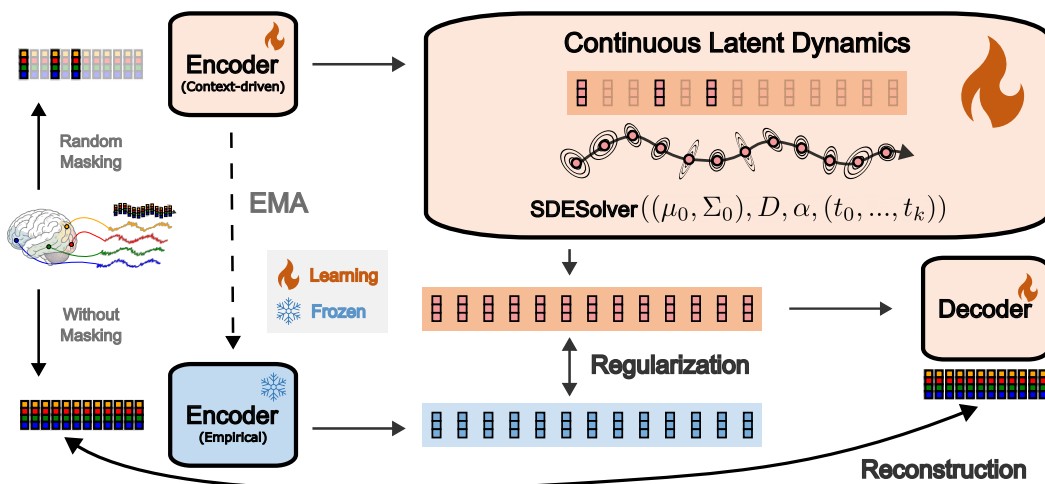

Figure 6: Overview of representation learning of BDO. Randomly masked fMRI time-series $\mathcal{Y}_{\tt ctx}$ are encoded into latent states; encoded control signals $\{\alpha_t^\theta\}_{t \in \mathcal{T}_{\tt ctx}}$ steer the SDE to predict latent states at the masked time points $\mathcal{T}_{\tt tar}$, which are then used to reconstruct the missing observations $\mathcal{Y}_{\tt tar}$. A slowly updated EMA encoder provides latent targets, preventing representation collapse.

that serves as a stable, self-generated latent prediction target. This mechanism, developed to ensure feature stability, shares its core philosophy with the joint embedding predictive architecture (JEPA) framework (LeCun, 2022; Assran et al., 2023), which also leverages latent prediction to shape representations. Grounded in SOC theory, our formulation successfully unifies these two complementary SSL schemes into a single framework, yielding robust and informative representations $\mathbb{A}$.

**Masked Autoencoders.** To learn the control signals that align the latent dynamics with the observed data, we use an MAE, which has been successfully applied in various domains (He et al., 2022). With an encoder-decoder pair $(\alpha^\theta, \mathbf{D}_\phi)$, the MAE approach can be adapted to our SOC framework. Specifically, we randomly mask a ratio $\gamma$ of data in $\mathcal{Y}$. The objective is then to reconstruct missing target $\mathcal{Y}_{\tt tar}$ based on the remaining context $\mathcal{Y}_{\tt ctx}$. We denote corresponding time points as $\mathcal{T}_{\tt tar}$ and $\mathcal{T}_{\tt ctx}$. In this formulation, the optimal control policy $\alpha^\star$ is determined by solving the SOC problem:

$$\alpha^\star = \arg\min_\theta \mathbb{E}_{\mathbf{X}^\alpha \sim (3)} \left[ \int_0^T \frac{1}{2} \left\| \alpha^\theta(t, \mathbf{X}_t^\theta; \mathcal{Y}_{\tt ctx}) \right\|^2 dt - \sum_{t \in \mathcal{T}_{\tt tar}} \log g_\psi(\mathbf{y}_t | \mathbf{X}_t^\theta) \right] \quad (6)$$

Here, the control $\alpha^\theta$ is generated by encoding the context observations $\mathcal{Y}_{\tt ctx}$. However, this approach may be suboptimal for highly noisy data modalities like fMRI as the naïve negative log-likelihood function, $-\log g_\psi(\mathbf{y}|\mathbf{x}) = \frac{1}{2\sigma_\mathbf{D}^2} \left\| \mathbf{y} - \mathbf{D}_\psi(\mathbf{x}) \right\|^2$, directly decodes the latent states $\mathbf{X}_t^\theta$ into the observed raw signals $\mathbf{y}_t$. In noisy data like fMRI, this can lead to the encoder-decoder pair $(\alpha^\theta, \mathbf{D}_\phi)$ overfitting to noise, as the model attempts to fit the irrelevant fluctuations in the observed signals (Assran et al., 2023; Dong et al., 2024). As a result, the model may not effectively capture semantically meaningful features of the underlying dynamics, potentially compromising the generalizability of $\mathbb{A}$.

**Joint Embedding Prediction.** To mitigate overfitting to noise inherent in the pure signal reconstruction, we introduce an auxiliary variable $\tilde{\alpha}_t$ as the latent predictive target for the control policy $\alpha_t$, and incorporate additional structure into the observation likelihood function $g_\psi(\cdot|\mathbf{x})$. Formally, we propose a hierarchical generative process over the likelihood function $g_\psi$ in (6) as follows:

$$g_\psi(\mathbf{y}_t | \mathbf{X}_t^\theta) := \int \zeta_\psi(\mathbf{y}_t | \tilde{\alpha}_t) \pi(\tilde{\alpha}_t | \mathbf{X}_t^\theta) d\tilde{\alpha}_t, \quad (7)$$

where $\zeta_\psi(\mathbf{y}_t|\cdot) := \mathcal{N}(\mathbf{y}|\mathbf{D}_\psi(\cdot), \sigma_\zeta^2 \mathbf{I})$ represents the latent decoder mapping back to the observation space, and $\pi(\tilde{\alpha}_t | \mathbf{X}_t^\theta)$ is the latent distribution of the auxiliary variable $\tilde{\alpha}_t$. To ensure the latent target $\tilde{\alpha}_t$ is both predictable from the given context $\mathcal{Y}_{\tt ctx}$ and informative about the masked target $\mathcal{Y}_{\tt tar}$, we specifically define the latent distribution $\pi(\cdot|\mathbf{X}_t^\theta)$ as a geometric mixture of two components:

$$\pi_{\bar{\theta}}(\tilde{\alpha}_t | \mathbf{X}_t^\theta) \propto p_\theta(\tilde{\alpha}_t | \mathbf{X}_t^\theta, \mathcal{Y}_{\tt ctx})^\lambda q_{\bar{\theta}}(\tilde{\alpha}_t | \mathbf{X}_t^\theta, \mathcal{Y}_{\tt tar})^{(1-\lambda)}, \quad (8)$$

where $p(\tilde{\alpha}_t|\mathbf{X}_t^\theta, \mathcal{Y}_{\texttt{ctx}}) = \mathcal{N}(\tilde{\alpha}_t|\alpha^\theta(t, \mathbf{X}_t^\theta, \mathcal{Y}_{\texttt{ctx}}), \sigma_p^2\mathbf{I})$ is the *auxiliary posterior*, $q_{\bar{\theta}}(\tilde{\alpha}_t|\mathbf{X}_t^\theta, \mathcal{Y}_{\texttt{tar}}) = \mathcal{N}(\tilde{\alpha}_t|\alpha^{\bar{\theta}}(t, \mathbf{X}_t^\theta, \mathcal{Y}_{\texttt{tar}}), \sigma_q^2\mathbf{I})$ is the *empirical prior*, and $\lambda \in [0, 1]$ balances these two components.

The auxiliary posterior $p_\theta$ serves as the probabilistic estimate of the latent target $\tilde{\alpha}_t$, which is generated by the primary encoder $\alpha^\theta$ conditioned solely on the contextual information $\mathcal{Y}_{\texttt{ctx}}$. Essentially, this component represents the model's probabilistic belief about the target based solely on the observed context, before integrating any information from target-specific observations via the empirical prior.

Conversely, the empirical prior $q_{\bar{\theta}}$ serves as the predictive target distribution. It is generated by a separate target encoder $\alpha^{\bar{\theta}}$, which has access to the actual masked observations $\mathcal{Y}_{\texttt{tar}}$. The target encoder parameters $\bar{\theta}$ are not updated by backpropagation. Instead, they track the online primary encoder parameters $\theta$ via EMA, yielding a slowly evolving and stable reference (Assran et al., 2023).

This design ensures that $q_{\bar{\theta}}$ provides consistent, evolving predictive targets and the primary online encoder $\alpha^\theta$ is then trained to align its predictions with the stable targets. This process is a form of self-distillation that prevents the model from overfitting to noise and helps it learn robust features $\mathbb{A}$, sharing its core philosophy with the established JEPA framework (LeCun, 2022; Assran et al., 2023).

**Training Objective.** The above formulations allow the model to flexibly combine information from both contextual understanding and direct evidence from target data, thereby aiming to produce robust and generalizable features. Combining both modeling approaches within a single SOC framework, we define the following training objective properly scaled by variances of the likelihoods $(\sigma_\zeta^2, \sigma_q^2)$:

$$\hat{\mathcal{L}}_{\theta,\psi} = \mathop{\mathbb{E}}_{\mathbf{x}^\theta \sim (3)} \left[ \int_0^T \sigma_q^2 \left\| \alpha_t^\theta \right\|^2 dt - \sum_{t \in \mathcal{T}_{\texttt{tar}}} \mathop{\mathbb{E}}_{\tilde{\alpha}_t^\theta \sim p_\theta(\cdot|\mathbf{X}_t^\theta)} \left[ \left\| \mathbf{y}_t - \mathbf{D}_\psi(\tilde{\alpha}_t^\theta) \right\|^2 + \tau \left\| \tilde{\alpha}_t^\theta - \alpha_t^{\bar{\theta}} \right\|^2 \right] \right], \quad (9)$$

where $\tau = \frac{(1-\lambda)\sigma_\zeta^2}{\sigma_q^2}$ is a balancing factor. The parameters for the initial latent state and dynamics $\{\mu_0^\theta, \Sigma_0^\theta, \alpha^\theta\}$ and the decoder network $\{\mathbf{D}_\psi\}$ are jointly optimized by minimizing the training objective (9). The detailed derivation of this training objective is provided in Appendix B.3.

## 3.3 EFFICIENT MODELING OF LATENT DYNAMICS

The numerical simulation of (3) via SDE solvers is computationally demanding. To tackle these computational challenges, we turn to *simulation-free* methods. These methods suggest that complex data modeling can be achieved using linear SDEs (Schirmer et al., 2022; Smith et al., 2023; Ansari et al., 2023; Deng et al., 2024; Park et al., 2024), thereby avoiding costly numerical solvers. In particular, by approximating the drift function of controlled SDEs in (3) with a linear model, we obtain closed-form solutions for the latent states, removing the need for numerical integration. This approach drastically reduces computational demands, which is especially beneficial for high-dimensional data such as fMRI. Moreover, when combined with the assumption of piecewise locally linear dynamics, it enables robust simulation-free inference.

**Theorem 3.2** (Simulation-Free Inference). *Let us consider a sequence of semi-positive definite (SPD) matrices $\mathbf{D}_{t \in \mathcal{T}}$ where each $\mathbf{D}_{t_i} \in \mathbb{R}^{d \times d}$ admits the eigen-decomposition $\mathbf{D}_{t_i} = \mathbf{V}\boldsymbol{\Lambda}_{t_i}\mathbf{V}^\top$ with eigen-basis $\mathbf{V} \in \mathbb{R}^{d \times d}$ and eigen-values $\boldsymbol{\Lambda}_{t_i} \in diag(\mathbb{R}^d)$ for all $i \in \{1, \cdots, k\}$ and approximation of controls $\alpha_{t \in \mathcal{T}}^\theta$, where each $\alpha_t^\theta \in \mathbb{R}^d$. Then, for an interval $[t_i, t_{i-1})$, consider the SDE:*

$$d\mathbf{X}_t^\theta = \left[ -\mathbf{D}_{t_i}\mathbf{X}_t^\theta + \alpha_{t_i}^\theta(\mathcal{Y}) \right] dt + d\mathbf{W}_t, \quad (10)$$

*where $\mathbf{X}_0^\theta \sim \mathcal{N}(\mu_0, \Sigma_0)$. Then, for any time-stamps $t_i \in \mathcal{T}$, the marginal distribution of the solution of (10) is a Gaussian distribution i.e., $\mathbf{X}_{t_i}^\theta \sim \mathcal{N}(\mu_{t_i}, \Sigma_{t_i})$ whose parameters are computed as*

$$\mu_{t_i} = \mathbf{V}\left( e^{-\sum_{j=0}^{i-1}(t_{j+1}-t_j)\boldsymbol{\Lambda}_{t_j}}\hat{\mu}_{t_0} - \sum_{l=0}^{i-1} e^{-\sum_{j=l}^{i-1}(t_{j+1}-t_j)\boldsymbol{\Lambda}_{t_j}}\boldsymbol{\Lambda}_{t_l}^{-1}\left(\mathbf{I} - e^{(t_{l+1}-t_l)\boldsymbol{\Lambda}_{t_l}}\right)\hat{\alpha}_{t_l} \right) \quad (11)$$

$$\Sigma_{t_i} = \mathbf{V}\left( e^{-2\sum_{j=0}^{i-1}(t_{j+1}-t_j)\boldsymbol{\Lambda}_{t_j}}\hat{\Sigma}_{t_0} - \frac{1}{2}\sum_{l=0}^{i-1} e^{-2\sum_{j=l}^{i-1}(t_{j+1}-t_j)\boldsymbol{\Lambda}_{t_j}}\boldsymbol{\Lambda}_{t_l}^{-1}\left(\mathbf{I} - e^{2(t_{l+1}-t_l)\boldsymbol{\Lambda}_{t_l}}\right) \right)\mathbf{V}^\top,$$

*where $\hat{\mathbf{X}}_t^\alpha = \mathbf{V}^\top\mathbf{X}_t^\alpha$, $\hat{\alpha}_{t_i} = \mathbf{V}^\top\alpha_{t_i}$, $\hat{\mathbf{W}}_t = \mathbf{V}^\top\mathbf{W}_t$, $\hat{\mu}_0 = \mathbf{V}^\top\mu_0$ and $\hat{\Sigma}_0 = \mathbf{V}^\top\Sigma_0\mathbf{V}$.*

Hence, given the matrices and controls $\{\mathbf{D}_t, \alpha_t\}_{t\in\mathcal{T}}$, we can derive a closed-form solution for the latent states $\mathbf{X}_t^\theta$. Moreover, this parameterization enables the use of the parallel scan algorithm (Blelloch, 1990), allowing parallel computation of the moments $\{\mu_t, \Sigma_t\}_{t\in\mathcal{T}}$. Consequently, we can evaluate the objective function (9) in $\mathcal{O}(\log k)$ time. Further details can be found in Appendix C.2.

**Remark 3.3.** *A key advantage of this parameterization, once the model is trained via the SOC formulation, is that it enables the direct inference of the optimal policy $\alpha^\star$ for downstream tasks. In other words, obtaining $\alpha^\star$ does not require simulation of $\mathbf{X}_t^\star$ for downstream tasks. Instead, any necessary instances of $\mathbf{X}_t^\star$ are themselves efficiently inferred using the trained encoder.*

## 4 EXPERIMENTS

In this section, we present empirical results demonstrating that our novel SSL method, BDO, yields superior representations for fMRI analysis. BDO was pre-trained using the large-scale UK Biobank (UKB) dataset in a self-supervised manner, leveraging resting-state fMRI recordings from 41,072 participants (Alfaro-Almagro et al., 2018). To evaluate its applicability, we conducted experiments across various downstream tasks, including trait prediction and psychiatric diagnosis classification. These experiments were performed on four datasets: Human Connectome Project in Aging (HCP-A; Bookheimer et al., 2019), Autism Brain Imaging Data Exchange (ABIDE; Di Martino et al., 2014), Attention Deficit Hyperactivity Disorder 200 (ADHD200; Brown et al., 2012), and Human Connectome Project for Early Psychosis (HCP-EP; Prunier & Shenton Martha; Breier, 2021; Jacobs et al., 2024). All fMRI data were preprocessed by dividing brain into 450 ROIs, using Schaefer-400 for cortical regions and Tian-Scale III for subcortical areas (Schaefer et al., 2017; Tian et al., 2020).

We compared our performance against both task-specific (TS) models and self-supervised learning (SSL) models. Specifically, we compared BDO with TS models: BrainNetCNN (Kawahara et al., 2017), BrainGNN (Li et al., 2021), and BrainNetTF (also known as BNT) (Kan et al., 2022), as well as four SSL models: MoCo (He et al., 2020) and BYOL (Grill et al., 2020) as general SSL methods; BrainLM (Caro et al., 2024), and BrainMass (Yang et al., 2024) as fMRI-specific approaches. For a fair comparison, all results are averaged over three runs with different data splits. The best-performing results are highlighted in **bold**[3].

Table 1: Internal prediction on UKB 20% held-out.

| Methods | | Age | | Gender | |
|---|---|---|---|---|---|
| | | MSE ↓ | $\rho$ ↑ | ACC (%) ↑ | F1 (%) ↑ |
| TS | BrainNetCNN | 0.648 ±.018 | 0.621 ±.012 | 90.89 ±0.14 | 90.87 ±0.12 |
| | BrainGNN | 0.914 ±.024 | 0.430 ±.010 | 79.07 ±1.08 | 79.03 ±1.09 |
| | BrainNetTF | 0.561 ±.004 | 0.673 ±.003 | 91.19 ±0.51 | 91.17 ±0.50 |
| LP | MoCo (90M) | 0.933 ±.022 | 0.413 ±.010 | 80.11 ±0.73 | 80.11 ±0.73 |
| | BYOL (90M) | 0.859 ±.006 | 0.380 ±.006 | 72.98 ±0.13 | 72.97 ±0.13 |
| | BrainLM (85M) | 0.737 ±.019 | 0.547 ±.011 | 84.51 ±0.64 | 84.51 ±0.64 |
| | BrainMass (90M) | 1.104 ±.013 | 0.358 ±.006 | 80.33 ±0.86 | 80.34 ±0.86 |
| | BDO (86M) | **0.600 ±.004** | **0.635 ±.005** | **88.25 ±0.78** | **88.21 ±0.79** |
| FT | MoCo (90M) | 0.751 ±.023 | 0.545 ±.013 | 85.95 ±1.01 | 85.90 ±1.02 |
| | BYOL (90M) | 0.824 ±.006 | 0.476 ±.003 | 83.17 ±.006 | 83.10 ±.006 |
| | BrainLM (85M) | 0.648 ±.024 | 0.620 ±.018 | 89.31 ±1.13 | 89.29 ±1.12 |
| | BrainMass (90M) | 0.727 ±.015 | 0.573 ±.006 | 87.03 ±0.86 | 86.99 ±0.87 |
| | BDO (86M) | **0.481 ±.010** | **0.722 ±.007** | **92.59 ±0.68** | **92.57 ±0.69** |

### 4.1 INTERNAL AND EXTERNAL EVALUATION.

We assessed robustness and transferability using both linear probing (LP) and full fine-tuning (FT) protocols, with detailed procedures described in Appendix D.4. Notably, even under the simpler linear probing evaluation, BDO matched or exceeded the performance of task-specific models on both internal and external benchmarks, demonstrating strong generalizability across datasets.

**Internal tasks: Age and Gender Prediction.** To evaluate in-domain generalization, we performed age and gender prediction on a 20% held-out subset of the UKB data. As shown in Table 1, BDO achieved state-of-the-art results, surpassing both task-specific and SSL models on all metrics.

**External tasks: Trait and Diagnosis Prediction.** For external validation, we tested BDO on trait and psychiatric diagnosis prediction across four public datasets: HCP-A, ABIDE, ADHD200, and HCP-EP. Tasks included predicting demographics and cognitive scores in HCP-A as shown in Table 2, and classifying autism, ADHD, and schizophrenia in the other datasets as presented in Table 3. BDO outperformed all baselines in accuracy and F1 score, demonstrating strong clinical potential. Notably, we also confirmed that these powerful representations, pre-trained exclusively on resting-state fMRI, generalize effectively to task-based fMRI paradigms. See Appendix D.6 for the detailed analysis.

---

[3]Experimental details are provided in Appendix D.

Table 2: Demographics and trait prediction on HCP-A.

| | Methods | Age | | Gender | | Flanker | |
|---|---|---|---|---|---|---|---|
| | | MSE ↓ | $\rho$ ↑ | ACC (%) ↑ | F1 (%) ↑ | MSE ↓ | $\rho$ ↑ |
| TS | BrainNetCNN | 0.472 ±.054 | 0.727 ±.040 | 72.36 ±3.66 | 71.42 ±4.03 | 1.001 ±.097 | 0.310 ±.083 |
| | BrainGNN | 0.570 ±.050 | 0.657 ±.031 | 66.81 ±2.54 | 65.22 ±2.14 | 1.137 ±.049 | 0.229 ±.051 |
| | BrainNetTF | 0.389 ±.038 | 0.780 ±.036 | 75.00 ±2.28 | 74.06 ±2.78 | 0.959 ±.058 | 0.357 ±.071 |
| LP | MoCo (90M) | 0.817 ±.037 | 0.591 ±.007 | 64.12 ±1.18 | 64.06 ±1.25 | 1.572 ±.158 | 0.283 ±.022 |
| | BYOL (90M) | 0.609 ±.038 | 0.619 ±.030 | 64.81 ±3.32 | 64.58 ±3.49 | 0.960 ±.072 | 0.304 ±.071 |
| | BrainLM (85M) | 0.756 ±.057 | 0.636 ±.027 | 65.28 ±3.00 | 64.99 ±2.96 | 1.181 ±.081 | 0.375 ±.016 |
| | BrainMass (90M) | 0.743 ±.117 | 0.630 ±.077 | 66.20 ±0.65 | 66.17 ±0.56 | 1.082 ±.013 | 0.313 ±.013 |
| | BDO (86M) | **0.404 ±.010** | **0.768 ±.008** | **72.00 ±2.95** | **71.30 ±2.19** | **0.856 ±.049** | **0.450 ±.072** |
| FT | MoCo (90M) | 0.532 ±.023 | 0.697 ±.016 | 65.28 ±1.96 | 64.63 ±1.98 | 0.976 ±.063 | 0.370 ±.058 |
| | BYOL (90M) | 0.531 ±.037 | 0.694 ±.021 | 69.68 ±2.15 | 68.82 ±2.75 | 1.317 ±.031 | 0.216 ±.059 |
| | BrainLM (85M) | 0.340 ±.019 | 0.818 ±.012 | 72.78 ±2.12 | 72.36 ±2.22 | 0.859 ±.010 | 0.461 ±.015 |
| | BrainMass (90M) | 0.471 ±.030 | 0.728 ±.020 | 67.82 ±1.99 | 66.90 ±2.17 | 0.996 ±.055 | 0.339 ±.023 |
| | BDO (86M) | **0.273 ±.010** | **0.851 ±.006** | **79.40 ±4.07** | **78.98 ±4.38** | **0.847 ±.037** | **0.464 ±.072** |

Table 3: Psychiatric diagnosis prediction on clinical fMRI datasets.

| | Methods | ABIDE | | ADHD200 | | HCP-EP | |
|---|---|---|---|---|---|---|---|
| | | ACC (%) ↑ | F1 (%) ↑ | ACC (%) ↑ | F1 (%) ↑ | ACC (%) ↑ | F1 (%) ↑ |
| TS | BrainNetCNN | 64.39 ±2.17 | 64.23 ±2.27 | 55.49 ±4.39 | 53.62 ±5.15 | 70.29 ±6.90 | 58.07 ±9.52 |
| | BrainGNN | 56.82 ±3.40 | 56.73 ±3.43 | 52.78 ±3.27 | 51.59 ±2.89 | 73.14 ±6.90 | 65.46 ±9.06 |
| | BrainNetTF | 66.36 ±3.66 | 66.30 ±3.67 | 54.29 ±3.02 | 50.90 ±3.18 | 71.43 ±6.52 | 61.26 ±10.3 |
| LP | MoCo (90M) | 60.76 ±3.16 | 60.74 ±3.16 | 57.64 ±1.77 | 57.72 ±1.76 | 71.43 ±2.33 | 70.86 ±2.03 |
| | BYOL (90M) | 58.48 ±2.39 | 58.44 ±2.43 | 60.15 ±2.13 | 60.21 ±2.30 | 70.48 ±1.35 | 70.15 ±2.23 |
| | BrainLM (85M) | 59.92 ±3.88 | 59.77 ±3.77 | 60.44 ±2.37 | 60.27 ±2.27 | 72.38 ±3.56 | 73.04 ±3.10 |
| | BrainMass (90M) | 62.27 ±1.93 | 62.18 ±2.07 | 61.15 ±2.16 | 60.39 ±2.62 | 73.33 ±3.56 | 73.32 ±3.20 |
| | BDO (86M) | **66.67 ±1.13** | **66.58 ±1.02** | **61.40 ±1.97** | **61.49 ±2.92** | **76.19 ±4.86** | **74.63 ±4.96** |
| FT | MoCo (90M) | 65.45 ±2.26 | 65.16 ±2.30 | 61.15 ±1.28 | 59.39 ±2.13 | 75.24 ±1.35 | 74.78 ±2.37 |
| | BYOL (90M) | 60.76 ±1.30 | 60.61 ±1.41 | 58.15 ±0.35 | 57.16 ±0.53 | 72.38 ±1.35 | 73.08 ±1.06 |
| | BrainLM (85M) | 61.36 ±4.28 | 61.28 ±4.19 | 61.65 ±2.68 | 59.27 ±3.22 | 74.29 ±4.04 | 74.12 ±3.92 |
| | BrainMass (90M) | 67.27 ±3.34 | 66.66 ±3.46 | 63.91 ±1.23 | 62.55 ±0.74 | 76.19 ±3.56 | 76.25 ±1.97 |
| | BDO (86M) | **69.32 ±2.24** | **68.32 ±1.78** | **64.16 ±1.15** | **64.27 ±1.08** | **82.86 ±2.86** | **82.87 ±2.65** |

## 4.2 SCALABILITY AND EFFICIENCY.

**Scalability.** We evaluated BDO's scalability by varying both model size and pre-training data volume as depicted in Figure 7. For model scaling, we pre-trained three BDO variants with increasing parameter sizes: 5M, 21M, and 86M, and tracked HCP-A age-regression performance trajectories over pre-training epochs as well as disease prediction accuracy across various datasets. Additionally, to examine the effect of pre-training data volume, we trained BDO (86M) on progressively larger subsets of the UKB dataset. In both experiments, larger models and greater data sizes clearly yielded higher performance. Detailed performance metrics and spatial scalability analysis are available in Appendix D.7.1 and D.7.2, respectively.

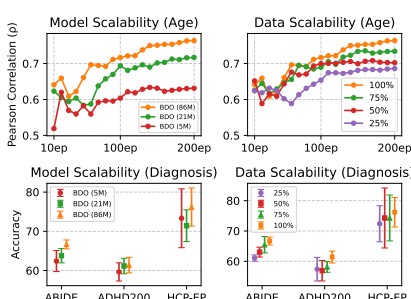

Figure 7: **(Left)** Model scalability **(Right)** Data scalability on HCP-A age regression and diagnosis prediction.

**Efficiency.** Figure 1 shows that BDO significantly outperforms other SSL methods in both resource and parameter efficiency. Remarkably, even the smallest BDO variant (5M) achieves HCP-A age regression performance comparable to other SSL models (He et al., 2020; Grill et al., 2020; Caro et al., 2024; Yang et al., 2024). See Appendix D.8 for the detailed metrics and experimental setup.

### 4.3 ABLATION STUDY.

**Mask ratio $\gamma$.** Figure 8 illustrates the effect of the mask ratio $\gamma$ for MAE in Section 3.2. The performance increases with $\gamma$, reaching an optimal peak at 75%. Beyond this point, performance degrades, likely due to excessive information loss hindering reconstruction. Additionally, a detailed analysis of the importance of the MAE objective itself can be found in Appendix D.9.

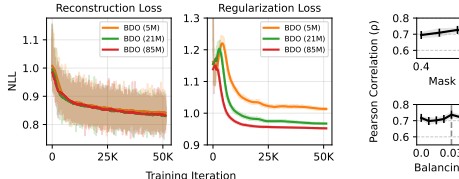

Figure 8: **(Left)** Training curve **(Right)** HCP-A age regression Pearson correlation $\rho$ as the mask ratio $\gamma$ and balancing factor $\tau$ are varied.

**Balancing factor $\tau$.** Our ablation on the balancing factor $\tau$ in (9) reveals that incorporating the JEPA regularizer ($\tau > 0$) enhances performance over the MAE-only setting ($\tau = 0$), as shown in Figure 8. This demonstrates that the empirical prior contributes to learning more robust and meaningful representations. See Appendix D.10 for a more comprehensive MAE–JEPA ablation.

**Number of timesteps.** During the pre-training, BDO learns from sequence segments randomly sampled from full fMRI recordings. The length of these segments, referred to as the number of timesteps, directly impacts downstream performance, as shown in Table 4. While using more timesteps consistently boosts performance, likely by capturing richer brain dynamics, it also raises the computational cost of pre-training stage.

Table 4: LP performance of BDO (86M) with a varying number of timesteps (NoTs) on HCP-A.

| NoTs | Age (MSE) ↓ | Age ($\rho$) ↑ | Gender (ACC) ↑ | Gender (F1) ↑ |
|---|---|---|---|---|
| 80 | 0.587 ±.045 | 0.645 ±.026 | 63.67 ±2.00 | 61.98 ±0.92 |
| 160 | 0.404 ±.010 | 0.768 ±.008 | 72.00 ±2.95 | 71.30 ±2.19 |
| 240 | 0.348 ±.015 | 0.805 ±.009 | 72.22 ±1.13 | 71.34 ±1.35 |

### 4.4 INTERPRETABILITY OF THE UNIVERSAL FEATURE $\mathbb{A}$

**Embedding Space of Universal Feature $\mathbb{A}$.** To evaluate the clinical relevance of the universal feature $\mathbb{A}$, we projected $\mathbb{A}$ into a two-dimensional space using PCA and UMAP, as shown in Figure 9. PCA revealed clear linear patterns closely aligned with age distributions, while UMAP preserved these meaningful separations. This indicates that $\mathbb{A}$ successfully encodes biologically meaningful neural variations associated with aging. The ability to characterize aging is clinically critical, as deviations from typical aging trajectories may signal early vulnerability to cognitive decline or psychiatric disorders (Elliott et al., 2021). These results demonstrate that our framework learns meaningful representations reflecting aging-related neural dynamics, highlighting its clinical potential. Further qualitative analyses, covering feature attribution and latent dynamics, are detailed in Appendix D.11.

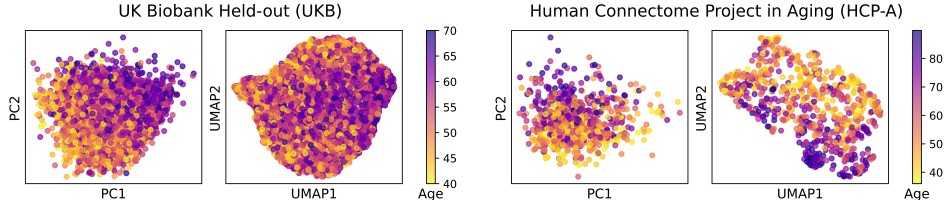

Figure 9: Projected 2D features of the universal feature $\mathbb{A}$ using PCA and UMAP, coloured by age. **(Left)** Embedding space of UKB held-out split. **(Right)** Embedding space of HCP-A dataset.

## 5 CONCLUSION AND LIMITATIONS

We introduced BDO, a novel SSL framework leveraging SOC to model continuous latent brain dynamics from fMRI, yielding a robust and transferable universal feature $\mathbb{A}$. BDO achieves state-of-the-art performance across diverse demographic and clinical downstream tasks, demonstrating strong generalization to external cohorts and notable computational efficiency.

While our simulation-free inference based on locally linear approximations has proven effective for scalability, it may introduce a variational gap and potential error accumulation over long-term temporal analyses. Further challenges include the inherent complexity of the BDO framework and achieving a detailed neurobiological interpretation of learned continuous dynamics beyond the aggregated universal feature $\mathbb{A}$. Future work may focus on developing more flexible inference approximations and enhanced interpretability methods for such sophisticated models of brain activity.

## ACKNOWLEDGMENTS

This research was supported by NRF grant funded by the Korea government (MSIT) (RS-2021-NR056917) and the Bio & Medical Technology Development Program of the NRF funded by the MSIT (RS-2025-02263045).

This work was also supported by Institute of Information & Communications Technology Planning & Evaluation (IITP) grant funded by the Korea government (MSIT) (No. RS-2019-II190075, Artificial Intelligence Graduate School Program (KAIST); No. RS-2022-II220713, Meta-learning Applicable to Real-world Problems; No. RS-2024-00509279, Global AI Frontier Lab), the National Institute of Health (NIH) research project (No. 2025-ER0806-00), and a grant of the Korean ARPH-H Project through the Korea Health Industry Development Institute (KHIDI), funded by the Ministry of Health & Welfare, Republic of Korea (RS-2025-25455095).

## REPRODUCIBILITY STATEMENT.

Appendix D contains all implementation details required for reproducibility, including a full breakdown of our datasets, preprocessing pipelines, network architectures, and training protocols with hyperparameters. Our source code is available in the supplementary material.

## ETHICS STATEMENT.

Our work introduces a novel SSL framework for fMRI analysis, intended strictly as a tool to advance neuroscience research in areas such as demographic trait analysis and psychiatric diagnosis. We acknowledge the significant ethical responsibilities associated with such technology, particularly the risks of perpetuating societal biases from training data and the potential for misapplication in clinical settings. Crucially, this model is not a validated diagnostic tool and should not be used for such purposes without rigorous clinical trials and expert oversight. This study relies on publicly available, anonymized data; any future applications on sensitive clinical data would require strict privacy protocols and institutional review. A large language model (LLM) was used solely to improve the manuscript's grammar and clarity, and not for generating scientific ideas or results.

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

## A    RELATED WORK

To facilitate a detailed understanding of the theoretical foundations, we present an additional related work section on self-supervised learning and stochastic optimal control for continuous-time dynamics.

**Self-Supervised Learning.**    SSL has emerged as a powerful paradigm for extracting rich, generalizable representations from large-scale unlabeled data. In the vision domain, prominent approaches fall into two main categories: latent representation alignment and reconstruction. Alignment-based methods, such as MoCo (He et al., 2020) and BYOL (Grill et al., 2020), learn invariant features by maximizing agreement between different views of the same instance, often utilizing contrastive loss or momentum bootstrapping. Conversely, reconstruction-based frameworks like MAE (He et al., 2022) force the model to infer missing patches from partial observations, encouraging the capture of global structure. More recently, JEPA has demonstrated the effectiveness of predicting latent representations rather than raw pixels (Assran et al., 2023; Bardes et al., 2024), thereby learning high-level semantics while avoiding low-level noise.

**Extended Review of Self-Supervised Learning in fMRI.**    Image-based SSL approaches for fMRI, which patchify fMRI time-series into spatiotemporal tokens, represent each scan as a fixed grid of spatiotemporal tokens. BrainLM pioneered the image-style paradigm and used a MAE objective to reconstruct masked patches (Caro et al., 2024), and Brain-JEPA builds on this by replacing reconstruction with a JEPA that learns to predict the latent embedding of one view from another (Dong et al., 2024). Beyond the loss of fine-grained temporal dynamics during patchification and the difficulty of aligning temporal resolution across datasets as pointed out in Section 2, a key limitation of this design is its rigid dependence on sequence length in the SSL scenario. For example, a model pre-trained on UKB data with 160 time points grouped into 10 non-overlapping patches of length 16 expects exactly the same 10-token layout during inference. However, datasets such as ABIDE contain scans with as few as 76 time points, which cannot be processed without upsampling. This length rigidity undermines the core premise of SSL, namely broad out-of-the-box transfer across heterogeneous unlabeled cohorts, and therefore poses a critical limitation for SSL in fMRI.

In contrast, graph-based approaches compress the fMRI time-series into a single static functional-connectivity (FC) graph, freeing them from constraints such as heterogeneous temporal resolution or scan-length variability; however, this coarse summarization inevitably discards fine-grained temporal dynamics in BOLD signals. BrainMass, for example, generates augmented static graphs by randomly dropping time points to produce pseudo-FC variants, and leverages latent representation alignment and masked-ROI prediction (Yang et al., 2024). ST-JEMA tries to restore temporal information by partitioning the scan into multiple sliding windows and building a dynamic graph sequence on which it performs JEPA-style SSL objective (Choi et al., 2024). While this design captures slow changes in connectivity, it still averages out intra-window dynamics and therefore inherits the same limitations as image-based patchification such as loss of high-frequency temporal details and the practical difficulty of matching window size to the diverse TRs across different datasets as pointed out in Section 1.

**Stochastic Optimal Control for Continuous-Time Dynamics.**    SOC provides a rigorous mathematical foundation for optimizing policies in stochastic systems governed by SDEs. In the context of generative modeling, Li et al. (2020) bridged SOC and variational inference by deriving the ELBO to optimize non-linear latent dynamics. This control-theoretic perspective was subsequently adapted to approximate smoothing distributions in various settings (Heng et al., 2020; Lu & Wang, 2024) and to estimate Doob's $h$-transform for online filtering (Chopin et al., 2023). Building on these developments, Park et al. (2024) recently leveraged SOC to enable efficient amortized inference.

While previous works primarily utilized SOC for efficient inference or filtering, we extend SOC to the domain of representation learning by fundamentally reframing the optimal control policy as a latent encoder. By framing the SSL objective as a control cost, our framework naturally integrates masked autoencoding (MAE) and joint-embedding prediction (JEPA) tasks into a single continuous-time optimization problem, enabling robust feature extraction from irregular fMRI data.

# B PROOFS AND DERIVATIONS

## B.1 PROOF OF PROPOSITION 3.1

*Proof.* Bayesian filtering and smoothing techniques Särkkä (2013) are foundational tools for estimating latent states in probabilistic dynamical systems. The goal is to recover the posterior distribution of the latent trajectories given the observations:

$$p(\mathbf{X}_{[0,T]} \mid \mathcal{Y}) = \frac{1}{Z(\mathcal{Y})} \prod_{t \in \mathcal{T}} g_\psi(\mathbf{y}_{t_i} | \mathbf{X}_{t_i}) p(\mathbf{X}_{[0,T]}), \tag{12}$$

where $Z(\mathcal{Y})$ is a normalization constant:

$$Z(\mathcal{Y}) = \mathbb{E}_{\mathbf{X} \sim (1)} \left[ \prod_{t \in \mathcal{T}} g_\psi(\mathbf{y}_{t_i} | \mathbf{X}_{t_i}) \right]. \tag{13}$$

Expanding this expectation, we have

$$\log \mathbf{Z}(\mathcal{Y}) = \log \mathbb{E}_{\mathbf{X} \sim (1)} \left[ p(\mathcal{Y} | \mathbf{X}_{[0,T]}) \right] \tag{14}$$

$$= \log \mathbb{E}_{\mathbf{X} \sim (1)} \left[ p(\mathcal{Y} | \mathbf{X}_{[0,T]}^\alpha) \frac{p(\mathbf{X}_{[0,T]})}{p(\mathbf{X}_{[0,T]}^\alpha)} \right] \tag{15}$$

$$\overset{(i)}{\geq} \mathbb{E}_{\mathbf{X} \sim (10)} \left[ \log p(\mathcal{Y} | \mathbf{X}_{[0,T]}^\alpha) + \log \frac{p(\mathbf{X}_{[0,T]})}{p(\mathbf{X}_{[0,T]}^\alpha)} \right] \tag{16}$$

$$= \mathbb{E}_{\mathbf{X} \sim (10)} \left[ \sum_{t \in \mathcal{T}} g(\mathbf{y}_t | \mathbf{X}_t^\alpha) + \log \frac{p(\mathbf{X}_{[0,T]})}{p(\mathbf{X}_{[0,T]}^\alpha)} \right] \tag{17}$$

$$\overset{(ii)}{=} \mathbb{E}_{\mathbf{X} \sim (10)} \left[ \sum_{t \in \mathcal{T}} g(\mathbf{y}_t | \mathbf{X}_t^\alpha) - \frac{1}{2} \int_0^T \|\alpha_t\|^2 \, dt + \int_0^1 \alpha_t d\mathbf{W}_s \right] \tag{18}$$

$$\overset{(iii)}{=} \mathbb{E}_{\mathbf{X} \sim (10)} \left[ \sum_{t \in \mathcal{T}} g(\mathbf{y}_t | \mathbf{X}_t^\alpha) - \frac{1}{2} \int_0^T \|\alpha_t\|^2 \, dt \right] \tag{19}$$

$$= -\mathcal{J}(\alpha, \mathcal{Y}), \tag{20}$$

where $(i)$ results from Jensen's inequality, $(ii)$ follows by applying Girsanov's theorem (Baldi, 2017, Theorem 12.1), and in the final equality, $(iii)$ holds because $\mathbf{W}_t$ is a martingale process.

□

## B.2 PROOF OF THEOREM 3.2

*Proof.* Since each SPD matrix $\mathbf{D}_t$ for $t \in \mathcal{T}$ admits an eigen-decomposition $\mathbf{D}_{t_i} = \mathbf{V} \mathbf{\Lambda}_{t_i} \mathbf{V}^\top$, we can transform the original process $\mathbf{X}_t^\alpha$, which is expressed in the canonical basis, into a new process $\hat{\mathbf{X}}_t^\alpha = \mathbf{V}^\top \mathbf{X}_t^\alpha$ that resides in the space spanned by the eigenbasis $\mathbf{V}$. With this transformation, the dynamics in (10) can be rewritten, for any interval $[t_i, t_{i+1}]$, as:

$$d\hat{\mathbf{X}}_t^\alpha = \left[ -\mathbf{\Lambda}_{t_i} \hat{\mathbf{X}}_t^\alpha + \alpha_{t_i} \right] dt + d\hat{\mathbf{W}}_t, \tag{21}$$

where $\hat{\mathbf{X}}_t^\alpha = \mathbf{V}^\top \mathbf{X}_t^\alpha$, $\hat{\alpha}_{t_i} = \mathbf{V}^\top \alpha_{t_i}$, $\hat{\mathbf{W}}_t = \mathbf{V}^\top \mathbf{W}_t$ and initial condition $\hat{\mathbf{X}}_0^\alpha \sim \mathcal{N}(\hat{\mu}_0, \hat{\Sigma}_0)$ with $\hat{\mu}_0 = \mathbf{V}^\top \mu_0$ and $\hat{\Sigma}_0 = \mathbf{V}^\top \Sigma_0 \mathbf{V}$. Since $\mathbf{V}$ is orthonormal, $\hat{\mathbf{W}}_t$ retains the distribution $\hat{\mathbf{W}}_t \overset{d}{=} \mathbf{W}_t$ for all $t \in [0, T]$, allowing $\hat{\mathbf{W}}_t$ to be treated as a standard Wiener process. Now, given that $\mathbf{\Lambda}_{t_i}$ is diagonal, the linear SDE in equation (21) admits a closed-form solution for any $t \in [t_i, t_{i+1})$:

$$\hat{\mathbf{X}}_t^\alpha = e^{-(t-t_i)\mathbf{\Lambda}_{t_i}} \left( \hat{\mathbf{X}}_{t_i}^\alpha + \int_{t_i}^t e^{(s-t_i)\mathbf{\Lambda}_{t_i}} \hat{\alpha}_{t_i} \, ds + \int_{t_i}^t e^{(s-t_i)\mathbf{\Lambda}_{t_i}} \, d\hat{\mathbf{W}}_s \right). \tag{22}$$

Since the initial condition $\hat{\mathbf{X}}_0^\alpha$ is Gaussian and the SDE is linear with Gaussian noise, the process $\hat{\mathbf{X}}_t^\alpha$ remains Gaussian. Therefore, its first two moments—the mean and covariance—can be derived from

the solution above. To derive the moments, we firstly evaluate the deterministic integral involving $\hat{\alpha}_{t_i}$:

$$\int_{t_i}^{t} e^{(s-t_i)\mathbf{\Lambda}_{t_i}} \hat{\alpha}_{t_i} \, ds = -\mathbf{\Lambda}_{t_i}^{-1} \left( \mathbf{I} - e^{(t-t_i)\mathbf{\Lambda}_{t_i}} \right) \hat{\alpha}_{t_i}. \tag{23}$$

Taking the expectation of $\hat{\mathbf{X}}_t^{\alpha}$, and using the martingale property of the Wiener process $\hat{\mathbf{W}}_t$, we obtain:

$$\hat{\mu}_t = \mathbb{E}_{\hat{\mathbf{X}}^{\alpha} \sim (21)} \left[ \hat{\mathbf{X}}_t^{\alpha} \right] = e^{-(t-t_i)\mathbf{\Lambda}_{t_i}} \hat{\mu}_{t_i} - e^{-(t-t_i)\mathbf{\Lambda}_{t_i}} \mathbf{\Lambda}_{t_i}^{-1} \left( \mathbf{I} - e^{(t-t_i)\mathbf{\Lambda}_{t_i}} \right) \hat{\alpha}_{t_i}. \tag{24}$$

Next, compute the covariance of $\hat{\mathbf{X}}_t^{\alpha}$:

$$\hat{\Sigma}_t = \mathbb{E}_{\hat{\mathbf{X}}^{\alpha} \sim (21)} \left[ e^{-2(t-t_i)\mathbf{\Lambda}_{t_i}} \left( \mathbf{X}_{t_i} - \mu_{t_i} + \int_{t_i}^{t} e^{(s-t_i)\mathbf{\Lambda}_{t_i}} d\hat{\mathbf{W}}_s \right) \left( \mathbf{X}_{t_i} - \mu_{t_i} + \int_{t_i}^{t} e^{(s-t_i)\mathbf{\Lambda}_{t_i}} d\hat{\mathbf{W}}_s \right)^{\top} \right] \tag{25}$$

$$= e^{-2(t-t_i)\mathbf{\Lambda}_{t_i}} \mathbb{E}_{\hat{\mathbf{X}}^{\alpha} \sim (21)} \left[ (\mathbf{X}_{t_i} - \mu_{t_i}) (\mathbf{X}_{t_i} - \mu_{t_i})^{\top} + \left\| \int_{t_i}^{t} e^{(s-t_i)\mathbf{\Lambda}_{t_i}} d\hat{\mathbf{W}}_s \right\|_2^2 \right] \tag{26}$$

$$\stackrel{(i)}{=} e^{-2(t-t_i)\mathbf{\Lambda}_{t_i}} \mathbb{E}_{\hat{\mathbf{X}}^{\alpha} \sim (21)} \left[ (\mathbf{X}_{t_i} - \mu_{t_i}) (\mathbf{X}_{t_i} - \mu_{t_i})^{\top} + \int_{t_i}^{t} e^{2(s-t_i)\mathbf{\Lambda}_{t_i}} ds \right] \tag{27}$$

$$\stackrel{(ii)}{=} e^{-2(t-t_i)\mathbf{\Lambda}_{t_i}} \hat{\Sigma}_{t_i} - \frac{1}{2} e^{-2(t-t_i)\mathbf{\Lambda}_{t_i}} \mathbf{\Lambda}_{t_i}^{-1} \left( \mathbf{I} - e^{2(t-t_i)\mathbf{\Lambda}_{t_i}} \right), \tag{28}$$

where $(i)$ follows from the martingale property of $\hat{\mathbf{W}}_t$ and $(ii)$ follows from Itô isometry:

$$\mathbb{E}_{\hat{\mathbf{X}}^{\alpha} \sim (21)} \left[ \left\| \int_{t_i}^{t} e^{(s-t_i)\mathbf{\Lambda}_{t_i}} d\hat{\mathbf{W}}_s \right\|_2^2 \right] = \mathbb{E}_{\hat{\mathbf{X}}^{\alpha} \sim (21)} \left[ \int_{t_i}^{t} e^{2(s-t_i)\mathbf{\Lambda}_{t_i}} ds \right]. \tag{29}$$

Using the recursive forms for the mean and covariance, we can determine these moments at each discrete time step $t_i$. For the mean $\hat{\mu}_{t_i}$, the recurrence relation is:

$$\hat{\mu}_{t_1} = e^{-(t_1-t_0)\mathbf{\Lambda}_{t_0}} \hat{\mu}_{t_0} - e^{-(t_1-t_0)\mathbf{\Lambda}_{t_1}} \mathbf{\Lambda}_{t_0}^{-1} \left( \mathbf{I} - e^{(t_1-t_0)\mathbf{\Lambda}_{t_0}} \right) \hat{\alpha}_{t_0} \tag{30}$$

$$\hat{\mu}_{t_2} = e^{-\sum_{j=0}^{1}(t_{j+1}-t_j)\mathbf{\Lambda}_{t_j}} \hat{\mu}_{t_0} \tag{31}$$
$$- e^{-\sum_{j=0}^{1}(t_{j+1}-t_j)\mathbf{\Lambda}_{t_j}} \mathbf{\Lambda}_{t_0}^{-1} \left( \mathbf{I} - e^{(t_1-t_0)\mathbf{\Lambda}_{t_0}} \right) \hat{\alpha}_{t_0}$$
$$- e^{-(t_2-t_1)\mathbf{\Lambda}_{t_1}} \mathbf{\Lambda}_{t_1}^{-1} \left( \mathbf{I} - e^{(t_2-t_1)\mathbf{\Lambda}_{t_1}} \right) \hat{\alpha}_{t_1}$$

$$\vdots$$

$$\hat{\mu}_{t_i} = e^{-\sum_{j=0}^{i-1}(t_{j+1}-t_j)\mathbf{\Lambda}_{t_j}} \hat{\mu}_{t_0} - \sum_{l=0}^{i-1} e^{-\sum_{j=l}^{i-1}(t_{j+1}-t_j)\mathbf{\Lambda}_{t_j}} \mathbf{\Lambda}_{t_l}^{-1} \left( \mathbf{I} - e^{(t_{l+1}-t_l)\mathbf{\Lambda}_{t_l}} \right) \hat{\alpha}_{t_l} \tag{32}$$

Similarly, for the covariance $\hat{\Sigma}_{t_i}$, the recurrence relation is:

$$\hat{\Sigma}_{t_1} = e^{-2(t_1-t_0)\mathbf{\Lambda}_{t_0}} \hat{\Sigma}_{t_0} - \frac{1}{2} e^{-2(t_1-t_0)\mathbf{\Lambda}_{t_1}} \mathbf{\Lambda}_{t_0}^{-1} \left( \mathbf{I} - e^{2(t_1-t_0)\mathbf{\Lambda}_{t_0}} \right) \tag{33}$$

$$\hat{\Sigma}_{t_2} = e^{-\sum_{j=0}^{1} 2(t_{j+1}-t_j)\mathbf{\Lambda}_{t_j}} \hat{\Sigma}_{t_0} \tag{34}$$
$$- \frac{1}{2} e^{-\sum_{j=0}^{1} 2(t_{j+1}-t_j)\mathbf{\Lambda}_{t_j}} \mathbf{\Lambda}_{t_0}^{-1} \left( \mathbf{I} - e^{2(t_1-t_0)\mathbf{\Lambda}_{t_0}} \right) - \frac{1}{2} e^{-2(t_2-t_1)\mathbf{\Lambda}_{t_1}} \mathbf{\Lambda}_{t_1}^{-1} \left( \mathbf{I} - e^{2(t_2-t_1)\mathbf{\Lambda}_{t_1}} \right)$$

$$\vdots$$

$$\hat{\Sigma}_{t_i} = e^{-2\sum_{j=0}^{i-1}(t_{j+1}-t_j)\mathbf{\Lambda}_{t_j}} \hat{\Sigma}_{t_0} - \frac{1}{2} \sum_{l=0}^{i-1} e^{-2\sum_{j=l}^{i-1}(t_{j+1}-t_j)\mathbf{\Lambda}_{t_j}} \mathbf{\Lambda}_{t_l}^{-1} \left( \mathbf{I} - e^{2(t_{l+1}-t_l)\mathbf{\Lambda}_{t_l}} \right). \tag{35}$$

Now, since $\hat{\mathbf{X}}_t^\alpha = \mathbf{V}^\top \mathbf{X}_t^\alpha$, with $\hat{\mu}_0 = \mathbf{V}^\top \mu_0$ and $\hat{\Sigma}_0 = \mathbf{V}^\top \Sigma_0 \mathbf{V}$, we can express the mean and covariance in the original canonical basis as follows. For the mean $\hat{\mu}_{t \in \mathcal{T}}$, which is given by

$$
\mathbf{V}\hat{\mu}_{t_i} = \mathbf{V}\left(e^{-\sum_{j=0}^{i-1}(t_{j+1}-t_j)\mathbf{\Lambda}_{t_j}}\hat{\mu}_{t_0} - \sum_{l=0}^{i-1} e^{-\sum_{j=l}^{i-1}(t_{j+1}-t_j)\mathbf{\Lambda}_{t_j}}\mathbf{\Lambda}_{t_l}^{-1}\left(\mathbf{I} - e^{(t_{l+1}-t_l)\mathbf{\Lambda}_{t_l}}\right)\hat{\alpha}_{t_l}\right)
$$
(36)

$$
= \mathbf{V}\left(e^{-\sum_{j=0}^{i-1}(t_{j+1}-t_j)\mathbf{\Lambda}_{t_j}}\mathbf{V}^\top\mu_0 - \sum_{l=0}^{i-1} e^{-\sum_{j=l}^{i-1}(t_{j+1}-t_j)\mathbf{\Lambda}_{t_j}}\mathbf{\Lambda}_{t_l}^{-1}\left(\mathbf{I} - e^{(t_{l+1}-t_l)\mathbf{\Lambda}_{t_l}}\right)\mathbf{V}^\top\alpha_{t_l}\right)
$$
(37)

$$
= e^{-\sum_{j=0}^{i-1}(t_{j+1}-t_j)\mathbf{D}_{t_j}}\mu_0 - \mathbf{V}\left(\sum_{l=0}^{i-1} e^{-\sum_{j=l}^{i-1}(t_{j+1}-t_j)\mathbf{\Lambda}_{t_j}}\mathbf{\Lambda}_{t_l}^{-1}\left(\mathbf{I} - e^{(t_{l+1}-t_l)\mathbf{\Lambda}_{t_l}}\right)\mathbf{V}^\top\alpha_{t_l}\right)
$$
(38)

$$
= e^{-\sum_{j=0}^{i-1}(t_{j+1}-t_j)\mathbf{D}_{t_j}}\mu_0 - \mathbf{V}\left(\sum_{l=0}^{i-1} e^{-\sum_{j=l}^{i-1}(t_{j+1}-t_j)\mathbf{\Lambda}_{t_j}}\mathbf{V}^\top\mathbf{D}_{t_l}^{-1}\mathbf{V}\left(\mathbf{I} - e^{(t_{l+1}-t_l)\mathbf{\Lambda}_{t_l}}\right)\mathbf{V}^\top\alpha_{t_l}\right)
$$
(39)

$$
= e^{-\sum_{j=0}^{i-1}(t_{j+1}-t_j)\mathbf{D}_{t_j}}\mu_0 - \sum_{l=0}^{i-1} e^{-\sum_{j=l}^{i-1}(t_{j+1}-t_j)\mathbf{D}_{t_j}}\mathbf{D}_{t_l}^{-1}\left(\mathbf{I} - e^{(t_{l+1}-t_l)\mathbf{D}_{t_l}}\right)\alpha_{t_l}
$$
(40)

$$
= \mu_{t_i}
$$
(41)

where we used $\mathbf{D}_{t_j} = \mathbf{V}\mathbf{\Lambda}_{t_j}\mathbf{V}^\top$ and the orthonormality of $\mathbf{V}$. Similarly, for the covariance $\hat{\Sigma}_{t \in \mathcal{T}}$, we have

$$
\mathbf{V}\hat{\Sigma}_{t_i}\mathbf{V}^\top = \mathbf{V}\left(e^{-2\sum_{j=0}^{i-1}(t_{j+1}-t_j)\mathbf{\Lambda}_{t_j}}\hat{\Sigma}_{t_0} - \frac{1}{2}\sum_{l=0}^{i-1} e^{-2\sum_{j=l}^{i-1}(t_{j+1}-t_j)\mathbf{\Lambda}_{t_j}}\mathbf{\Lambda}_{t_l}^{-1}\left(\mathbf{I} - e^{2(t_{l+1}-t_l)\mathbf{\Lambda}_{t_l}}\right)\right)\mathbf{V}^\top
$$
(42)

$$
= \mathbf{V}\left(e^{-2\sum_{j=0}^{i-1}(t_{j+1}-t_j)\mathbf{\Lambda}_{t_j}}\mathbf{V}^\top\Sigma_0\mathbf{V} - \frac{1}{2}\sum_{l=0}^{i-1} e^{-2\sum_{j=l}^{i-1}(t_{j+1}-t_j)\mathbf{\Lambda}_{t_j}}\mathbf{\Lambda}_{t_l}^{-1}\left(\mathbf{I} - e^{2(t_{l+1}-t_l)\mathbf{\Lambda}_{t_l}}\right)\right)\mathbf{V}^\top
$$
(43)

$$
= e^{-2\sum_{j=0}^{i-1}(t_{j+1}-t_j)\mathbf{D}_{t_j}}\Sigma_0 - \mathbf{V}\left(\frac{1}{2}\sum_{l=0}^{i-1} e^{-2\sum_{j=l}^{i-1}(t_{j+1}-t_j)\mathbf{\Lambda}_{t_j}}\mathbf{\Lambda}_{t_l}^{-1}\left(\mathbf{I} - e^{2(t_{l+1}-t_l)\mathbf{\Lambda}_{t_l}}\right)\right)\mathbf{V}^\top
$$
(44)

$$
= e^{-2\sum_{j=0}^{i-1}(t_{j+1}-t_j)\mathbf{D}_{t_j}}\Sigma_0 - \mathbf{V}\left(\sum_{l=0}^{i-1} e^{-2\sum_{j=l}^{i-1}(t_{j+1}-t_j)\mathbf{\Lambda}_{t_j}}\mathbf{V}^\top\mathbf{D}_{t_l}^{-1}\mathbf{V}\left(\mathbf{I} - e^{2(t_{l+1}-t_l)\mathbf{\Lambda}_{t_l}}\right)\right)\mathbf{V}^\top
$$
(45)

$$
= e^{-2\sum_{j=0}^{i-1}(t_{j+1}-t_j)\mathbf{D}_{t_j}}\Sigma_0 - \frac{1}{2}\sum_{l=0}^{i-1} e^{-2\sum_{j=l}^{i-1}(t_{j+1}-t_j)\mathbf{D}_{t_j}}\mathbf{D}_{t_l}^{-1}\left(\mathbf{I} - e^{2(t_{l+1}-t_l)\mathbf{D}_{t_l}}\right)
$$
(46)

$$
= \Sigma_{t_i}
$$
(47)

Thus, both the mean $\mu_{t_i}$ and the covariance $\Sigma_{t_i}$ of $\mathbf{X}_t^\alpha$ at each time step $t_i$ are correctly recovered, completing the proof. $\square$

## B.3   DERIVATION OF ELBO

We start the derivation by integrating the mixture distribution in (8) into the SOC problem (6) as follows:

$$\log p(\mathcal{Y}_{\text{tar}}|\mathbf{X}_{[0,T]}^\theta) = \log \int \zeta_\psi(\mathbf{y}_t|\tilde{\alpha}_t)\pi_{\bar{\theta}}(\tilde{\alpha}_t|\mathbf{X}_t^\theta)d\tilde{\alpha}_t \tag{48}$$

$$= \log \int \zeta_\psi(\mathbf{y}_t|\tilde{\alpha}_t)\frac{1}{\mathbf{Z}(\mathbf{X}_t^\theta)}\left[p_\theta(\tilde{\alpha}_t|\mathbf{X}_t^\theta)^\lambda q_{\bar{\theta}}(\tilde{\alpha}_t|\mathcal{Y}_{\text{tar}})^{1-\lambda}\right]d\tilde{\alpha}_t \tag{49}$$

$$= \log \int \zeta_\psi(\mathbf{y}_t|\tilde{\alpha}_t)\left[\frac{p_\theta(\tilde{\alpha}_t|\mathbf{X}_t^\theta)^\lambda q_{\bar{\theta}}(\tilde{\alpha}_t|\mathcal{Y}_{\text{tar}})^{1-\lambda}}{\mathbf{Z}(\mathbf{X}_t^\theta)h(\tilde{\alpha}_t)}\right]h(\tilde{\alpha}_t)d\tilde{\alpha}_t - \log \mathbf{Z}(\mathbf{X}_t^\theta) \tag{50}$$

$$\overset{(i)}{\geq} \int \left[\log \zeta_\psi(\mathbf{y}_t|\tilde{\alpha}_t) + \lambda \log p_\theta(\tilde{\alpha}_t|\mathbf{X}_t^\theta) + (1-\lambda)\log q_{\bar{\theta}}(\tilde{\alpha}_t|\mathcal{Y}_{\text{tar}}) - \log h(\tilde{\alpha}_t)\right]h(\tilde{\alpha}_t)d\tilde{\alpha}_t - \log \mathbf{Z}(\mathbf{X}_t^\theta) \tag{51}$$

$$\overset{(ii)}{=} \int \left[\log \zeta_\psi(\mathbf{y}_t|\tilde{\alpha}_t) + (\lambda-1)\log p_\theta(\tilde{\alpha}_t|\mathbf{X}_t^\theta) + (1-\lambda)\log q_{\bar{\theta}}(\tilde{\alpha}_t|\mathcal{Y}_{\text{tar}})\right]p_\theta(\tilde{\alpha}_t|\mathbf{X}_t^\theta)d\tilde{\alpha}_t - \log \mathbf{Z}(\mathbf{X}_t^\theta) \tag{52}$$

$$\overset{(iii)}{=} \int \left[\log \zeta_\psi(\mathbf{y}_t|\tilde{\alpha}_t) + (1-\lambda)\log q_{\bar{\theta}}(\tilde{\alpha}_t|\mathcal{Y}_{\text{tar}})\right]p_\theta(\tilde{\alpha}_t|\mathbf{X}_t^\theta)d\tilde{\alpha}_t + (1-\lambda)C - \log \mathbf{Z}(\mathbf{X}_t^\theta) \tag{53}$$

$$\overset{(iv)}{\geq} \mathbb{E}_{\tilde{\alpha}_t \sim p(\tilde{\alpha}_t|\mathbf{X}_t^\theta)}\Bigg[\underbrace{\log \zeta_\psi(\mathbf{y}_t|\tilde{\alpha}_t)}_{\text{MAE}} + (1-\lambda)\underbrace{\log q_{\bar{\theta}}(\tilde{\alpha}_t|\mathcal{Y}_{\text{tar}})}_{\text{JEPA}}\Bigg] \tag{54}$$

$$= \mathbb{E}_{\tilde{\alpha}_t \sim p(\tilde{\alpha}_t|\mathbf{X}_t^\theta)}\left[\frac{1}{2\sigma_\zeta^2}\|\mathbf{y}_t - \mathbf{D}_\psi(\tilde{\alpha}_t)\|^2 + \frac{(1-\lambda)}{2\sigma_q^2}\left\|\tilde{\alpha}_t - \alpha^{\bar{\theta}}\right\|^2\right], \tag{55}$$

where $(i)$ follows from Jensen's inequality, and $(ii)$ follows by setting proposal distribution $h = p_\theta$, $(iii)$ follows from the definition of $p_\theta$, since the entropy of Gaussian with constant covariance:

$$\int (\lambda-1)\log p_\theta(\tilde{\alpha}_t|\mathbf{X}_t^\theta)p_\theta(\tilde{\alpha}_t|\mathbf{X}_t^\theta)d\tilde{\alpha}_t = (1-\lambda)\int -\log p_\theta(\tilde{\alpha}_t|\mathbf{X}_t^\theta)p_\theta(\tilde{\alpha}_t|\mathbf{X}_t^\theta)d\tilde{\alpha}_t = (1-\lambda)C \geq 0. \tag{56}$$

Finally, $(iv)$ follows from $(1-\lambda)C \geq 0$ and since the normalization constant $\mathbf{Z}(\mathbf{X}_t^\theta)$ is calculated as:

$$\mathbf{Z}(\mathbf{X}_t^\theta) = \int \zeta_\psi(\tilde{\alpha}_t|\mathbf{X}_t^\theta)^\lambda q_{\bar{\theta}}(\tilde{\alpha}_t|\mathcal{Y}_{\text{tar}})^{1-\lambda}d\tilde{\alpha}_t \tag{57}$$

$$= \int \mathbf{C}_1 \exp\left[-\frac{\lambda}{2\sigma_p^2}\left\|\tilde{\alpha}_t - \alpha_t^\theta\right\|^2 - \frac{(1-\lambda)}{2\sigma_q^2}\left\|\tilde{\alpha}_t - \alpha^{\bar{\theta}}\right\|^2\right] \tag{58}$$

$$= \int \mathbf{C}_1 \exp\left[-\frac{1}{2}(\tilde{\alpha}_t-)^\top \mathbf{S}^{-1}(\tilde{\alpha}_t-) + \frac{1}{2}\left(^\top \mathbf{S}^{-1} - \frac{\lambda}{\sigma_p^2}\left\|\alpha_t^\theta\right\|^2 - \frac{1-\lambda}{\sigma_q^2}\left\|\alpha_t^{\bar{\theta}}\right\|^2\right)\right] \tag{59}$$

$$= \mathbf{C}_3 \exp\left[\frac{1}{2}\left(^\top \mathbf{S}^{-1} - \frac{\lambda}{\sigma_p^2}\left\|\alpha_t^\theta\right\|^2 - \frac{1-\lambda}{\sigma_q^2}\left\|\alpha_t^{\bar{\theta}}\right\|^2\right)\right], \tag{60}$$

where $\mathbf{C}_1 = \frac{1}{(2\pi)^{d/2}(\sigma_1^2)^{\frac{\lambda d}{2}}(\sigma_3^2)^{\frac{(1-\lambda)d}{2}}}$, $\mathbf{C}_3 = \frac{1}{\left(\frac{\lambda}{\sigma_1^2}+\frac{1-\lambda}{\sigma_3^2}\right)^{d/2}(\sigma_1^2)^{\frac{\lambda d}{2}}(\sigma_3^2)^{\frac{(1-\lambda)d}{2}}}$,

$$= \mathbf{S}\left(\frac{\lambda}{\sigma_p^2}\mathbf{X}_t^\theta + \frac{1-\lambda}{\sigma_q^2}\mathbf{T}_{\bar{\theta}}(t,\mathcal{Y}_{\text{tar}})\right), \text{ and } \mathbf{S} = \left(\frac{\lambda}{\sigma_p^2} + \frac{1-\lambda}{\sigma_q^2}\right)^{-1}\mathbf{I}. \tag{61}$$

Consequently, we get

$$
\mathbf{Z}(\mathbf{X}_t^\theta) = \mathbf{C}_3 \exp\left[ \frac{1}{2}\left( \frac{\left(\frac{\lambda}{\sigma_p^2}\alpha_t^\theta + \frac{1-\lambda}{\sigma_q^2}\alpha_t^{\bar{\theta}}\right)^\top \left(\frac{\lambda}{\sigma_p^2}\alpha_t^\theta + \frac{1-\lambda}{\sigma_q^2}\alpha_t^{\bar{\theta}}\right)}{\left(\frac{\lambda}{\sigma_p^2} + \frac{1-\lambda}{\sigma_q^2}\right)} \right) - \frac{\lambda}{\sigma_p^2}\left\|\alpha_t\right\|^2 - \frac{1-\lambda}{\sigma_q^2}\left\|\alpha_t^{\bar{\theta}}\right\|^2 \right]
$$

$$
= \mathbf{C}_3 \exp\left[ -\frac{\frac{\lambda(1-\lambda)}{\sigma_1^2\sigma_2^3}}{2\left(\frac{\lambda}{\sigma_1^2} + \frac{1-\lambda}{\sigma_3^2}\right)}\left\|\alpha_t^\theta - \alpha_t^{\bar{\theta}})\right\|^2 \right]. \tag{62}
$$

It implies that $-\log\tilde{\alpha}(\mathbf{X}_t^\theta) \geq 0$. Hence we can derive the desired inequality in (9):

$$
-\log p(\mathcal{Y}_{\text{tar}}|\mathcal{Y}_{\text{ctx}}) \leq \mathbb{E}_{\mathbf{X}^\theta \sim (10)}\left[ \int_0^T \frac{1}{2}\left\|\alpha_t^\theta\right\|^2 dt - \sum_{t\in\mathcal{T}} \mathbb{E}_{p(\tilde{\alpha}_t|\mathbf{X}_t^\theta)}\left[\log\zeta_\psi(\mathbf{y}_t|\tilde{\alpha}_t) + (1-\lambda)\log q_{\bar{\theta}}(\tilde{\alpha}_t|\mathcal{Y}_{\text{tar}})\right] \right]
$$

$$
\tag{63}
$$

$$
= \mathbb{E}_{\mathbf{X}^\theta \sim (10)}\left[ \int_0^T \frac{1}{2}\left\|\alpha_t^\theta\right\|^2 dt - \sum_{t\in\mathcal{T}} \mathbb{E}_{\tilde{\alpha}_t \sim p(\tilde{\alpha}_t|\mathbf{X}_t^\theta)}\left[ \frac{1}{2\sigma_\zeta^2}\left\|\mathbf{y}_t - \mathbf{D}_\psi(\tilde{\alpha}_t)\right\|^2 + \frac{(1-\lambda)}{2\sigma_q^2}\left\|\tilde{\alpha}_t - \alpha_t^{\bar{\theta}}\right\|^2 \right] \right]
$$

$$
\tag{64}
$$

$$
= \mathcal{L}(\theta, \psi). \tag{65}
$$

For stable learning, we train our model with rescaled training objective:

$$
\hat{\mathcal{L}}(\theta, \psi) = \mathbb{E}_{\mathbf{X}^\theta \sim (10)}\left[ \int_0^T \sigma_q^2\left\|\alpha_t^\theta\right\|^2 dt - \sum_{t\in\mathcal{T}_{\text{obs}}} \mathbb{E}_{\tilde{\alpha}_t \sim p(\tilde{\alpha}_t|\mathbf{X}_t^\theta)}\left[ \underbrace{\left\|\mathbf{y}_t - \mathbf{D}_\psi(\tilde{\alpha}_t)\right\|^2}_{\text{reconstruction}} + \tau\underbrace{\left\|\tilde{\alpha}_t - \alpha_t^{\bar{\theta}}\right\|^2}_{\text{regularization}} \right] \right],
$$

$$
\tag{66}
$$

Here, $\tau = \frac{(1-\lambda)\sigma_\zeta^2}{\sigma_q^2}$ determines the balance between reconstruction and regularization. See Section 4.3 for details on how controlling the regularization influences the performance of BDO.

## C  SIMULATION FREE INFERENCE

### C.1  NON-MARKOV CONTROL FORMULATION

In this section we clarify whether the ELBO derived in Proposition 3.1 is valid under various control formulations. We adapt the Donsker–Varadhan variational principle, which provides a variational formula for a wide range of Markov processes, including Itô diffusion processes (Hartmann et al., 2017; Tzen & Raginsky, 2019), into our formulation as follows.

**Lemma C.1** (Donsker–Varadhan variational principle). *Let $\mathbb{P}$ be a Markov path measure on $C([0,T],\mathbb{R}^d)$ and let $\mathcal{E}: C([0,T],\mathbb{R}^d) \to \mathbb{R}$ be a bounded and measurable functional. Then*

$$
-\log\mathbb{E}_{\mathbb{P}}\left[e^{-\mathcal{E}(\mathbf{X}_{[0,T]})}\right] = \min_{\mathbb{Q}\ll\mathbb{P}}\left[\mathbb{E}_{\mathbb{Q}}\left[\mathcal{E}(\tilde{\mathbf{X}}_{[0,T]})\right] + D_{\text{KL}}(\mathbb{Q}|\mathbb{P})\right], \tag{67}
$$

In particular, if we define the functional with the negative log-likelihood function with respect to $\mathcal{Y}$

$$
\mathcal{E}(\mathbf{X}_{[0,T]}) := -\log p(\mathcal{Y}|\mathbf{X}_{[0,T]}), \tag{68}
$$

and by plugging in the controlled path measure $\mathbb{P}^\alpha$ as a particular choice of the variational path measure $\mathbb{Q}$ on the right hand side of (69), then we recover the desired ELBO from Lemma C.1 with exactly the same computation as in (14–20):

$$
-\log\mathbb{E}_{\mathbb{P}}\left[p(\mathcal{Y}|\mathbf{X}_{[0,T]})\right] = \min_{\mathbb{Q}<<\mathbb{P}}\left[\mathbb{E}_{\mathbb{Q}}\left[-\log p(\mathcal{Y}|\tilde{\mathbf{X}}_{[0,T]})\right] + D_{\text{KL}}(\mathbb{Q}|\mathbb{P})\right] \tag{69}
$$

$$
\stackrel{(i)}{=} \min_{\mathbb{Q}<<\mathbb{P}}\left[\mathbb{E}_{\mathbb{Q}}\left[-\sum_{t\in\mathcal{T}}\log p(\mathcal{Y}|\tilde{\mathbf{X}}_{[0,T]}) + \int_0^T \frac{1}{2}\left\|\alpha_t^\theta\right\|^2 dt\right]\right] \tag{70}
$$

where $(i)$ follows from the Girsanov Theorem (Baldi, 2017). In other words, the ELBO in Proposition 3.1 can be viewed as the Donsker–Varadhan variational formula restricted to the family of controlled path measures $\mathbb{P}^\alpha$. Note that Lemma C.1 is stated for Markov processes $\mathbb{P}$ and $\mathbb{Q}$, hence the controlled path measure $\mathbb{P}^\alpha$ is a valid choice for $\mathbb{Q}$ in our formulation as long as the control $\alpha$ is an admissible and adapted process.

This should include (standard) Markov controls and under Markov control formulation, it is well known that the optimal control $\alpha^\star$ yields tight lower bound, via the relationship between Hamiltonian-Jacobi Bellman equation (Van Handel, 2007) and the ELBO through Hopf-cole transformation (Fleming & Soner, 2006). In this case, the value function induced by Markov optimal control solves the HJB equation and the corresponding optimal path measure attains the minimum of (69), so the ELBO bound becomes tight.

However, the ELBO formulation itself is not limited to Markov control cases. Because our specific non-Markov control formulation in (10) is still adapted with respect to the Brownian filtration, the ELBO derived in Proposition 3.1 remains valid under the non-Markov (open-loop) control parameterization, and the interpretation of the ELBO will be unchanged. A closely related discussion appears in (Park et al., 2024, Remark 3.9), where the Markov optimal control is approximated by a non-Markov control parameterized by a high-capacity neural network. They argue that if the network has sufficient capacity and is trained by gradient descent, the resulting local minimizer can approximate the optimal control under Markov formulation.

## C.2 PARALLEL SCAN ALGORITHM

The computation of the first two moments—the mean $\mu_{t \in \mathcal{T}}$ and covariance $\Sigma_{t \in \mathcal{T}}$—of the controlled distributions can be efficiently parallelized using the scan (all-prefix-sums) algorithm (Blelloch, 1990). Leveraging the associativity of the underlying operations, we reduce the computational complexity from $\mathcal{O}(k)$ to $\mathcal{O}(\log k)$ time with respect to the number of time steps $k$. We have established the linear recurrence in Theorem 3.2 for the mean and covariance at each time step $t_i$:

$$\mathbf{m}_{t_i} = \hat{\mathbf{A}}_i \mathbf{m}_{t_{i-1}} + \hat{\mathbf{B}}_i \alpha_{t_i}, \tag{71}$$

$$\boldsymbol{\Sigma}_{t_i} = \bar{\mathbf{A}}_i \boldsymbol{\Sigma}_{t_{i-1}} + \bar{\mathbf{B}}_i \mathbf{I}, \tag{72}$$

where we define $\Delta_i(t) = t - t_i$, $\hat{\mathbf{A}}_i = e^{-\Delta_{i-1}(t_i)\boldsymbol{\Lambda}_{t_i}}$, $\hat{\mathbf{B}}_i = -e^{-\Delta_{i-1}(t_i)\boldsymbol{\Lambda}_{t_i}} \boldsymbol{\Lambda}_{t_i}^{-1} \left( \mathbf{I} - e^{\Delta_{i-1}(t_i)\boldsymbol{\Lambda}_{t_i}} \right)$, $\bar{\mathbf{A}}_i = e^{-2\Delta_{i-1}(t_i)\boldsymbol{\Lambda}_{t_i}}$ and $\bar{\mathbf{B}}_i = -\frac{1}{2} e^{-2\Delta_{i-1}(t_i)\boldsymbol{\Lambda}_{t_i}} \boldsymbol{\Lambda}_{t_i}^{-1} \left( \mathbf{I} - e^{2\Delta_{i-1}(t_i)\boldsymbol{\Lambda}_{t_i}} \right)$. To apply the parallel scan algorithm to our recurrence, we define two separate sequences of tuples for the mean and covariance computations for all $i \in \{1, \cdots, k\}$:

$$\mathbf{M}_i = \left( \hat{\mathbf{A}}_i, \hat{\mathbf{B}}_i \alpha_{t_i} \right), \quad \mathbf{S}_i = \left( \bar{\mathbf{A}}_i, \bar{\mathbf{B}}_i \right) \tag{73}$$

Now, we define binary associative operators $\otimes$ for the sequences $\{\mathbf{M}_i\}$ and $\{\mathbf{S}_i\}$:

$$\mathbf{M}_i \otimes \mathbf{M}_j = \left( \hat{\mathbf{A}}_i \circ \hat{\mathbf{A}}_j, \hat{\mathbf{A}}_i \circ \hat{\mathbf{B}}_j \alpha_{t_j} + \hat{\mathbf{B}}_i \alpha_{t_i} \right), \tag{74}$$

$$\mathbf{S}_i \otimes \mathbf{S}_j = \left( \bar{\mathbf{A}}_i \circ \bar{\mathbf{A}}_j, \bar{\mathbf{A}}_i \circ \bar{\mathbf{B}}_j + \bar{\mathbf{B}}_i \right), \tag{75}$$

where $\circ$ denotes element-wise multiplication. We can verify that $\otimes$ is an associative operator since it satisfies:

$$(\mathbf{M}_s \otimes \mathbf{M}_t) \otimes \mathbf{M}_u = \left( \hat{\mathbf{A}}_t \circ \hat{\mathbf{A}}_s, \hat{\mathbf{A}}_t \circ \hat{\mathbf{B}}_s \alpha_{t_s} + \hat{\mathbf{B}}_t \alpha_{t_t} \right) \otimes \mathbf{M}_u \tag{76}$$

$$= \left( \hat{\mathbf{A}}_u \circ \left( \hat{\mathbf{A}}_t \circ \hat{\mathbf{A}}_s \right), \hat{\mathbf{A}}_u \circ \left( \hat{\mathbf{A}}_t \circ \hat{\mathbf{B}}_s \alpha_{t_s} + \hat{\mathbf{B}}_t \alpha_{t_t} \right) + \hat{\mathbf{B}}_u \alpha_{t_u} \right) \tag{77}$$

$$= \left( \left( \hat{\mathbf{A}}_u \circ \hat{\mathbf{A}}_t \right) \circ \hat{\mathbf{A}}_s, \left( \hat{\mathbf{A}}_u \circ \hat{\mathbf{A}}_t \right) \circ \hat{\mathbf{B}}_s \alpha_{t_s} + \hat{\mathbf{A}}_u \circ \hat{\mathbf{B}}_t \alpha_{t_t} + \hat{\mathbf{B}}_u \alpha_{t_u} \right) \tag{78}$$

$$= \mathbf{M}_s \otimes (\mathbf{M}_t \otimes \mathbf{M}_u). \tag{79}$$

Thus, we get $(\mathbf{M}_s \otimes \mathbf{M}_t) \otimes \mathbf{M}_u = \mathbf{M}_s \otimes (\mathbf{M}_t \otimes \mathbf{M}_u)$, confirming associativity for $\mathbf{M}_i$. Similarly,

$$(\mathbf{S}_s \otimes \mathbf{S}_t) \otimes \mathbf{S}_u = \left( \bar{\mathbf{A}}_t \circ \bar{\mathbf{A}}_s, \bar{\mathbf{A}}_t \circ \bar{\mathbf{B}}_s \mathbf{I} + \bar{\mathbf{B}}_t \mathbf{I} \right) \otimes \mathbf{M}_u \tag{80}$$

$$= \left( \bar{\mathbf{A}}_u \circ \left( \bar{\mathbf{A}}_t \circ \bar{\mathbf{A}}_s \right), \bar{\mathbf{A}}_u \circ \left( \bar{\mathbf{A}}_t \circ \bar{\mathbf{B}}_s \alpha_{t_s} + \bar{\mathbf{B}}_t \mathbf{I} \right) + \bar{\mathbf{B}}_u \mathbf{I} \right) \tag{81}$$

$$= \left( \left( \bar{\mathbf{A}}_u \circ \bar{\mathbf{A}}_t \right) \circ \bar{\mathbf{A}}_s, \left( \bar{\mathbf{A}}_u \circ \bar{\mathbf{A}}_t \right) \circ \bar{\mathbf{B}}_s \mathbf{I} + \bar{\mathbf{A}}_u \circ \bar{\mathbf{B}}_t \mathbf{I} + \bar{\mathbf{B}}_u \mathbf{I} \right) \tag{82}$$

$$= \mathbf{S}_s \otimes (\mathbf{S}_t \otimes \mathbf{S}_u). \tag{83}$$

**Algorithm 1** Parallel Scan for Mean and Covariance

1: **Input.** Given time stamps $\mathcal{T} = \{t_1, t_2, \ldots, t_K\}$, initial mean $\mu_{t_0}$ and covariance $\Sigma_{t_0}$, control policies $\{\alpha_{t_1}, \alpha_{t_2}, \ldots, \alpha_{t_K}\}$, matrices $\{\mathbf{\Lambda}_{t_1}, \mathbf{\Lambda}_{t_2}, \ldots, \mathbf{\Lambda}_{t_K}\}$.
2: **Initialize** sequences $\{\mathbf{M}_i\}_{i=1}^K$ and $\{\mathbf{S}_i\}_{i=1}^K$:
3: **for** $i = 1$ to $K$ **do in parallel**
4:    Compute $\Delta_i(t_i) = t_i - t_{i-1}$.
5:    Compute $\hat{\mathbf{A}}_i = e^{-\Delta_i(t_i)\mathbf{\Lambda}_{t_i}}$.
6:    Compute $\hat{\mathbf{B}}_i = -e^{-\Delta_i(t_i)\mathbf{\Lambda}_{t_i}}\mathbf{\Lambda}_{t_i}^{-1}\left(\mathbf{I} - e^{\Delta_i(t_i)\mathbf{\Lambda}_{t_i}}\right)$.
7:    Compute $\bar{\mathbf{A}}_i = e^{-2\Delta_i(t_i)\mathbf{\Lambda}_{t_i}}$.
8:    Compute $\bar{\mathbf{B}}_i = -\frac{1}{2}e^{-2\Delta_i(t_i)\mathbf{\Lambda}_{t_i}}\mathbf{\Lambda}_{t_i}^{-1}\left(\mathbf{I} - e^{2\Delta_i(t_i)\mathbf{\Lambda}_{t_i}}\right)$.
9:    Set $\mathbf{M}_i = \left(\hat{\mathbf{A}}_i, \hat{\mathbf{B}}_i\alpha_{t_i}\right)$.
10:   Set $\mathbf{S}_i = \left(\bar{\mathbf{A}}_i, \bar{\mathbf{B}}_i\right)$.
11: **end for**
12: Parallel Scan $\{\mathbf{M}'_i\}_{i=1}^K$ = ParallelScan($\{\mathbf{M}_i\}_{i=1}^K, \otimes$)
13: Parallel Scan $\{\mathbf{S}'_i\}_{i=1}^K$ = ParallelScan($\{\mathbf{S}_i\}_{i=1}^K, \otimes$)
14: **for** $i = 1$ to $K$ **do in parallel**
15:   $\mu_{t_i} = \mathbf{M}'^{(1)}_i\mu_{t_0} + \mathbf{M}'^{(2)}_i$
16:   $\Sigma_{t_i} = \mathbf{S}'^{(1)}_i\Sigma_{t_0} + \mathbf{S}'^{(2)}_i$
17: **end for**
18: **Return** $\mu_{t\in\mathcal{T}}, \Sigma_{t\in\mathcal{T}}$

**Algorithm 2** ParallelScan

1: **Input.** Sequence of tuples $\{\mathbf{T}_1, \mathbf{T}_2, \ldots, \mathbf{T}_K\}$, associative operator $\otimes$.
2: **Stage 1: Up-Sweep (Reduce).**
3: **for** $d = 0$ to $\lceil\log_2 K\rceil - 1$ **do**
4:   **for** each subtree of height $d$ in parallel **do**
5:     Let $i = 2^{d+1}k + 2^{d+1} - 1$ for $k = 0, 1, \ldots$
6:     **if** $i < K$ **then**
7:       $\mathbf{T}_i = \mathbf{T}_{i-2^d} \otimes \mathbf{T}_i$
8:     **end if**
9:   **end for**
10: **end for**
11: **Stage 2: Down-Sweep.**
12: $\mathbf{T}_K = \mathbf{I}$, where $\mathbf{I}$ is the identity element for $\otimes$.
13: **for** $d = \lceil\log_2 K\rceil - 1$ downto 0 **do**
14:   **for** each subtree of height $d$ in parallel **do**
15:     Let $i = 2^{d+1}k + 2^{d+1} - 1$ for $k = 0, 1, \ldots$
16:     **if** $i < K$ **then**
17:       $\mathbf{T}_{i-2^d} = \mathbf{T}_{i-2^d} \otimes \mathbf{T}_i$
18:       $\mathbf{T}_i = \mathbf{T}_{i-2^d}$
19:     **end if**
20:   **end for**
21: **end for**
22: **Return** Scanned sequence $\{\mathbf{T}'_1, \mathbf{T}'_2, \ldots, \mathbf{T}'_K\}$ where $\mathbf{T}'_i = \mathbf{T}_1 \otimes \mathbf{T}_2 \otimes \cdots \otimes \mathbf{T}_i$.

Hence, $(\mathbf{S}_s \otimes \mathbf{S}_t) \otimes \mathbf{S}_u = \mathbf{S}_s \otimes (\mathbf{S}_t \otimes \mathbf{S}_u)$, confirming associativity for $\mathbf{S}_i$. Now, we can apply the parallel scan described in Algorithm 1 for both $\mu_{t\in\mathcal{T}}$ and covariance $\Sigma_{t\in\mathcal{T}}$ based on the recurrence in (24, 28) and the defined associative operators $\otimes$. Employing the parallel scan algorithm offers significant computational benefits, especially for large-scale problems with numerous time steps $k$. The logarithmic time complexity ensures scalability, making it feasible to perform real-time computations or handle high-dimensional data efficiently.

# D   EXPERIMENTAL DETAILS

## D.1   DATA PREPROCESSING

**Preprocessing Pipeline.** The preprocessing pipeline involved several standard steps, including skull-stripping, slice-timing correction, motion correction, non-linear registration, and intensity normalization. All data were aligned to the Montreal Neurological Institute (MNI) standard space for consistency. A whole-brain mask was applied to exclude non-brain tissues, such as the skull, from further analysis. The fMRI data were parcellated into 450 regions of interest (ROIs), comprising 400 cortical parcels based on the Schaefer-400 atlas (Schaefer et al., 2017) and 50 subcortical parcels defined by Tian's Scale III atlas (Tian et al., 2020). The mean fMRI time-series for each ROI was extracted across all time points.

**Data Normalization.** To ensure comparability across participants and reduce inter-subject variability, we applied a two-step normalization process to the fMRI data. First, participant-wise zero-mean centering was performed by subtracting the mean signal from each ROI within each subject. Second, a robust scaling procedure was applied, where the median signal was subtracted, and the resulting values were divided by the interquartile range (IQR), computed across all participants for each ROI. After normalization, each fMRI sample was represented as a matrix of size $T \times N$, where $T$ corresponds to the number of timesteps and $N$ corresponds to the number of ROIs ($N = 450$).

**UK Biobank (UKB)** The UKB is a population-based prospective study comprising 500,000 participants in the United Kingdom, designed to investigate the genetic and environmental determinants of

disease (Sudlow et al., 2015). This study utilized 41,072 rs-fMRI scans from the publicly available, preprocessed UKB dataset (Alfaro-Almagro et al., 2018). The preprocessing pipeline included non-linear registration to MNI space using FSL's `applywarp` function, thereby ensuring standardized spatial alignment across participants (Jenkinson et al., 2012).

**Human Connectome Project in Aging (HCP-A)**   The HCP-A is a large-scale neuroimaging initiative focused on characterizing structural and functional connectivity changes associated with aging across a wide age range (Bookheimer et al., 2019). This study accessed 724 rs-fMRI samples from healthy individuals between 36 and 89 years of age. Preprocessed rs-fMRI volumes provided by the HCP-A dataset were utilized for subsequent analyses.

**Autism Brain Imaging Data Exchange (ABIDE)**   The ABIDE consortium aims to elucidate the neural mechanisms underlying autism spectrum disorder (Di Martino et al., 2014). In the present work, 1,102 rs-fMRI samples were obtained from the Neuro Bureau Preprocessing Initiative (Craddock et al., 2013a), which employs the Configurable Pipeline for the Analysis of Connectomes (C-PAC) (Craddock et al., 2013b). The preprocessing steps included slice-timing correction, motion realignment, intensity normalization (with a 4D global mean set to 1000), and nuisance signal removal. Nuisance regression involved a 24-parameter motion model, component-based noise correction (CompCor) (Behzadi et al., 2007) with five principal components derived from white matter and cerebrospinal fluid signals, and linear/quadratic trend removal. Functional-to-anatomical registration was performed via a boundary-based rigid-body approach, while anatomical-to-standard registration utilized ANTs. Band-pass filtering and global signal regression were not applied.

**Attention Deficit Hyperactivity Disorder 200 (ADHD200)**   The ADHD200 dataset comprises 776 rs-fMRI and anatomical scans collected from individuals aged 7 to 21, including 491 typically developing individuals and 285 participants diagnosed with ADHD (Brown et al., 2012). A total of 669 rs-fMRI datasets were selected for this study, specifically the preprocessed versions provided by the Neuro Bureau Preprocessing Initiative (Athena Pipeline) (Bellec et al., 2017).

**Human Connectome Project for Early Psychosis (HCP-EP)**   The HCP-EP is a neuroimaging initiative focused on understanding early psychosis, defined as the first five years following symptom onset, in individuals aged 16–35. The cohort includes participants with affective psychosis, non-affective psychosis, and healthy controls (Jacobs et al., 2024; Prunier & Shenton Martha; Breier, 2021). For this study, 176 rs-fMRI scans were analyzed. Preprocessing was conducted using fMRIPrep (Esteban et al., 2019), followed by denoising with Nilearn (Nilearn contributors, 2025). The denoising process employed a 24-parameter motion model (including translations, rotations, their derivatives, and quadratic terms) and CompCor-derived components extracted from white matter and cerebrospinal fluid masks. Additionally, all confound variables were demeaned to ensure consistency.

## D.2   PRE-TRAINING STAGE

**Pre-training Data.**   For self-supervised pre-training, we utilized the large-scale UKB dataset, which comprises resting-state fMRI recordings and medical records from 41,072 participants (Alfaro-Almagro et al., 2018). We utilized 80% of the dataset for pre-training, while the remaining 20% held-out data was reserved for downstream evaluation. We used a fixed random seed (42) to ensure reproducibility when partitioning the UKB dataset into pre-training and held-out subsets. All experiments, including the reproduction of foundation model baselines, were conducted using the same dataset split to maintain consistency.

**Irregular Multivariate Time-Series Sampling.**   We introduce irregularity in the time-series data by subsampling both the observation timestamps $\mathcal{T}_{obs}$ and the corresponding fMRI signals $\mathcal{Y}_{obs}$. Unlike conventional approaches that assume uniformly spaced time points (Caro et al., 2024; Dong et al., 2024), we select a uniformly sampled subset of timestamps from the full sequence, ensuring that only a fraction of the fMRI signal is observed. Specifically, from each full-length fMRI recording, we randomly sample 160 timesteps ($T = 160$), introducing variability in temporal resolution across different samples. This choice reflects the fundamental nature of brain dynamics, which evolve continuously rather than discretely, and encourages the model to infer missing states from incomplete sequences.

**Temporal Masking.** To encourage robust representation learning and improve generalization, we employ *temporal masking*, where a subset of the 160 sampled time points is randomly masked during training. We apply a masking ratio of $\gamma = 0.75$, meaning that 75% of the sampled timesteps are hidden while the model is trained to reconstruct them. In Figure 8, we vary $\gamma$ across $[0.4, 0.5, 0.6, 0.7, 0.75, 0.8, 0.9]$ to examine the effect of masking ratio in learning robust representations. Actual reconstruction results are provided in the internal and external datasets as visualized in Figures 15 and 16.

**Pre-training Algorithm.** The pre-training of BDO follows the procedure outlined in Algorithm 3. Given an observed fMRI time-series $\mathcal{Y}_{\text{obs}}$, we employ a masked reconstruction strategy, where a random proportion $\gamma$ of the temporal signals is masked to encourage the model to learn meaningful representations. The pre-training objective leverages amortized inference to approximate latent dynamics. At each iteration, a subset of observed time-series $\mathcal{Y}_{\text{ctx}}$ is used as context, while the masked portion $\mathcal{Y}_{\text{tar}}$ serves as the target for reconstruction. The encoder network $\mathbf{T}_\theta$ maps the context data to a sequence of latent states $\mathbf{z}_{t \in \mathcal{T}_{\text{ctx}}}$, which are then used to estimate drift terms and control policies, forming the basis for latent trajectory prediction. The decoder network $\mathbf{D}_\psi$ reconstructs the missing target states, optimizing a training objective $\mathcal{L}(\theta, \psi)$ that aligns the predicted and true trajectories.

**Pre-training Details.** We trained BDO using a batch size of 128 and a total of 200 pre-training epochs. The learning rate was scheduled using a cosine decay scheduler (Loshchilov & Hutter, 2016) with a 10-epoch warm-up phase. During warm-up, the initial learning rate was set to 0.0001, which increased to a peak learning rate of 0.001 before gradually decaying to a minimum learning rate of 0.0001. For optimization, we employed the Adam optimizer (Kingma & Ba, 2015). Across all BDO configurations, we used a fixed number of basis $l = 100$ and consistently multiplied the observation times by a time-scale of 0.1 for all datasets. To update $\bar{\theta}$, Exponential Moving Average (EMA) momentum is used and linearly increased from 0.996 to 1.0. It is worth noting that our models required minimal hyperparameter tuning, which demonstrates that the proposed approximation scheme operates stably and robustly.

## D.3 MODEL ARCHITECTURE

To maintain the structural advantages of our formulation, we designed our encoder network architecture in a straightforward manner. In this regard, the networks used for pre-training BDO are listed below, where N=450 is the number of ROIs and d is the dimension of latent space $\mathbb{R}^d$ as described in Table 5 for each model.

- **Encoder network** $q_\theta$:
  ```
  Input(N) → Linear(d) → ReLU() → LayerNorm(d) → Linear(d)
  → ReLU() → LayerNorm(d) → 12 × [LayerNorm(d) → Attn(d) →
  FFN(d)]
  ```

- **FFN**:
  ```
  Input(d) → LayerNorm(d) → Linear(4 × d) → GeLU() → Linear(d)
  → Residual(Input(d))
  ```

- **Attn**:
  ```
  Input(Q, K, V) → Normalize(Q) → Linear(Q) → Linear(K) →
  Linear(V) → Attention(Q, K) → Softmax(d) → Dropout() →
  Matmul(V) → LayerNorm(d) → Linear(d) → Residual(Q)
  ```

- **Decoder network** $\mathbf{D}_\psi$:
  ```
  Input(d) → Linear(N) → ReLU() → Dropout() → Linear(d)
  ```

**Locally Linear Approximation.** Recall that the locally linear SDE in (10) uses a time dependent drift matrix of the form $\mathbf{D}_{t_i} = \mathbf{V}\mathbf{\Lambda}_{t_i}\mathbf{V}^\top$, where $\mathbf{V} \in \mathbb{R}^{d \times d}$ is a eigen-basis and $\mathbf{\Lambda}_{t_i} \in \mathbb{R}^{d \times d}$ is a diagonal eigen-value matrix whose entries vary with the time index $t_i$. Here, we set the matrix $\mathbf{V}$ as a single trainable matrix shared across all time-interval $t \in [0, T]$ enforced to remain orthogonal following (Lezcano Casado, 2019). The time-dependent eigen-value $\mathbf{\Lambda}_{t_i}$ is defined as affine formulation with following components $\mathbf{\Lambda}_{t_i} = \sum_{l=1}^{L} w_{t_i}^{(l)} \mathbf{\Lambda}^{(l)}$, where $\{\mathbf{\Lambda}^{(l)}\}_{l=1}^{L}$ is a set of trainable diagonal base matrices. The weight coefficients $\mathbf{w}_{t_i} = \{w_{t_i}^{(1)}, \cdots, w_{t_i}^{(L)}\}$ are produced by neural network $W_\theta$, $\mathbf{w}_{t_i} = \text{softmax}(W_\theta(\alpha_{t_i}))$.

Table 5: Pre-training hyper-parameters

| BDO Variants | Train EP | Warm-up EP | LR | Initial LR | Minimum LR | Batch Size | $\mathbb{R}^d$ | # of base matrices (L) | EMA Momentum |
|---|---|---|---|---|---|---|---|---|---|
| BDO (5M) | 200 | 10 | 0.001 | 0.0001 | 0.0001 | 128 | 192 | 100 | [0.996, 1] |
| BDO (21M) | 200 | 10 | 0.001 | 0.0001 | 0.0001 | 128 | 384 | 100 | [0.996, 1] |
| BDO (86M) | 200 | 10 | 0.001 | 0.0001 | 0.0001 | 128 | 768 | 100 | [0.996, 1] |

---

**Algorithm 3** Pre-training BDO

1: **Input.** Time-series $\mathcal{Y}_{\text{obs}} = \mathbf{y}_{t \in \mathcal{T}_{\text{obs}}}$, masking ratio $\gamma$, encoder network $\mathbf{T}_\theta$, decoder network $\mathbf{D}_\psi$
2: **for** $m = 1, \cdots, M$ **do**
3:    Get $\mathcal{Y}_{\text{ctx}}, \mathcal{Y}_{\text{tar}}$ by masking $\gamma\%$ of temporal signals.
4:    Sample $\mathbf{z}_{t \in \mathcal{T}_{\text{ctx}}} \sim \prod_{t \in \mathcal{T}_{\text{ctx}}} q_\theta(\mathbf{z}_t | \mathcal{Y}_{\text{ctx}})$.
5:    Compute $\{\mathbf{D}_t, u_t, \alpha_t^\theta\}_{t \in \mathcal{T}_{\text{ctx}}}$
6:    Estimate $\{\mu_t, \Sigma_t\}_{t \in \mathcal{T}_{\text{tar}}}$ with parallel scan algorithm.
7:    Sample $\mathbf{X}_{t \in \mathcal{T}_{\text{tar}}}^\theta \overset{i.i.d}{\sim} \otimes_{t \in \mathcal{T}_{\text{tar}}} \mathcal{N}(\mu_t, \Sigma_t)$.
8:    Sample $\hat{\mathbf{z}}_{t \in \mathcal{T}_{\text{tar}}} \sim \prod_{t \in \mathcal{T}_{\text{tar}}} p(\hat{\mathbf{z}}_t | \mathbf{X}_t^\theta)$.
9:    Compute $\hat{\mathcal{L}}(\theta, \psi)$ using (66).
10:    Update $(\theta, \psi)$ with $\nabla_{\theta, \psi} \hat{\mathcal{L}}(\theta, \psi)$.
11:    Apply $\bar{\theta} \leftarrow \text{EMA}(\theta)$.
12: **end for**

**Algorithm 4** Fine tuning BDO for downstream tasks

1: **Input.** Time-series and label $(\mathcal{Y}_{\text{obs}}, \mathcal{O}_{\text{obs}})$, pre-trained encoder network $\mathbf{T}_{\theta^\star}$.
2: Sample $\mathbf{z}_{t \in \mathcal{T}_{\text{obs}}} \sim \prod_{t \in \mathcal{T}_{\text{obs}}} q_{\theta^\star}(\mathbf{z}_t | \mathcal{Y}_{\text{obs}})$.
3: Compute optimal control policy $\alpha_{t \in \mathcal{T}_{\text{obs}}} = \mathbf{B}_{\theta^\star} \mathbf{z}_{t \in \mathcal{T}_{\text{obs}}}$.
4: Compute the universal feature $\mathbb{A} = \frac{1}{|\mathcal{T}_{\text{obs}}|} \sum_{t \in \mathcal{T}_{\text{obs}}} \alpha_t$.
5: Predict $\hat{\mathcal{O}}_{\text{obs}} = h_\omega(\mathbb{A})$.
6: **if** *Linear probing* **then**
7:    Freeze the pre-trained encoder network $\mathbf{T}_{\theta^\star}$.
8:    Compute $\mathcal{L}(\theta^\star, \omega) = \mathcal{L}_{\text{task}}(\mathcal{O}_{\text{obs}}, \hat{\mathcal{O}}_{\text{obs}})$ using (84).
9:    Update $\omega$ with $\nabla_\omega \mathcal{L}(\theta^\star, \omega)$.
10: **else if** *Fine tuning* **then**
11:    Unfreeze the pre-trained encoder network $\mathbf{T}_{\theta^\star}$.
12:    Compute $\mathcal{L}(\theta^\star, \omega) = \mathcal{L}_{\text{task}}(\mathcal{O}_{\text{obs}}, \hat{\mathcal{O}}_{\text{obs}})$ using (84).
13:    Update $(\theta^\star, \omega)$ with $\nabla_{\theta^\star, \omega} \mathcal{L}(\theta^\star, \omega)$.
14: **end if**

Table 6: Dataset Subject Demographics

| Category | UKB | HCP-A | ABIDE | ADHD200 | HCP-EP |
|---|---|---|---|---|---|
| # of subjects | 41,072 | 724 | 1,102 | 669 | 176 |
| Age, mean (SD) | 54.98 (7.53) | 60.35 (15.74) | 17.05 (8.04) | 11.61 (2.97) | 23.39 (3.95) |
| Female, % (n) | 52.30 (21,480) | 56.08 (406) | 14.79 (163) | 36.17 (242) | 38.07 (67) |
| Patient, % (n) | - | - | 48.19 (531) | 58.15 (389) | 68.18 (120) |
| Target Population | Healthy Population | Healthy Population | ASD Healthy Population | ADHD Healthy Population | Psychotic Disorder Healthy Population |

## D.4 DOWNSTREAM EVALUATION STAGE

To assess the generalization and transferability of BDO, we conducted experiments across multiple datasets and tasks, encompassing both demographic and psychiatric prediction. Datasets used in this evaluation have distinct temporal resolutions and varying numbers of timesteps, reflecting the irregularity of real-world fMRI data acquisition. Additional details are described in Table 6. Note that in the downstream evaluation, irregular sampling and temporal masking were disabled. The full sequence of fMRI signals, timestamps, and corresponding labels was used, denoted as $(\mathcal{Y}_{\text{obs}}, \mathcal{T}_{\text{obs}}, \mathcal{O}_{\text{obs}})$.

**Internal Evaluation.** For *internal evaluation*, we utilized a 20% held-out subset of the UKB dataset, which was excluded from pre-training. This evaluation focused on age regression and gender classification, leveraging both LP and FT to analyze how well the model retains and transfers knowledge acquired during pre-training.

**External Evaluation.** For *external evaluation*, we examined the ability of BDO to generalize to unseen datasets. Demographic and trait prediction was performed on the HCP-A dataset, where LP and FT were employed to assess model performance on age, gender, neuroticism, and flanker scores. Beyond demographic characteristics, we evaluated psychiatric diagnosis classification using 3 clinical fMRI datasets, including ABIDE, ADHD200, and HCP-EP. These evaluations relied on LP, as it provides a controlled assessment of the learned representations and their applicability to clinical classification tasks.

**Random Splits.** All the datasets are partitioned into training, validation, and test sets using a 6:2:2 ratio to ensure fair and reproducible evaluation. To maintain consistency, we perform partitioning with 3 consecutive random seeds, 0, 1, and 2.

- For classification tasks, such as gender classification, stratified sampling is applied to preserve class distributions across the training, validation, and test sets.

- For regression tasks, such as age regression, binning-based stratified sampling is employed. In this approach, the continuous target variable is first discretized into bins before applying stratified sampling, ensuring a balanced distribution of the target variable and mitigating potential biases from uneven data partitioning. Additionally, to improve numerical stability and facilitate optimization, the target variable is normalized using Z-score normalization, where the mean is subtracted, and the result is divided by the standard deviation.

- The distributions of the three random splits for age regression tasks with the UKB and HCP-A datasets, and six classification tasks with UKB gender, HCP-A gender, ABIDE diagnosis, ADHD200 diagnosis, and HCP-EP diagnosis are described in Figures $12-14$.

**Extracting the Universal Feature** $\mathbb{A}$**.** To extract the *universal feature* $\mathbb{A}$, we define $f$ as *mean-pooling* over the sequence of control signals $\alpha_{t \in \mathcal{T}}$, given by $\mathbb{A} := f(\alpha_{t \in \mathcal{T}}) = \frac{1}{|\mathcal{T}|} \sum_{t \in \mathcal{T}} \alpha_t$. This formulation ensures that $\mathbb{A}$ serves as a compact and transferable representation of the underlying spatio-temporal dynamics captured by the optimal control signals. To enhance biological interpretability, mean-pooling is chosen as it provides a *global summary* of the temporal evolution of the control sequence while suppressing high-frequency fluctuations that may arise due to local variations in $\alpha_t$. Although we believe that mean-pooling provides a robust and scalable approach for summarizing temporal dynamics, we acknowledge that more sophisticated aggregation methods, such as weighted pooling or recurrent architectures, could further enhance downstream performance. These approaches may offer additional advantages for analyzing temporal dynamics, such as facilitating interpretability through attention weight analysis or capturing long-range dependencies. We leave the exploration of these advanced aggregation strategies for future work.

**Downstream Evaluation Algorithm.** To evaluate the effectiveness of BDO on downstream tasks, we follow the procedure outlined in Algorithm 4. Given an observed fMRI time-series $\mathcal{Y}_{\text{obs}}$ and its corresponding labels $\mathcal{O}_{\text{obs}}$, we extract the universal feature representation $\mathbb{A}$ using the pre-trained encoder $\mathbf{T}_{\theta^\star}$. This representation is subsequently used for classification or regression tasks through either LP or FT.

- In the LP setting, we freeze the pre-trained encoder $\mathbf{T}_{\theta^\star}$ and train only the task-specific head $h_\omega : \mathbb{R}^d \to \mathbb{R}^N$ (single linear layer). The objective function $\mathcal{L}(\theta^\star, \omega)$ measures the discrepancy between the predicted $\hat{\mathcal{O}}_{\text{obs}}$ and ground-truth $\mathcal{O}_{\text{obs}}$, and is optimized with respect to $\omega$.

- In the FT setting, the entire model, including $\mathbf{T}_{\theta^\star}$, is optimized. Both the encoder and task-specific head $h_\omega$ are jointly updated to refine the feature extraction process for the target task.

**Training Objective for Downstream tasks.** The loss function for downstream tasks is defined based on the nature of the prediction problem: classification tasks use Binary Cross-Entropy (BCE) loss, while regression tasks employ Mean Squared Error (MSE) loss.

**Model Selection.** To determine the optimal model for each downstream task, we performed a grid search over key hyperparameters such as learning rate and batch size. For each task, we evaluated multiple configurations using the validation set and selected the model that achieved the best performance based on the predefined metric. The set of hyperparameters is provided in Table 7.

$$
\mathcal{L}_{\text{task}}(\mathcal{O}_{\text{obs}}, \hat{\mathcal{O}}_{\text{obs}}) = \begin{cases} -\frac{1}{N} \sum_{i=1}^{N} \left[ \mathcal{O}_{\text{obs},i} \log \hat{\mathcal{O}}_{\text{obs},i} + (1 - \mathcal{O}_{\text{obs},i}) \log(1 - \hat{\mathcal{O}}_{\text{obs},i}) \right], & \text{if classification} \\ \frac{1}{N} \sum_{i=1}^{N} (\mathcal{O}_{\text{obs},i} - \hat{\mathcal{O}}_{\text{obs},i})^2. & \text{if regression} \end{cases}
$$

$$(84)$$

Table 7: Search space of end-to-end fine-tuning (FT) and linear probe (LP).

| Configurations | FT | LP |
|---|---|---|
| Optimizer | AdamW (Loshchilov, 2017) | Adam (Kingma & Ba, 2015) |
| Training epochs | 50 | 50 |
| Batch size | [16, 32] | [16, 32, 64] |
| LR scheduler | cosine decay | cosine decay |
| LR | [0.001] | [0.01, 0.005] |
| Minimum LR | [0, 0.0001, 0.001] | [0.001, 0.005] |
| Weight decay | [0, 0.01] | [0] |
| Layer-wise LR decay | [0.85, 0.90, 0.95] | N.A. |

## D.5 ADDITIONAL BASELINES

### D.5.1 COMPARISON WITH BRAIN-JEPA

In this section, we present a detailed comparison with Brain-JEPA (Dong et al., 2024), a prominent SSL baseline for fMRI analysis. To ensure the most rigorous comparison against the optimal version of the baseline, we utilized the official Brain-JEPA pre-trained weights (trained for 300 epochs) and compared Brain-JEPA against BDO (trained for 200 epochs) under both LP and FT protocols. To isolate the effect of preprocessing, we adopted Brain-JEPA's original preprocessing pipeline for both models. Note that BDO and Brain-JEPA share exactly the same Transformer backbone.

**Comparison on HCP-A.** This dataset aligns well with the pre-training conditions of Brain-JEPA. Specifically, the effective TR after downsampling in HCP-A ($0.72s \times 3 \approx 2.16s$) is very close to the pre-training TR derived from UKB ($0.735s \times 3 \approx 2.205s$), and the sequence length remains sufficient. Even in this favorable setting, BDO outperforms Brain-JEPA across the majority of tasks on HCP-A, with the exception of the Flanker regression task in the LP setting as shown in Table 8.

**Comparison on ABIDE, ADHD200, and HCP-EP.** We also extended the comparison to ABIDE, ADHD200, and HCP-EP as summarized in Table 9. However, unlike BDO, the official pre-trained Brain-JEPA model is constrained by fixed input dimensions of 450 ROIs (Schaefer-400 and Tian-Scale III) and a sequence length of 160 timesteps. Consequently, for datasets with sequences shorter than 160 timesteps, we explicitly applied data repetition to satisfy this input constraint.

ABIDE and ADHD200 are multi-site datasets characterized by heterogeneous TRs in addition to short sequence lengths. For a fixed-patch architecture like Brain-JEPA, this TR variability implies that input patches correspond to inconsistent physical durations across subjects, complicating the modeling of unified temporal dynamics. This structural mismatch, combined with the artifacts introduced by data repetition, likely limits performance.

This limitation is also evident in HCP-EP, where Brain-JEPA necessitates downsampling for TR alignment ($0.7s \times 3 \approx 2.1s$). Since this downsampling reduces the sequence length below the required threshold, we subsequently applied data repetition to meet the input constraints. In contrast, through continuous-time modeling, BDO natively adapts to subject-specific TRs and variable lengths, processing data directly without such artificial adjustments. This difference likely contributes to the observed performance gap, underscoring the benefits of continuous-time modeling.

Table 8: Performance comparison on HCP-A.

| | Methods | Age | | Gender | | Flanker | |
|---|---|---|---|---|---|---|---|
| | | MSE ↓ | ρ ↑ | ACC (%) ↑ | F1 (%) ↑ | MSE ↓ | ρ ↑ |
| LP | Brain-JEPA (86M) | 0.364 ±.023 | 0.804 ±.004 | 73.15 ±1.09 | 73.09 ±1.18 | **0.962** ±.131 | **0.360** ±.064 |
| | BDO (86M) | **0.298** ±.022 | **0.839** ±.010 | **74.48** ±1.82 | **74.52** ±3.81 | 0.966 ±.073 | 0.343 ±.059 |
| FT | Brain-JEPA (86M) | 0.275 ±.011 | 0.854 ±.008 | 82.42 ±1.21 | 82.38 ±1.20 | 0.891 ±.011 | 0.424 ±.017 |
| | BDO (86M) | **0.262** ±.009 | **0.862** ±.004 | **82.92** ±1.92 | **82.88** ±1.99 | **0.862** ±.043 | **0.452** ±.088 |

Table 9: Performance comparison on ABIDE, ADHD200, and HCP-EP.

| | Methods | ABIDE | | ADHD200 | | HCP-EP | |
|---|---|---|---|---|---|---|---|
| | | ACC (%) ↑ | F1 (%) ↑ | ACC (%) ↑ | F1 (%) ↑ | ACC (%) ↑ | F1 (%) ↑ |
| LP | Brain-JEPA (86M) | 59.92 ±3.74 | 58.85 ±3.72 | 56.22 ±1.06 | 56.34 ±1.11 | 72.38 ±3.56 | 71.35 ±3.89 |
| | BDO (86M) | **64.09 ±2.60** | **64.04 ±2.60** | **58.40 ±3.03** | **58.54 ±2.99** | **73.33 ±3.56** | **72.54 ±3.41** |
| FT | Brain-JEPA (86M) | 64.55 ±4.80 | 64.27 ±4.82 | 62.15 ±0.61 | 62.32 ±0.51 | 75.24 ±1.35 | 74.47 ±1.46 |
| | BDO (86M) | **67.90 ±2.68** | **67.88 ±2.67** | **63.15 ±2.81** | **63.32 ±2.72** | **79.05 ±2.69** | **78.65 ±2.63** |

### D.5.2 COMPARISON WITH SOLVER-BASED BASELINES

We further investigated the advantages of our simulation-free scheme compared to solver-based approaches. To this end, we conducted experiments comparing BDO against Latent ODE (Rubanova et al., 2019) and GRU-ODE-Bayes (De Brouwer et al., 2019). Since these frameworks were not originally designed for large-scale SSL scenarios, we established an empirical reference by training them in a supervised setting on the HCP-A dataset. Specifically, we employed a composite objective of reconstruction and task-specific losses, as described in their original papers.

Table 10: Comparison with solver-based continuous-time models on HCP-A.

| Methods | Age | | Gender | |
|---|---|---|---|---|
| | MSE ↓ | $\rho$ ↑ | ACC (%) ↑ | F1 (%) ↑ |
| Latent ODE | 0.556 ±.049 | 0.671 ±.042 | 64.22 ±3.88 | 63.10 ±3.32 |
| GRU-ODE-Bayes | 0.587 ±.046 | 0.652 ±.039 | 63.30 ±3.65 | 62.18 ±3.24 |
| BDO (86M) - LP | 0.404 ±.010 | 0.768 ±.008 | 72.00 ±2.95 | 71.30 ±2.19 |
| BDO (86M) - FT | **0.273** ±.010 | **0.851** ±.006 | **79.40** ±4.07 | **78.98** ±4.38 |

As presented in Table 10, BDO significantly outperforms both supervised solver-based baselines, even in the LP setting. This result highlights BDO's ability to leverage the benefits of large-scale pre-training to learn robust and generalizable representations. In contrast, while employing numerical solvers theoretically allows for modeling complex dynamics, their high computational cost acts as a severe bottleneck, preventing them from effectively scaling to massive datasets for pre-training.

By bypassing the bottlenecks of numerical integration, our simulation-free framework enables efficient large-scale SSL while achieving state-of-the-art performance. Thus, while the SOC formalism provides the theoretical foundation for our objective, the simulation-free scheme serves as the practical engine that enables scaling to the high dimensionality and complexity of neuroimaging data.

### D.5.3 COMPARISON WITH BRAINHARMONIX-F

We compare our framework with BrainHarmonix (Dong et al., 2025), a recently proposed multi-modal foundation model designed to unify brain structural morphology and functional dynamics. BrainHarmonix is originally designed to unify brain structure and function by pre-training two separate unimodal encoders (BrainHarmonix-S for T1-weighted structural MRI and BrainHarmonix-F for fMRI time-series) before fusing them through shared tokens. However, for a fair comparison, we specifically isolate the functional encoder, BrainHarmonix-F, and use it as a baseline.

BrainHarmonix-F was pre-trained on a massive combination of the UK Biobank (UKB) and Adolescent Brain Cognitive Development (ABCD) datasets, using the Schaefer-400 atlas. Thus, we also pre-trained our proposed model on the identical large-scale curation of datasets (UKB and ABCD) using the same parcellation scheme and preprocessing pipeline.

We evaluated the linear probing performance across multiple benchmark datasets. As shown in Table 11 and Table 12, compared to previous baselines such as Brain-JEPA, BrainHarmonix-F demonstrates overall performance improvements, which can likely be attributed to the inclusion of the large-scale ABCD dataset during pre-training and the use of data augmentation techniques. Notably,

the performance gain is particularly evident in disease diagnosis tasks, suggesting that the ability to handle heterogeneous TRs allows the model to be agnostic to temporal mismatches and arbitrary data repetition, thereby learning more robust clinical features. However, BDO's superior or comparable performance across most tasks further indicates that modeling the continuous stochasticity of brain dynamics offers a distinct advantage over discrete tokenization.

Table 11: Linear probing performance comparison on HCP-A.

| Methods | Age | | Gender | | Flanker | |
|---|---|---|---|---|---|---|
| | MSE ↓ | ρ ↑ | ACC (%) ↑ | F1 (%) ↑ | MSE ↓ | ρ ↑ |
| BrainHarmonix-F (86M) | 0.308 ±.019 | 0.837 ±.006 | 75.22 ±1.21 | 75.02 ±1.24 | 0.892 ±.093 | 0.424 ±.018 |
| BDO (86M) | **0.275 ±.021** | **0.850 ±.008** | **75.74 ±1.72** | **75.27 ±1.87** | **0.864 ±.081** | **0.447 ±.102** |

Table 12: Linear probing performance comparison on ABIDE, ADHD200, and HCP-EP.

| Methods | ABIDE | | ADHD200 | | HCP-EP | |
|---|---|---|---|---|---|---|
| | ACC (%) ↑ | F1 (%) ↑ | ACC (%) ↑ | F1 (%) ↑ | ACC (%) ↑ | F1 (%) ↑ |
| BrainHarmonix-F (86M) | 64.46 ±4.48 | 64.44 ±4.48 | **63.41 ±2.32** | **63.39 ±2.45** | 76.19 ±5.87 | 75.88 ±5.87 |
| BDO (86M) | **66.33 ±4.00** | **66.01 ±4.77** | 62.37 ±3.15 | 62.39 ±3.14 | **77.14 ±2.33** | **76.92 ±2.13** |

## D.6 GENERALIZATION FROM RESTING-STATE TO TASK-BASED fMRI

To conduct a challenging test of BDO's generalization capabilities, we evaluated whether its representations, learned exclusively from unconstrained resting-state fMRI, could be effectively transferred to structured task-based fMRI, where brain dynamics are driven by explicit external stimuli.

For this experiment, we used the BDO-86M model, which was pre-trained solely on resting-state UKB data. We then evaluated its performance on three distinct and cognitively demanding task paradigms from the HCP-A dataset. The evaluation was performed under the LP setting.

The results are summarized in Table 13. While there is a moderate performance decrease compared to the in-domain resting-state baseline (HCP-A-Rest), the model still achieves strong predictive performance across all three task paradigms (HCP-A-VisMotor/FaceName/CARIT).

This successful transfer demonstrates that BDO learns fundamental, task-relevant neural dynamics that are not limited to the resting state. This underscores BDO's broad applicability as a powerful feature extractor for diverse fMRI paradigms, even without any task-specific fine-tuning.

Table 13: Generalization from resting-state to task-based fMRI.

| Dataset | Age (MSE) ↓ | Age (Pearson) ↑ | Gender (Acc.) ↑ | Gender (F1) ↑ |
|---|---|---|---|---|
| HCP-A-VisMotor | 0.526 ±.018 | 0.691 ±.015 | 68.53 ±3.57 | 67.39 ±3.36 |
| HCP-A-FaceName | 0.459 ±.012 | 0.732 ±.009 | 66.20 ±3.44 | 65.29 ±3.72 |
| HCP-A-CARIT | 0.488 ±.025 | 0.713 ±.020 | 67.60 ±1.74 | 66.79 ±1.29 |
| HCP-A-Rest | 0.404 ±.010 | 0.768 ±.008 | 72.00 ±2.95 | 71.30 ±2.19 |

## D.7 SCALABILITY ANALYSIS

### D.7.1 MODEL AND DATA SCALABILITY

The numerical results used to generate the scalability analysis plot (Figure 7) are presented in Table 14. The table includes detailed linear probing performance for three BDO model variants (5M, 21M, and 86M), evaluated on HCP-A age regression and classification tasks across ABIDE, ADHD200, and HCP-EP datasets. Additionally, results from the data scalability experiment, conducted exclusively with the largest model (86M), are reported at varying proportions (25%, 50%, and 75%) of the total

dataset. The entry labeled BDO (86M) corresponds to the model trained with the full dataset (100%), serving as the reference for both model and data scalability experiments.

Table 14: LP performance used for scalability analysis in Figure 7.

| Variants | HCP-A
Age ($\rho \uparrow$) | ABIDE
ACC (%) $\uparrow$ | ADHD200
ACC (%) $\uparrow$ | HCP-EP
ACC (%) $\uparrow$ |
|---|---|---|---|---|
| BDO (5M) | 0.635 ±.031 | 62.42 ±2.68 | 59.65 ±2.30 | 73.33 ±7.50 |
| BDO (21M) | 0.729 ±.011 | 63.79 ±1.83 | 61.15 ±1.97 | 71.43 ±4.04 |
| BDO (25%) | 0.686 ±.010 | 61.06 ±1.05 | 57.39 ±3.90 | 72.38 ±5.95 |
| BDO (50%) | 0.702 ±.014 | 63.03 ±1.63 | 56.89 ±3.38 | 74.29 ±9.90 |
| BDO (75%) | 0.734 ±.011 | 65.45 ±2.70 | 58.15 ±1.78 | 74.29 ±7.56 |
| BDO (86M) | 0.768 ±.008 | 66.67 ±1.13 | 61.40 ±1.97 | 76.19 ±4.86 |

### D.7.2 ROI PARCELLATION SCALABILITY

To investigate the impact of spatial resolution, we conducted an ablation study varying the granularity of the ROI parcellation. We trained BDO (86M) variants using the Schaefer atlas with 200, 400, 600, 800, and 1000 parcels (Schaefer et al., 2017). All models in this experiment were pre-trained for 100 epochs and evaluated on the HCP-A dataset under the LP protocol.

Table 15: LP performance for ROI scalability analysis on HCP-A.

| Atlas | Age | | Gender | |
|---|---|---|---|---|
| | MSE $\downarrow$ | $\rho \uparrow$ | ACC (%) $\uparrow$ | F1 (%) $\uparrow$ |
| Schaefer-200 | 0.616 ±.061 | 0.641 ±.042 | 67.82 ±0.87 | 66.04 ±2.79 |
| Schaefer-400 | 0.563 ±.026 | 0.668 ±.014 | 68.06 ±1.50 | 67.02 ±1.31 |
| Schaefer-600 | 0.557 ±.038 | 0.678 ±.028 | 69.91 ±1.43 | 68.76 ±1.32 |
| Schaefer-800 | 0.571 ±.058 | 0.669 ±.039 | 73.38 ±0.87 | 72.82 ±1.20 |
| Schaefer-1000 | 0.537 ±.068 | 0.695 ±.050 | 74.31 ±3.00 | 73.75 ±2.84 |

As summarized in Table 15, we observed a general trend where increasing spatial resolution leads to improved downstream performance. This aligns with prior findings suggesting that finer parcellations capture richer representations of neural activity (Sacerdote & Giraudo, 2012). Specifically, the model trained with 1000 ROIs achieved the best performance across all metrics, confirming that our framework effectively scales to utilize more granular spatial information.

It is important to highlight the computational efficiency of the BDO framework regarding spatial scalability. Many existing transformer-based fMRI models typically tokenize ROIs into a sequence (Caro et al., 2024; Dong et al., 2024; Yang et al., 2024). In such architectures, increasing the number of ROIs ($N$) linearly extends the input sequence length, which in turn leads to a quadratic increase ($\mathcal{O}(N^2)$) in the computational and memory costs of the self-attention mechanism. This makes high-resolution modeling prohibitively expensive.

In contrast, BDO projects the $N$-dimensional ROI space at each time step into a fixed $d$-dimensional latent space via a linear projection before temporal processing. Therefore, increasing $N$ only affects the number of parameters in the initial projection and final decoder layers, while the sequence length ($T$) and the computational complexity of the Transformer encoder ($\mathcal{O}(T^2)$) remain unchanged. This structural design ensures that BDO scales efficiently with higher spatial resolutions, enabling feasible training even with fine-grained parcellations like 1000 ROIs. Furthermore, this architectural efficiency is a key factor contributing to the superior training speed of BDO compared to other SSL baselines, as detailed in the following section Section D.8.

Table 16: Comparison of pre-training efficiency and linear probing performance across SSL models.

| Model (Parameters) | Age (Pearson) ↑ | Gender (Acc.) ↑ | GPU Hours (x 4 GPUs) ↓ |
|---|---|---|---|
| MoCo (90M) | 0.591 | 64.12 | 174 hrs |
| BYOL (90M) | 0.619 | 64.81 | 165 hrs |
| BrainLM (85M) | 0.636 | 65.28 | 496 hrs |
| BrainMass (90M) | 0.630 | 66.20 | 244 hrs |
| BDO (86M) | **0.768** | **72.00** | **15 hrs** |

### D.8 COMPARISON OF SSL MODEL EFFICIENCY

This subsection presents the detailed experimental settings and the exact numerical results used to construct Figure 1, which shows that BDO surpasses other SSL models in both resource and parameter efficiency.

To evaluate the efficiency of SSL models, we measured the pre-training time using 4 NVIDIA RTX 3090 GPUs, calculated in GPU hours as the total CUDA time recorded with the `PyTorch` library and multiplied by the number of GPUs. Each model was trained for 200 epochs using the largest batch size that fully utilized available GPU memory.

Table 16 presents the linear probing performance of each pre-trained model on age and gender prediction tasks on HCP-A dataset, alongside their respective pre-training times. Our results demonstrate that BDO achieves superior efficiency in pre-training, requiring significantly fewer GPU hours compared to other SSL methods while maintaining competitive or superior performance. This efficiency highlights the scalability of BDO, making it a practical choice for large-scale applications.

### D.9 DETAILED MASK RATIO ANALYSIS

In our framework, the MAE objective plays a critical role by explicitly requiring the reconstruction of masked temporal segments. This encourages the model to capture detailed temporal dependencies and fine-grained dynamics inherent in fMRI signals. To directly validate the importance of the MAE objective, we extended our ablation study to include a zero mask ratio ($\gamma = 0$), which effectively removes the MAE reconstruction task. Starting from the no-masking condition, the HCP-A age regression performance in the LP setting for BDO-86M results are summarized below.

Table 17: Extended ablation study on the mask ratio $\gamma$.

| Mask Ratio ($\gamma$) | Age (MSE) ↓ | Age (Pearson) ↑ |
|---|---|---|
| 0.0 (No Masking) | 0.793 ± 0.014 | 0.445 ± 0.020 |
| 0.2 | 0.487 ± 0.036 | 0.711 ± 0.027 |
| 0.4 | 0.513 ± 0.016 | 0.695 ± 0.015 |
| 0.6 | 0.476 ± 0.019 | 0.727 ± 0.011 |
| **0.75 (Optimal)** | **0.466 ± 0.025** | **0.738 ± 0.014** |
| 0.8 | 0.526 ± 0.014 | 0.686 ± 0.006 |

In the no-masking condition, we observed severely degraded downstream task performance, indicating that the model failed to learn meaningful, transferable representations. This result implies that without the reconstruction challenge introduced by masking, SSL pre-training becomes ineffective in capturing the complex temporal structures necessary for high-quality representation learning.

### D.10 DISSECTING THE CONTRIBUTIONS OF MAE AND JEPA OBJECTIVES

To dissect the individual contributions of the MAE and JEPA objectives, we conducted an ablation study by controlling their relative influence with the balancing factor $\tau$. A setting of $\tau = 0$ corresponds to an MAE-only model, and we also evaluated a JEPA-only model without the reconstruction term.

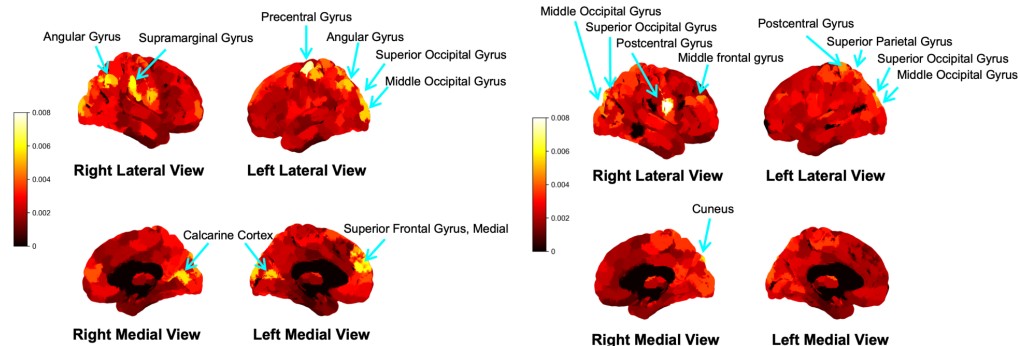

Figure 10: Brain surface visualization of IG scores. **(Left)** Age regression on the HCP-A dataset. **(Right)** Psychotic disorder diagnosis on the HCP-EP dataset.

The results in Table 18 reveal a clear synergy. The MAE-only model ($\tau = 0$) establishes a strong performance baseline, while the JEPA-only model performs poorly, indicating the latent prediction task alone is insufficient. Crucially, the unified model with an optimal balance ($\tau = 0.03$) surpasses the MAE-only baseline. This demonstrates that the JEPA objective acts as a beneficial regularizer that refines the learned features via MAE, highlighting the complementary nature of the two objectives.

Table 18: Ablation study on the MAE and JEPA components.

|  | Age (MSE) ↓ | Age (Pearson) ↑ |
|---|---|---|
| JEPA-only | $0.719 \pm 0.040$ | $0.521 \pm 0.036$ |
| $\tau = 0$ | $0.480 \pm 0.010$ | $0.717 \pm 0.006$ |
| $\tau = 0.03$ | **$0.466 \pm 0.025$** | **$0.738 \pm 0.014$** |
| $\tau = 0.1$ | $0.663 \pm 0.027$ | $0.572 \pm 0.028$ |

### D.11 EXTENDED NEUROBIOLOGICAL AND QUALITATIVE ANALYSES

#### D.11.1 NEUROBIOLOGICAL INTERPRETATION VIA INTEGRATED GRADIENTS

Integrated Gradients (IG) is an attribution analysis method from the field of explainable AI (XAI) that quantifies the contribution of each input feature—in our case, each brain ROI—to a model prediction (Sundararajan et al., 2017). The resulting IG score reflects how much a given ROI positively or negatively contributes to the model output, relative to a reference baseline input. Scores near zero indicate minimal influence. To highlight the most decisive features, we computed the absolute IG scores for each subject and normalized them across ROIs to enable comparison of relative importance. IG scores were computed from the trained models for both the HCP-A age regression and HCP-EP schizophrenia diagnosis tasks using the `Captum` (Kokhlikyan et al., 2020) library. The top 10 ROIs with the highest absolute IG scores are summarized in Table 19 and Table 20 for each task, alongside their corresponding Yeo-7 network and AAL atlas labels.

The spatial distribution of these attribution scores is visualized in Figure 10. For the age regression in HCP-A, the analysis highlighted regions integral to motor, cognitive, and sensory functions known to undergo aging-related alterations, specifically the left precentral gyrus (Zhou et al., 2020), left medial superior frontal gyrus (Lamballais et al., 2020), and bilateral angular and occipital gyri (Fjell et al., 2009). In the HCP-EP diagnosis, the analysis emphasized areas crucial to sensory perception, executive control, and self-awareness, all domains notably impaired in psychotic disorders including the right postcentral gyrus (Ferro et al., 2015), bilateral superior occipital gyri (Tohid et al., 2015), right middle frontal gyrus (Stoyanov et al., 2021), and left superior parietal gyrus (Guo et al., 2014).

#### D.11.2 QUALITATIVE ANALYSIS OF LATENT DYNAMICS

To explore the characteristics of the learned continuous-time latent dynamics, we performed a qualitative analysis of the step-wise latent displacement, defined as $\Delta z_t = z_{t+1} - z_t$, across time for

Table 19: Top 10 ROIs with the highest IG scores in the HCP-A age prediction task.

| Rank | Yeo-7 Network Label | IG Score | AAL Atlas Label |
|---|---|---|---|
| 1 | 7Networks_LH_SomMot_26 | 0.0074 | Precentral_L |
| 2 | 7Networks_LH_Default_PFC_13 | 0.0061 | Frontal_Sup_Medial_L |
| 3 | 7Networks_RH_SalVentAttn_TempOccPar_6 | 0.0060 | SupraMarginal_R |
| 4 | 7Networks_RH_Default_Par_5 | 0.0059 | Angular_R |
| 5 | 7Networks_RH_Vis_19 | 0.0057 | Calcarine_R |
| 6 | 7Networks_RH_SalVentAttn_TempOccPar_5 | 0.0056 | SupraMarginal_R |
| 7 | 7Networks_LH_Vis_24 | 0.0056 | Occipital_Sup_L |
| 8 | 7Networks_LH_Vis_21 | 0.0055 | Calcarine_L |
| 9 | 7Networks_LH_Default_Par_6 | 0.0054 | Angular_L |
| 10 | 7Networks_LH_Vis_19 | 0.0054 | Occipital_Mid_L |

Table 20: Top 10 ROIs with the highest IG scores in the HCP-EP diagnosis prediction task.

| Rank | Yeo-7 Network Label | IG Score | AAL Atlas Label |
|---|---|---|---|
| 1 | 7Networks_RH_SomMot_16 | 0.0082 | Postcentral_R |
| 2 | 7Networks_RH_Vis_29 | 0.0056 | Cuneus_R |
| 3 | 7Networks_LH_Vis_29 | 0.0049 | Occipital_Sup_L |
| 4 | 7Networks_LH_Vis_27 | 0.0047 | Occipital_Mid_L |
| 5 | 7Networks_LH_SomMot_36 | 0.0047 | Postcentral_L |
| 6 | 7Networks_RH_SalVentAttn_PFCl_1 | 0.0046 | Frontal_Mid_2_R |
| 7 | 7Networks_RH_Vis_26 | 0.0045 | Occipital_Sup_R |
| 8 | 7Networks_LH_DorsAttn_Post_14 | 0.0042 | Parietal_Sup_L |
| 9 | 7Networks_LH_SomMot_14 | 0.0042 | Postcentral_L |
| 10 | 7Networks_RH_DorsAttn_Post_4 | 0.0041 | Occipital_Mid_R |

each participant. We specifically examined the distribution of these displacements between healthy controls and participants diagnosed with ADHD from the ADHD200 dataset.

As visualized in Figure 11, we noted that healthy controls tended to exhibit larger $\Delta z$ magnitudes compared to participants diagnosed with ADHD. While the precise neurobiological meaning of the latent space requires further verification, this pattern could be interpreted as reflecting more active moment-to-moment transitions in the latent representation of healthy brain dynamics. This observation shares similarities with recent resting-state fMRI studies, which have reported reduced neural flexibility and diminished state-transition fluidity in individuals with ADHD (Yin et al., 2022; de Lacy & Calhoun, 2018). Although establishing a direct link between our latent trajectories and these neurophysiological concepts remains

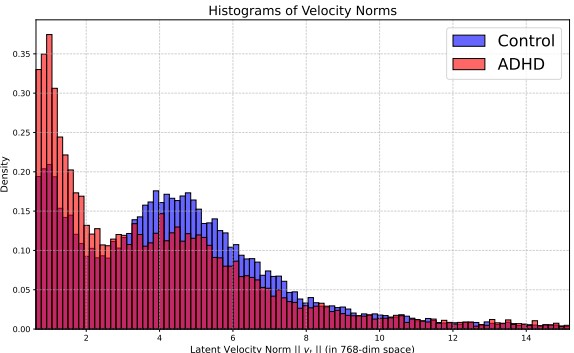

Figure 11: Qualitative analysis of learned latent dynamics via step-wise latent displacement in ADHD200. The plot compares the distribution of step-wise latent displacements ($\Delta z$) between Healthy Controls and ADHD patients. The observed lower magnitude of $\Delta z$ in the ADHD group indicates less fluid state transitions.

a subject for future research, we present this as an interesting preliminary finding that may offer a computational perspective on the dynamical differences associated with ADHD.

## D.12 ATTENTION-BASED TEMPORAL AGGREGATION STRATEGIES

In our main framework, we employed a simple mean-pooling strategy to aggregate the learned latent control signals $\alpha_t$ into the universal feature $\mathbb{A}$. This design choice was primarily driven by the need for architectural simplicity and computational efficiency, as well as to ensure a fair comparison with existing baselines that predominantly utilize mean-pooling (Caro et al., 2024; Dong et al., 2024).

However, we acknowledge that simple averaging inherently abstracts away temporal granularity, which may limit the model's ability to capture precise time-dependent dynamics, such as transient neural events or rapid neurophysiological fluctuations. To empirically investigate the potential benefits of preserving richer temporal structures, we conducted an additional ablation study using Pooling by Multihead Attention (PMA) (Lee et al., 2019). Unlike static averaging, PMA allows the model to dynamically weigh temporal features based on their relevance.

We compared the standard mean-pooling approach with PMA-based aggregation on the HCP-A dataset under the LP protocol. As summarized in Table 21, the PMA-based aggregation yields performance improvements across both age regression and gender classification tasks. These results suggest that sophisticated aggregation mechanisms capable of attending to specific temporal structures can enhance the quality of the final representation. While mean-pooling serves as a robust and efficient standard for our current framework, exploring advanced temporal aggregation methods represents a promising avenue for future research to further improve temporal fidelity and performance.

Table 21: Comparison of aggregation strategies on HCP-A.

|  | Age | | Gender | |
| --- | --- | --- | --- | --- |
|  | MSE ↓ | $\rho$ ↑ | ACC (%) ↑ | F1 (%) ↑ |
| BDO (86M) - Mean Pooling | 0.404 ±.010 | 0.768 ±.008 | 72.00 ±2.95 | 71.30 ±2.19 |
| BDO (86M) - PMA | **0.385 ±.021** | **0.772 ±.012** | **73.84 ±1.43** | **73.21 ±1.30** |

## D.13 SIMULATION STUDY FOR THE IMPACT OF TEMPORAL MISMATCH

To empirically validate the impact of temporal mismatch on representation learning, we conducted a simulation study on the HCP-A dataset. We compared our model against variants where the TR information provided to the model was intentionally distorted. This setup simulates scenarios where the modeled time scale deviates from the actual physical time scale of the data. Specifically, we tested two distortion scenarios: **Compressed TR**, where the input TR provided to the model was artificially decreased, and **Dilated TR**, where the input TR was artificially increased.

Table 22: Impact of TR distortion on HCP-A.

| Method | Age | | Gender | |
| --- | --- | --- | --- | --- |
|  | MSE ↓ | $\rho$ ↑ | ACC (%) ↑ | F1 (%) ↑ |
| BDO (86M) - Compressed TR | 0.532 ±.022 | 0.678 ±.017 | 67.59 ±2.62 | 66.48 ±2.84 |
| BDO (86M) - Dilated TR | 0.558 ±.018 | 0.660 ±.014 | 67.82 ±1.99 | 66.98 ±1.83 |
| BDO (86M) - Standard | **0.404 ±.010** | **0.768 ±.008** | **72.00 ±2.95** | **71.30 ±2.19** |

As shown in Table 22, both types of distortion led to consistent performance degradation across all metrics. These results suggest that deviating from the true physical time scale limits the model's ability to capture robust representations. This highlights the critical importance of the exact integration scheme employed in BDO, which enables the model to effectively learn from heterogeneous datasets without suffering from the temporal mismatches that plague discrete, fixed-grid approaches.

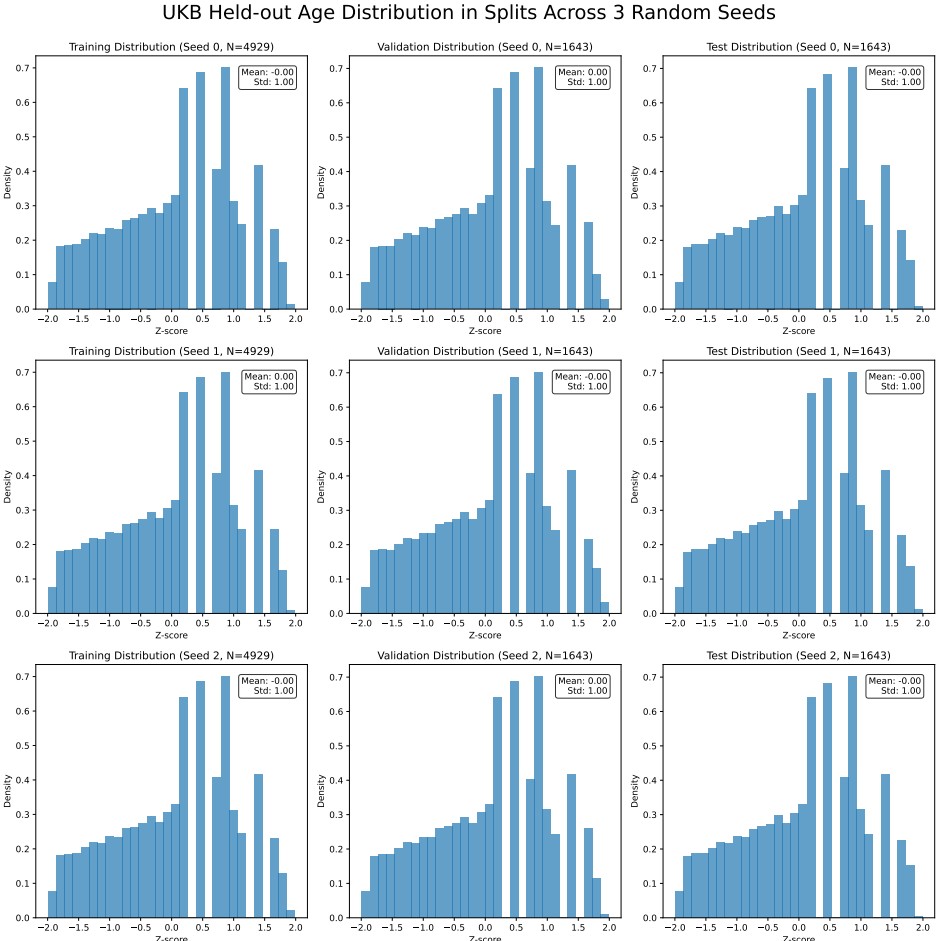

Figure 12: Age distribution across training, validation, and test splits for the UKB held-out age regression task under three different random seeds (0, 1, and 2). The dataset is partitioned using a 6:2:2 ratio, with binning-based stratified sampling applied to maintain a balanced target variable distribution. To enhance numerical stability, Z-score normalization is applied to the age variable. Each row represents a different random seed, illustrating the consistency of the sampling procedure across splits.

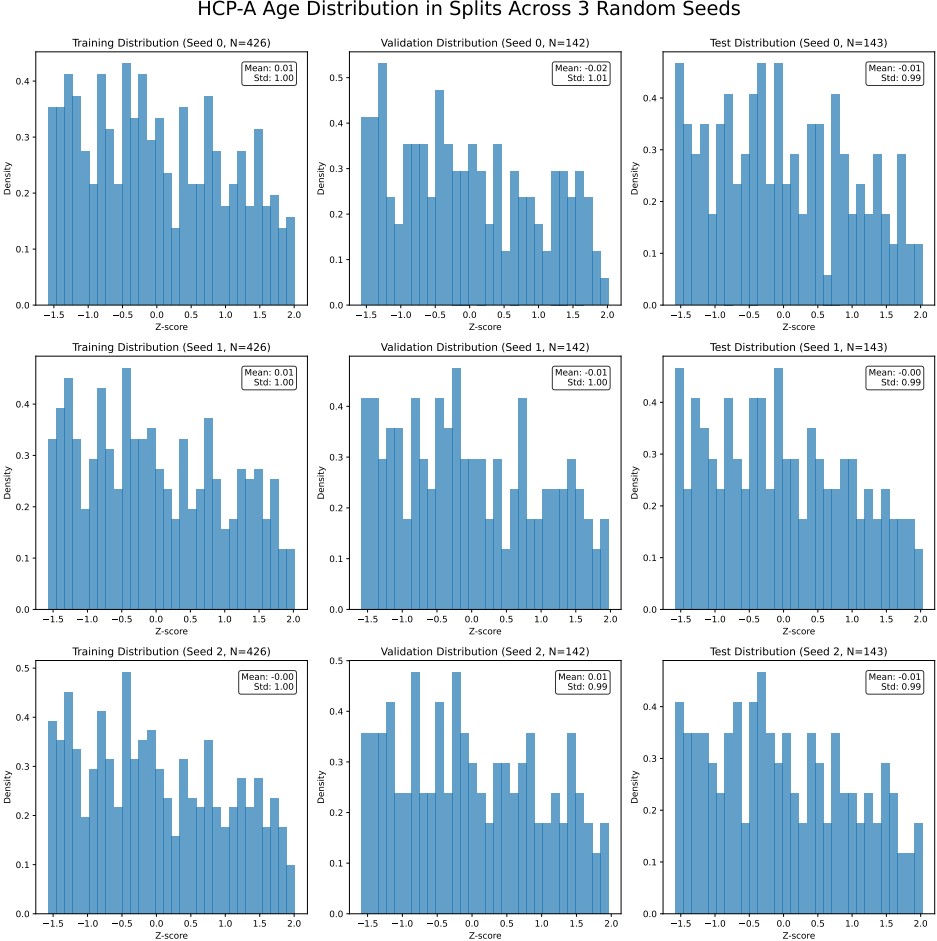

Figure 13: Age distribution across training, validation, and test splits for the HCP-A age regression task under three different random seeds (0, 1, and 2). The dataset is partitioned using a 6:2:2 ratio, with binning-based stratified sampling applied to maintain a balanced target variable distribution. To enhance numerical stability, Z-score normalization is applied to the age variable. Each row represents a different random seed, illustrating the consistency of the sampling procedure across splits.

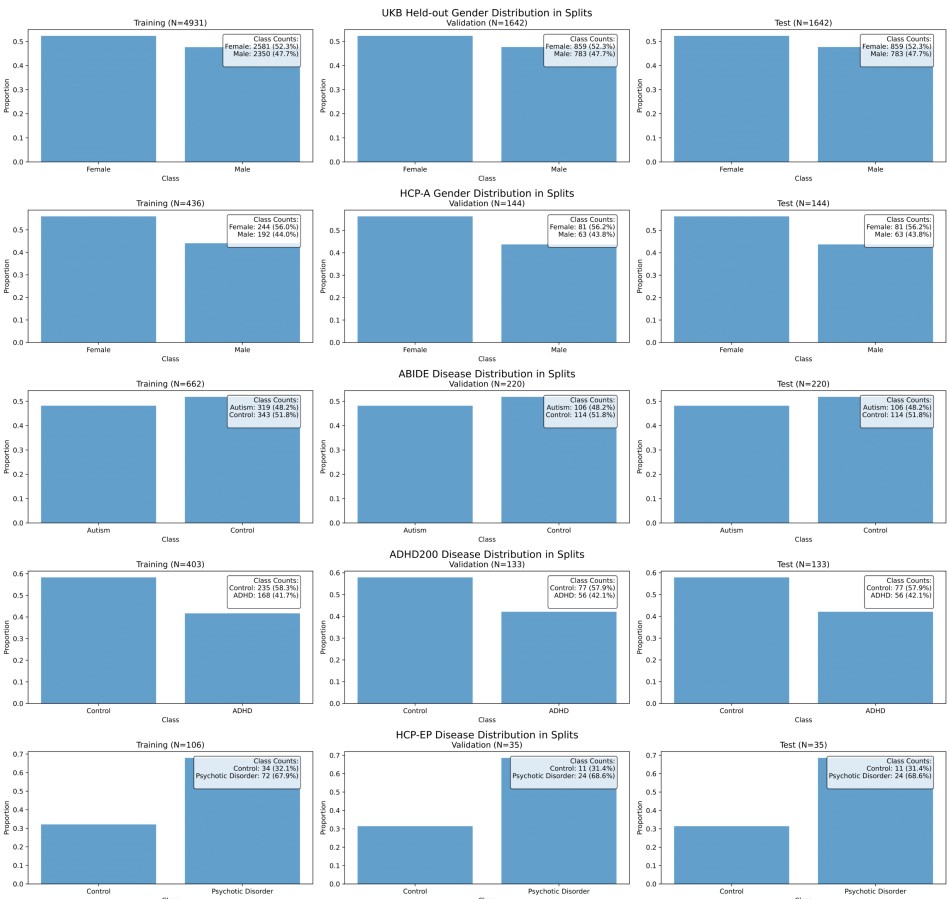

Figure 14: Label distributions across six classification tasks (UKB held-out gender, HCP-A gender, ABIDE autism, ADHD200 ADHD, and HCP-EP psychotic disorder) for training, validation, and test splits. Each row corresponds to a different task, with columns representing the proportion of samples per class across data splits. Stratified sampling ensures that label distributions remain consistent across splits, despite variations in sample composition. To illustrate this, we visualize the distributions using a single random seed (0). Gender classification tasks are divided into Female/Male categories, while disease classification tasks distinguish between Control and Patient groups (ASD vs. Control for ABIDE, ADHD vs. Control for ADHD200, and Psychotic disorder vs. Control for HCP-EP).

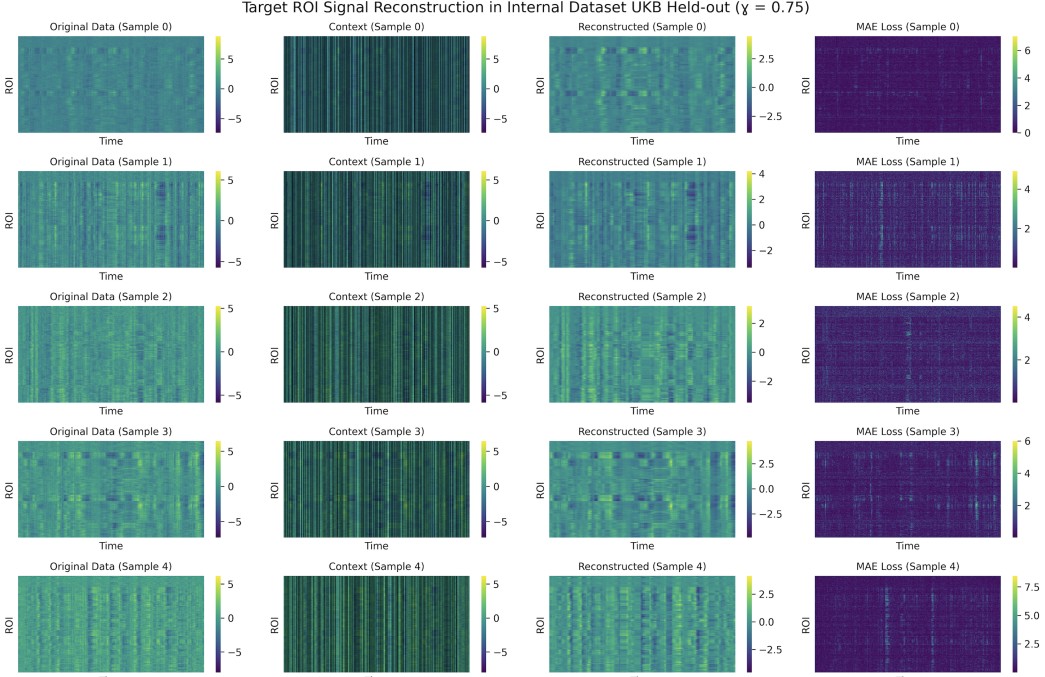

Figure 15: Reconstruction quality of BDO in the UKB held-out subset (internal dataset). Five samples are randomly drawn for visualization, with a mask ratio of $\gamma = 0.75$. Each column represents the original fMRI sample, context with masking patterns, reconstructed sample, and MAE (Mean Absolute Error) heatmaps. Although we set the mask ratio as high as $75\%$, the reconstruction quality remains robust, demonstrating that BDO efficiently captures the underlying brain dynamics and successfully reconstructs missing regions with high fidelity.

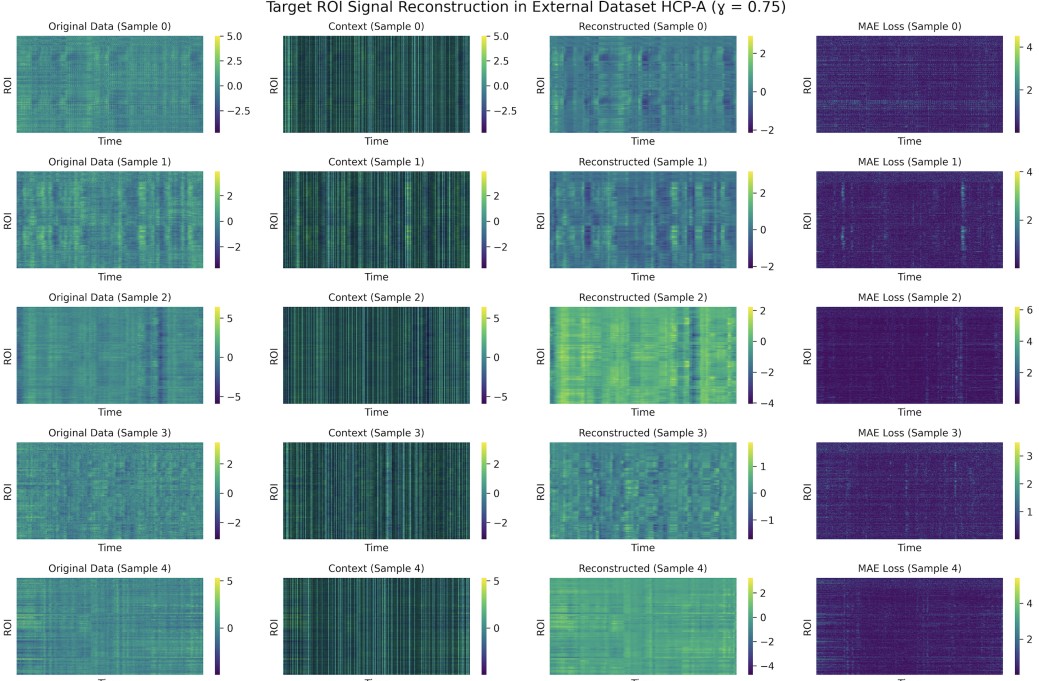

Figure 16: Reconstruction quality of BDO in HCP-A (external dataset). Five samples are randomly drawn for visualization, with a mask ratio of $\gamma = 0.75$. Each column represents the original fMRI sample, context with masking patterns, reconstructed sample, and MAE (Mean Absolute Error) heatmaps. Although we set the mask ratio as high as $75\%$, the reconstruction quality remains robust, demonstrating that BDO efficiently captures the underlying brain dynamics and successfully reconstructs missing regions with high fidelity.

