# OpenReview forum: "Stochastic Optimal Control for Continuous-Time fMRI Representation Learning"
_ICLR.cc/2026/Conference — ICLR 2026 Poster_

### Official Review · Reviewer_8VFs · 2025-10-31

**Soundness:** 3
**Presentation:** 3
**Contribution:** 3
**Rating:** 4
**Confidence:** 2

**Summary:**

This paper proposes BDO, a deep latent model for self-supervised representation learning from fMRI time series with heterogeneous temporal resolutions. The approach formulates representation learning as a stochastic optimal control (SOC) problem over continuous-time latent dynamics governed by SDEs, where the control policy acts as an encoder. The model combines masked autoencoding (MAE) and joint-embedding prediction (JEPA) in a single SOC framework and introduces a simulation-free inference scheme for computational efficiency. Trained on large-scale resting-state fMRI (UK Biobank) and evaluated on multiple public datasets (HCP-A, ABIDE, ADHD200, HCP-EP), BDO achieves higher downstream task performance (e.g., age and diagnosis prediction) compared to prior SSL baselines.

**Strengths:**

- Tackles a highly relevant and timely topic: representation learning for continuous, irregularly sampled neuroimaging data.
- The SOC formulation is mathematically elegant and could offer a principled way to model latent neural dynamics.
- Integration of MAE and JEPA in a single control-theoretic objective is technically interesting.
- Results on multiple benchmarks are empirically strong and suggest robustness across datasets.

**Weaknesses:**

1.	Limited insight into learned dynamics.
The latent SDE component is central, but no qualitative or quantitative analysis of the learned latent trajectories or their neurobiological structure is provided. As a result, it is unclear whether the model learns meaningful neural dynamics or primarily acts as an advanced denoising / feature-compression scheme.

2.	Clarity and accessibility.
The exposition is mathematically dense and often assumes familiarity with SOC and variational inference. The connection to reinforcement learning, while conceptually appealing, is not sufficiently explained or justified. Figures are mostly schematic and do not substantially aid understanding of the pipeline.

3.	Positioning and novelty.
The work seems closely related to recent latent SDE and neural ODE formulations for neuroimaging (e.g., ElGazzar & van Gerven, 2024), yet these are not cited or compared. It remains unclear whether the main novelty lies in the SOC framing or in the MAE-JEPA combination, and whether either leads to qualitatively different behavior.

4.	Evaluation design.
The chosen downstream tasks (e.g., age regression, gender classification) are convenient but not strongly motivated as demonstrations of dynamic modeling. More suitable evaluations—such as temporal forecasting, dynamic functional connectivity analysis, or representational alignment across TRs—would have made the contribution clearer.

5.	Presentation and writing.
The introduction and motivation sections are somewhat opaque; key intuitions are missing. While the results are broad, the paper feels like a mathematically strong model attached post-hoc to standard neuroimaging tasks rather than driven by neuroscientific questions.

ElGazzar, A., & Gerven, M. van. (2024). Generative Modeling of Neural Dynamics via Latent Stochastic Differential Equations (No. arXiv:2412.12112). arXiv. https://doi.org/10.48550/arXiv.2412.12112

**Questions:**

- Did you examine or visualize the inferred latent trajectories to assess whether they capture meaningful brain dynamics (e.g., temporal smoothness, subject-specific signatures)?
- How sensitive are the results to the dimensionality of the latent space?
- How do your “simulation-free” results compare against a full latent-SDE baseline with numerical solvers?
- Could the same results be achieved by a neural SDE-VAE or GRU-ODE-Bayes setup without the SOC formalism?

---

> ### Author Response · Authors · 2025-11-19
>
> Thank you for acknowledging our contributions and for your thorough, insightful review. Detailed responses to each of your comments are provided below.
>
> ----
>
> **1. Positioning and novelty**
>
> * We thank the reviewer for this question and for pointing us to this related work, which we will cite and discuss in our revision.
> To address the concern regarding our positioning, we clarify that our core novelty lies in the fundamental reframing of fMRI representation learning as a SOC problem. This SOC formulation provides a principled, mathematical foundation for our entire approach. A key result is that our objective (Eq. 9) is not an ad-hoc combination of MAE and JEPA. Instead, the _reconstruction_ (MAE-like) and _regularization_ (JEPA-like) terms are mathematically derived as the necessary components for solving the optimal control problem. In other words, the SOC framing is the primary contribution, and our specific SSL objective emerges as a natural consequence of this formulation.
> * Furthermore, our approach is fundamentally different from ElGazzar & van Gerven (2024). While both papers adopt continuous-time dynamics and latent SDEs in the neuroimaging field, their work employs a SDE-VAE with the goal of generative modeling, learning the data distribution to generate new neural spike data. In contrast, our goal is not to generate fMRI data, but to learn a powerful representation ($\mathbb{A}$) from unlabeled data that transfers effectively to diverse downstream tasks in SSL settings.
>
> ----
>
> **2. Comparison with solver-based baselines**
>
> * We thank the reviewer for raising this question regarding the benefits of our simulation-free scheme compared to standard solver-based approaches.
>
> * To empirically address this, we conducted additional experiments comparing BDO against two baselines: Latent ODE [1] and GRU-ODE-Bayes [2], utilizing their official code repositories. As their original frameworks do not address SSL scenarios, we established an empirical reference by training models in a supervised setting on the HCP-A dataset for age regression and gender classification, utilizing a composite objective of reconstruction and task-specific losses, as described in their original papers.
>
> || Age (MSE) ↓ | Age (Pearson) ↑ | Gender (Acc.) ↑ | Gender (F1) ↑ |
> |-|-|-|-|-|
> | Latent ODE | 0.556 ± .049 | 0.671 ± .042 | 64.22 ± 3.88 | 63.10 ± 3.32 |
> | GRU-ODE-Bayes | 0.587 ± .046 | 0.652 ± .039 | 63.30 ± 3.65 | 62.18 ± 3.24 |
> | BDO(86M)-LP | 0.404 ± .010 | 0.768 ± .008 | 72.00 ± 2.95 | 71.30 ± 2.19 |
> | BDO(86M)-FT | 0.273 ± .010 | 0.851 ± .006 | 79.40 ± 4.07 | 78.98 ± 4.38 |
>
> * As shown in the table, BDO significantly outperforms both supervised baselines even in the LP setting. While employing numerical solvers could theoretically allow for modeling even more complex ODE/SDE dynamics, their high computational cost acts as a severe bottleneck for pre-training on massive datasets. By bypassing the bottlenecks of numerical integration, our simulation-free framework enables efficient large-scale SSL, achieving SOTA performance. Thus, while the SOC formalism provides the theoretical basis for our SSL objective, the simulation-free scheme serves as the practical engine that enables scaling to the high dimensionality and complexity of neuroimaging data.

---

> ### Author Response · Authors · 2025-11-19
>
> **3. Qualitative analysis**
>
> * We thank the reviewer for this constructive feedback. To assess whether the latent SDE captures meaningful neural dynamics, we analyzed the step-wise latent displacement $\Delta z = z_{t+1}− z_{t}$ across time for each participant. Interestingly, we observed that healthy controls exhibited larger $\Delta z$ magnitudes than ADHD participants, indicating richer moment-to-moment transitions in latent state space as illustrated in the newly added Figure 16 of Appendix E.
>
> * Although research on the latent dynamical structure of ADHD is still limited, recent resting-state fMRI studies have shown reduced neural flexibility and diminished state-transition fluidity in individuals with ADHD [3,4]. The trend we observed in latent displacement is broadly consistent with these findings, although the exact meaning of the latent trajectories is not yet fully understood. We consider this an interesting observation that may motivate future work linking latent SDE models with established frameworks of brain dynamics.
>
> ----
>
> **4. Quantitative analysis**
>
> * We appreciate the reviewer’s suggestion to include evaluations that more directly assess dynamic modeling capabilities, such as temporal forecasting. We agree that while our downstream classification tasks imply the capture of dynamic features, forecasting provides a more rigorous test of whether the model has learned the underlying causal dynamics.
>
> * To address this, we refer to the masked interpolation analysis provided in Appendix Figures 14 and 15. In these experiments, we evaluated the model's ability to reconstruct missing time points under a high mask ratio of 75%. Notably, despite being pre-trained solely on the UKB dataset, the model successfully generalizes to the unseen HCP-A dataset. The results demonstrate that BDO effectively preserves global temporal patterns and continuity even when 75% of the sequence is masked.
>
> * While the reconstruction quality is decent, we emphasize that perfect signal reconstruction is not our primary objective. In our SOC framework, the reconstruction objective serves as a pretext task to force the control policy to learn generalizable representations. Therefore, the model's ability to approximate these missing signals suggests that the learned representation captures the essential temporal structure of the fMRI signal.
>
> ----
>
> **5. Latent dimensionality**
> * We thank the reviewer for this question. Actually, the choice of **latent dimensionality** was tied to our scalability analysis (Section 3.2 and Appendix D.6), where we systematically evaluated model variants with varying latent dimensions (Tiny : 192, Small : 384, Base : 768). Specifically, we evaluated the three BDO variants with increasing model parameters and latent dimensions. These experiments demonstrate a clear and positive correlation between the latent dimensionality and downstream task performance. This confirms that a larger latent space allows the model to capture more meaningful and transferable representations, validating our choice of $d=768$ for the primary BDO (86M) model.
> ----
>
> **6. Clarity and accessibility**
> * We appreciate the reviewer’s feedback regarding the density of the mathematical exposition. If the reviewer could kindly point out any other specific sections that were particularly opaque, we would be eager to address them directly in the revision. We plan to utilize the additional 1-page allowance granted during the rebuttal to include a dedicated schematic figure and extended text that visually illustrates the core mechanism of our SOC framework and explicitly builds intuition before delving into the formal mathematical derivations.
> ----
>
> We sincerely thank the reviewer for their thoughtful and detailed feedback. Your questions regarding the validation of learned dynamics and comparison with other baselines have driven us to significantly strengthen our empirical evaluation. We hope that our response addresses most of the concerns raised. We remain fully available to answer any further questions or provide additional clarifications during the rebuttal period.
>
> ----
>
>     [1] Rubanova et al., Latent ODEs for Irregularly-Sampled Time Series
>     [2] De Brouwer et al., GRU-ODE-Bayes: Continuous modeling of sporadically-observed time series
>     [3] Yin et al., Altered neural flexibility in children with attention-deficit/hyperactivity disorder
>     [4] Lacy and Calhoun, Dynamic connectivity and the effects of maturation in youth with attention deficit hyperactivity disorder

---

> > ### Comment · Reviewer_8VFs · 2025-11-27
> >
> > Thank you for the detailed and constructive response. The clarification of the SOC formulation as the primary contribution, together with the newly added comparisons to solver-based latent models and the additional latent-dynamics analyses, significantly strengthens the paper. In particular, the empirical comparison to Latent ODE and GRU-ODE-Bayes helps clarify the practical advantages of the simulation-free approach for large-scale SSL. While some interpretability questions remain open, I am satisfied that the main concerns raised in my initial review have been addressed.

---

> > > ### Author Response · Authors · 2025-12-01
> > >
> > > We sincerely appreciate your constructive comments throughout the review process. We are pleased that the clarification on the SOC formulation and the new empirical comparisons with solver-based models helped resolve your concerns. Your feedback has been invaluable in improving the quality of our work.

---

### Official Review · Reviewer_cuG3 · 2025-10-31

**Soundness:** 4
**Presentation:** 4
**Contribution:** 4
**Rating:** 8
**Confidence:** 4

**Summary:**

The paper proposes a self-supervised learning framework called Brain Dynamics with Optimal Control (BDO) particularly developed to meet the challenges arising when dealing with fMRI data. It consists in explicitly modelling continuous-time latent brain dynamics using stochastic optimal control (SOC). Unlike existing methods that discretize or average fMRI signals and lose temporal information, BDO treats brain activity as a stochastic differential equation (SDE), capturing multi-scale temporal structure while being robust to heterogeneous sampling protocols. Within the SOC formulation, the framework unifies two major SSL paradigms, Masked Autoencoding (MAE) and Joint Embedding Predictive Architecture (JEPA), allowing the model to learn compact, control-based representations of brain dynamics. Furthermore, to ensure scalability, the authors introduce a simulation-free inference strategy based on locally linear SDE approximations. Experiments on large multi-site fMRI datasets show that BDO achieves state-of-the-art performance and strong transferability across diverse downstream tasks.

**Strengths:**

First, I find paper exceptionally well written. It posses all key qualities: clarity, smooth flow of ideas, good balance presenting both intuition and technical complexity, appendix complements adequately the main text.

Second, the paper tackles an important problem and proposes a method that is applicable beyond targeted fMRI applications.

Third, the innovation to use SOC in SSL context is, up to my knowledge, clear and it also consists in combing MAE and JEPA.  While, focusing on linear drifts and exploiting Gaussian distributions to scale-up is relatively standard idea, it is executed well and is a very good first strategy for the proposed method.

Finally, the experimental setup and baselines are adequate, and clearly shows the main benefits coming from BDO.

**Weaknesses:**

While I do not find any strong weakness of the paper, here is a few minor issues:

**Minor**
- While full names of acronyms such as JEPA and MAE are introduced in the abstract, it would be helpful to still write them in the main text when the concepts are introduced.

- Typo in Eq. (3): in drift part should read $\alpha^\theta$

**Questions:**

Ca authors add positioning of their work w.r.t. following two nicked references, and potentially add this in Appendix A?

- Hassan et al. *Identifying Latent Stochastic Differential Equations*.IEEE Signal Processing 2021

- Bartosh et al.  *SDE Matching: Scalable and Simulation-Free Training of Latent Stochastic Differential Equations*. ICML2025

---

> ### Author Response · Authors · 2025-11-19
>
> We sincerely appreciate the reviewer’s acknowledgement of our contributions, depth of analysis, and positive assessment. We are particularly encouraged by the reviewer's recognition of our SOC formulation as a mathematically elegant and innovative strategy for unifying MAE and JEPA. We also thank the reviewer for their constructive suggestions, all of which will be incorporated into the revised manuscript. Our responses to each point are provided below.
>
> ----
>
> **1. Clarifications of acronyms (MAE and JEPA)**
>
> * We agree with the reviewer's suggestion. In the revised manuscript, we will ensure that the full names for Masked Autoencoding (MAE) and Joint Embedding Predictive Architecture (JEPA) are explicitly stated when they are first introduced in the main text, in addition to the abstract, to improve clarity.
>
> ----
>
> **2. Comparison with recent SDE works**
>
> * We thank the reviewer for pointing us to these two highly relevant papers. They provide an excellent context for positioning our work, particularly regarding latent SDE identification and efficient training. As requested, we will add a detailed discussion in the Related Work section to contextualize BDO with respect to these works [1,2].
> * We will clarify that while these valuable works focus on the important problem of identifying or matching the parameters of latent SDEs from data, our work's primary contribution is to propose using the _optimal control policy_ that steers the latent dynamics to match observations as the learned, transferable representation for downstream tasks. We will make it clear that our novelty lies in formulating the SSL problem as an optimal control task, where the control policy becomes the representation itself guided by a unified MAE-JEPA objective.
>
> ----
> **3. Typo in Eq. (3)**
>
> * We thank the reviewer for spotting this error. We have carefully reviewed Equation (3) and will correct the drift term in the revised manuscript.
>
> ----
>
> Once again, we thank the reviewer for their exceptionally thorough and supportive review. These insights have helped us further strengthen and polish the manuscript.
>
> ----
>     [1] Hassan et al. Identifying Latent Stochastic Differential Equations.
>     [2] Bartosh et al. SDE Matching: Scalable and Simulation-Free Training of Latent Stochastic Differential Equations

---

> > ### Comment · Reviewer_cuG3 · 2025-11-25
> >
> > I thank the authors for their reply. I will keep my positive score.

---

> > > ### Author Response · Authors · 2025-12-01
> > >
> > > We are glad to have your support. Thank you again for your time and constructive feedback throughout the review process.

---

### Official Review · Reviewer_SCN9 · 2025-10-31

**Soundness:** 3
**Presentation:** 3
**Contribution:** 2
**Rating:** 4
**Confidence:** 4

**Summary:**

This paper proposes Brain Dynamics with Optimal control (BDO), a novel self-supervised learning
(SSL) framework for learning representations from fMRI time-series. The core problem this work
addresses is the temporal irregularity and noise inherent in multi-site fMRI datasets, which is often
handled by existing methods by discarding or discretizing temporal information.

BDO reframes fMRI representation learning as a Stochastic Optimal Control (SOC) problem. It models
latent brain activity as a continuous-time stochastic differential equation (SDE) and learns a control
policy (parameterized by a Transformer) that steers a prior SDE to a posterior SDE that best explains
the observed, noisy fMRI data. This control policy itself serves as the learned representation.

The framework provides a principled unification of Masked Autoencoding (MAE) and Joint-Embedding
Predictive Architectures (JEPA). The SOC objective is shown to be equivalent to an ELBO (MAEstyle reconstruction), while a JEPA-style latent target (from an EMA-updated encoder) is introduced
to prevent the model from overfitting to signal noise. To ensure scalability, the paper introduces a
simulation-free inference method based on a locally linear SDE assumption, which admits a closed-form
solution. In addition, the model is pre-trained on the large-scale UKB dataset and demonstrates stateof-
the-art performance on a wide range of downstream tasks, including trait and diagnosis prediction,
on both internal (UKB held-out) and external (HCP-A, ABIDE, etc.) datasets.

**Strengths:**

1. Novel and Elegant Formulation: Reframing SSL for fMRI as a Stochastic Optimal Control
problem is a highly novel and theoretically compelling contribution. This approach is a natural
and powerful fit for modeling continuous-time, noisy, and irregularly-sampled data, which is a
key challenge in fMRI analysis.

2. Principled Unification of MAE and JEPA: The paper does an excellent job of motivating
and technically unifying two powerful SSL paradigms. Deriving the MAE-like objective from
the SOC formulation (Prop 2.1) is elegant. The subsequent introduction of a JEPA-style latent
target (Eq. 7-9) to regularize this objective and avoid overfitting to noise is a well-reasoned and
technically sound improvement.

3. Impressive Computational Efficiency: A major contribution is the simulation-free inference
(Theorem 2.2). By using a locally linear SDE approximation, the authors derive a closed-form
Gaussian posterior for the latent states, which can be computed in parallel with O(log k) complexity.
This makes a theoretically complex SDE/SOC framework computationally practical for
massive datasets like UKB. The empirical result of 15 GPU hours for pre-training (Table 10) is
extremely strong compared to baselines.

4. Strong and Comprehensive Empirical Validation: The experimental results are state-of-the-art and convincing. BDO outperforms a wide array of baselines (both task-specific and
fMRI-specific SSL models) across numerous diverse tasks (age, gender, diagnosis) and datasets
(UKB, HCP-A, ABIDE, ADHD200, HCP-EP). The generalization from resting-state pre-training
to task-based fMRI (Appendix D.5) is also a strong result.

**Weaknesses:**

1. Inconsistency in Control Policy Formulation: There appears to be a critical inconsistency
between the theoretical formulation and the practical implementation of the control policy α.
The SOC theory (Eq. 3, Prop 2.1) defines a closed-loop control policy α(t,Xθt ;Y), which depends
on the current latent state Xθt . This is standard for optimal control. However, the simulation-free implementation (Eq. 10) and the architecture diagram (Fig. 5) use what appears to be an
open-loop policy αθti(Y), which is pre-computed from the fMRI data Y and does not depend on
the current state Xθt . This is a significant simplification that deviates from the theory. While it
enables the efficient closed-form solution, the paper needs to explicitly state this simplification
and justify how the SOC theory (Prop 2.1) still holds for this open-loop policy.

2. Simplistic Final Representation: The model’s universal feature A is derived by a simple
mean-pooling of the control policy sequence {αt}. This design choice seems to contradict the
paper’s primary motivation, as it discards the rich temporal structure that the continuous-time
SDE model worked so hard to capture. It is surprising that this simple pooling outperforms
other methods, and it feels like a missed opportunity.

3. Clarity on Linear SDE Parameters: The locally linear SDE in Eq. 10 (dXθ
t = [−DtiXθt +αθti ]dt + dWt) is key to the model’s efficiency. However, it is unclear how the drift matrix Dti
is parameterized and learned. Theorem 2.2 states Dti = V ΛtiV ⊤, which implies it changes over
time. How are the shared eigenbasis V and the eigenvalues Λti generated by the encoder network?
This is a crucial, missing architectural detail.

4. Baseline Reproducibility: The authors note in Appendix D.11 that they could not reproduce
the Brain-JEPA baseline with their own preprocessing, which is a minor red flag. While they
provide a fair comparison using the original Brain-JEPA preprocessing (where BDO still wins),
it would be more convincing to understand *why* the reproduction failed.

**Questions:**

1. Can you please clarify the apparent inconsistency in the control policy? Is the implemented
policy αθti(Y) (open-loop) or α(t,Xθt ,Y) (closed-loop)? If it is open-loop, how is this reconciled
with the SOC theory in Proposition 2.1, which is based on a state-dependent policy?

2. The final feature A is a mean-pooling of the control policy, which discards all temporal information.
Why was this chosen? Have you experimented with using the learned latent states Xt
or a more temporally-aware aggregation of {αt} (e.g., attention, final hidden state) as the final
representation?

3. How is the drift matrix Dti = V ΛtiV ⊤ in the linear SDE (Eq. 10) parameterized? Does the
Transformer encoder output V and all {Λti}? Is V shared across all time steps and subjects?

4. In Appendix D.11, you mention a difference in per-sample zero-mean normalization might be
responsible for the failed Brain-JEPA reproduction. Could you elaborate on this? This seems
like a subtle difference to cause a full failure, and understanding it would strengthen the baseline
comparisons.

**Details Of Ethics Concerns:**

This is a technically interesting paper . The formulation of fMRI SSL as an SOC problem is a
novel, first-principles approach that is both theoretically elegant and a natural fit for the problem
domain. The unification of MAE and JEPA is well-motivated, and the engineering to make this complex
framework (SDEs) scalable via a simulation-free approach is a significant achievement. Moreover, the
empirical results are state-of-the-art, comprehensive, and robust. However, the paper would benefit
from more comprehensive comparisons to continuous-time SSL baselines, deeper ablation studies on
feature aggregation and long-term modeling, and a more thorough discussion of ethical implications.
The primary weaknesses are a lack of clarity on a key theoretical vs. implementation detail (open-loop
vs. closed-loop control) and a surprisingly simple final feature representation (mean-pooling) that
seems to under-utilize the model’s dynamic capabilities. These weaknesses are addressable and do not
detract from the novelty and significance of the core contributions. The paper presents a new and
powerful direction for fMRI representation learning.

---

> ### Author Response · Authors · 2025-11-19
>
> Thank you for carefully reviewing our manuscript and for your helpful comments. Our responses to each comment are provided below.
>
> ----
> **1. Remarks on control formulation.**
>  * We really appreciate the reviewer for pointing out the control formulation. Our SOC theory (in particular the line 178 “where the equality holds at the optimal control policy” at Proposition 2.1) is indeed derived under a standard Markov (closed loop) control, because the result follows by verification of the HJB equation. However, if we look only at the ELBO itself, from a variational inference point of view our ELBO can also be written as $\text{ELBO} \geq D_{\text{KL}}(\mathbb{P}^{\alpha^{\theta}} | \mathbb{P}^{\star}) = -\mathcal{J}(\alpha^{\theta}, \mathcal{Y})$.
>
> * Above inequality holds for any admissible control policy, so it remains valid in the non-Markov (open-loop) case as well. Hence, in Eq.(10), we intentionally restrict the class of control functions for optimization problems to a more restricted form (open-loop) and instead only depend on the observation Y, in order to enable simulation free inference as the reviewer correctly noted and the resulting controlled process have more explicit dependency with Y. We agree that  optimizing over this restricted family make bound may become looser, but the ELBO itself and its interpretation as a lower bound are not affected.
> * In practice, as long as the non-Markov (open loop) control parameterization has sufficiently high capacity, we **expect** that gradient based optimization can drive the loss close to the optimal ELBO (a similar discussion appears in [1], Remark 3.9), and the resulting control will be a valid approximation of the true optimal control and suitable for the purposes of our SSL objective.
> * This simplification of the control formulation is worth explicit discussion, and we deeply thank the reviewer for raising this point. We add a discussion of the control formulation in Appendix C.1.
> ----
>
> **2. Aggregation strategy**
>
> * Our motivation for modeling continuous-time dynamics primarily arises from the need for SSL methods that can naturally accommodate heterogeneous datasets with varying temporal resolutions and irregular sampling intervals, as commonly encountered in large-scale fMRI studies.
>
> * To achieve this task-agnostic goal, we deliberately employed a simple mean-pooling aggregation of the learned latent control signals $\alpha^{\star}$ to derive the universal feature $\mathbb{A}$. This design choice ensures architectural simplicity, computational efficiency, and fair comparison with existing methods, which commonly use the same mean-pooling strategies.
>
> * We fully agree that mean-pooling inherently abstracts away temporal granularity, potentially limiting the fidelity of time-dependent information. Indeed, alternative aggregation strategies such as attention-based pooling [2,3] could preserve richer temporal structures, which may be beneficial for certain tasks sensitive to precise temporal dynamics—for example, detecting transient neural events or characterizing rapid neurophysiological fluctuations.
>
> * To empirically validate this, we conducted an additional experiment using Pooling by Multihead Attention (PMA) [2], which allows the model to dynamically weigh temporal features rather than simply averaging them. We compared this against our standard mean-pooling approach on the HCP-A linear probing task.
>
> || Age (MSE) ↓ | Age (Pearson) ↑ | Gender (Acc.) ↑ | Gender (F1) ↑ |
> |-|-|-|-|-|
> | BDO(86M)-LP | 0.404 ± .010 | 0.768 ± .008 | 72.00 ± 2.95 | 71.30 ± 2.19 |
> | BDO(86M)-LP-PMA | 0.385 ± .021 | 0.772 ± .012 | 73.84 ± 1.43 | 73.21 ± 1.30 |
>
> * As shown in the table, the PMA-based aggregation yields performance improvements. This confirms the reviewer's hypothesis that preserving richer temporal structures through attention mechanisms enhances the quality of the final representation.
>
> * Future extensions of our framework could benefit from exploring more sophisticated temporal aggregation methods, potentially improving both temporal fidelity and task-specific performance. Such directions represent promising avenues for subsequent research.

---

> ### Author Response · Authors · 2025-11-19
>
> **3.SPD matrix parametrization**
>
> * We appreciate the reviewer for highlighting that this architectural detail was not clearly described. In our implementation, the eigenbasis $V$ is a single trainable matrix shared across all time steps. We parameterized $V$ as orthogonal during training, following the orthogonalization scheme in [4]. The time dependent diagonal-eigenvalues $\Lambda_{t_i}$ are produced from the encoder features through an affine combination of a small set of learned base matrices.
> * Specifically, we introduce $L$ base trainable matrices $\{\Lambda^l\}{l=1}^L$ and for each step $t_i$, the transformer encoder outputs a vector of coefficients $w_{t_i}$, then we form $\Lambda_{t_i} = \sum_{l=1}^L w^l_{t_i} \Lambda^l$, which yields a different spectrum at each $t_i$​ while keeping the parameterization compact and stable. This parameterization strategy is followed from [1]. We add a paragraph in Appendix D.3 and provide the parametrization strategy to make the construction of $D_{t_i}$ fully explicit.
> ----
>
> **4. Baseline reproducibility**
>
> * We thank the reviewer for this insightful question. We agree that understanding the cause of the reproduction issue under our preprocessing pipeline is scientifically important. We attribute the strong sensitivity of Brain-JEPA to our pipeline to the inherent susceptibility of standard JEPA objectives against input distribution shifts.
>
> * Existing JEPA frameworks are known to be brittle and rely on a delicate balance of hyperparameters to prevent representation collapse, which makes them sensitive to changes in data distribution [5,6]. In our setting, the per-sample zero-mean centering step removes subject-level global offsets and thus fundamentally alters these statistics, especially for demographic tasks where the global mean carries substantial signal. This shift likely pushes the data distribution outside the effective range of the original baseline heuristics tuned for uncentered signals.
>
> * Our ablation study further supports this interpretation: when we remove the MAE reconstruction term and retain only the JEPA objective, HCP-A age regression LP performance drops sharply as shown in Table 12 of Appendix D.9 (attached below). This suggests that a pure JEPA objective is indeed prone to failure under our preprocessing, whereas in BDO the MAE component provides a robust learning signal and the JEPA term acts primarily as a secondary regularizer.
>
> ||Age (MSE) ↓|Age (Pearson) ↑|
> |-|-|-|
> |JEPA-only|0.719±.040|0.521±.036|
> |$\tau\=0$ (MAE only)|0.480±.010|0.717±.006|
> |$\tau\=0.03$ |0.466±.025|0.738±.014|
> |$\tau\=0.1$ |0.663±.027|0.572±.028|
>
> ----
>
> We sincerely thank the reviewer for their technical insights, regarding the control formulation and architectural specifications. We believe these revisions have significantly improved the quality and precision of our manuscript. We are of course happy to discuss any remaining questions in further detail.
>
> ----
>
>     [1] Park et al., Amortized Control of Continuous State Space Feynman-Kac Model for Irregular Time Series
>     [2] Lee et al., Set Transformer: A Framework for Attention-based Permutation-Invariant Neural Networks
>     [3] Horn et al., Set Functions for Time Series
>     [4] Casado, Trivializations for Gradient-Based Optimization on Manifolds
>     [5] Bardes et al., VICReg: Variance-Invariance-Covariance Regularization for Self-Supervised Learning
>     [6] Balestriero and LeCun, LeJEPA: Provable and Scalable Self-Supervised Learning Without the Heuristics

---

### Official Review · Reviewer_9gcz · 2025-10-31

**Soundness:** 3
**Presentation:** 3
**Contribution:** 2
**Rating:** 4
**Confidence:** 4

**Summary:**

The authors propose a stochastic optimal control framework as a way to view fMRI pre-training. The model they end up with is intuitively similar to a variational auto encoder that is trained with masked auto encoding and is regularized with a Joint Embedding Predictive Architecture (JEPA) objective function. The model outperforms baselines across a variety of tasks and datasets, and the model exhibits good scalability.

**Strengths:**

The authors perform a wide range of experiments, describe an interesting framework for fMRI pre-training, and thoroughly ablate their model.

**Weaknesses:**

Major weaknesses:
1) No comparison to recent foundation models, and limited related work in the main text. The authors have moved most of their related work to the Appendix. Especially given how quickly the foundation model field is evolving, I believe it is important to have a more substantial related work section than is currently present in the paper. Moreover, a few key foundation models are missing from the evaluations, nor do they discuss in depth differences between their approach and BrainJEPA for example. Namely, BrainJEPA [1] and BNT [2]. I cannot recommend the paper for publication without fair comparisons to these works.
2) The authors claim that their stochastic optimal control framework solves heterogeneity in repetition time (TR) for the fMRI data, but it is unclear why this is true given the authors' model choices. Specifically, they mention on Lines 42-45 that "... existing SSL methods often fail to fully capture the inherent temporal dynamics of fMRI by treating the time-series data as segmented patches or static connectivity graphs as described in Figure 2.". However, this is not necessarily true. Transformers can still capture dynamics even when they use temporal patching. For example POYO [3] and NDT [4] that show that transformers can still capture dynamics even with temporal patching in the computational neuroscience domain. Moreover, the authors say their framework solves heterogeneous TRs, but still use a temporal transformer (see Figure 5) to create embeddings. It is unclear to me how the authors fix heterogeneous TRs when the transformer is pre-trained on a dataset with a certain spacing between patches. The current transformer does not seem to solve this issue even when incorporated into a stochastic optimal control policy.

Minor weaknesses/types/grammar:
- The authors keep mentioning (e.g. on Lines 76-79) that fMRI data is high-dimensional, which is true, but then also use anatomical regions of interest (ROIs) to reduce the data to 450 spatial dimensions. It is unclear to me why the authors seem to focus so much on the high dimensionality of the data when they do not contribute a specific way to address this fact.
- To the previous point, the authors do not ablate the ROI choice.
- Theorem 2.2 can be moved to the Appendix because it is not a novel theorem but a direct consequence of the papers cited in the previous paragraph. Especially given the fact that the authors moved their related work to the Appendix, I believe these types of derivations can be swapped for a more substantial related work section. If anything some of the mathematical derivations, including but not limited to this theorem, take away from the main contributions this paper makes.
- The "Masked auto encoders" paragraph on page 5 repeats many of the ideas/claims in paragraph 2.2, I think either of these paragraphs can be shortened.
- Lines 152-153: "The SOC proves ..." -> SOC provides
- Lines 242-243: "..., we use the MAE ..." -> we use an MAE

[1] Dong, Z., Li, R., Wu, Y., Nguyen, T. T., Chong, J., Ji, F., ... & Zhou, J. H. (2024). Brain-jepa: Brain dynamics foundation model with gradient positioning and spatiotemporal masking. Advances in Neural Information Processing Systems, 37, 86048-86073. \
[2] Kan, X., Dai, W., Cui, H., Zhang, Z., Guo, Y., & Yang, C. (2022). Brain network transformer. Advances in Neural Information Processing Systems, 35, 25586-25599. \
[3] Azabou, M., Arora, V., Ganesh, V., Mao, X., Nachimuthu, S., Mendelson, M., ... & Dyer, E. (2023). A unified, scalable framework for neural population decoding. Advances in Neural Information Processing Systems, 36, 44937-44956. \
[4] Ye, J., & Pandarinath, C. (2021). Representation learning for neural population activity with neural data transformers. arXiv preprint arXiv:2108.01210.

**Questions:**

1) On lines 84-85 what do the authors mean by "..., which encodes multi-scale brain activity onto a single continuous real-time axis."?
2) Line 121-122: Why do the authors assume the data is being governed by an Ito diffusion process? There is a rich literature of biophysical dynamical models for fMRI data, including the FitzHugh–Nagumo model [1], the Wilson-Cowan model [2], and the Kuramoto model [3]. The authors provide no biophysical or neuroscientific explanation for this design choice.
3) In the same paragraph on Lines 131-132 the authors mention "... we initially assume the underlying changes are purely random, acknowledging the difficulty in modeling the intricate nature of fMRI signals beforehand.". Why do the authors not just use a learnable prior or infer an initial condition?



[1] Ghosh, A., Rho, Y., McIntosh, A. R., Kötter, R., & Jirsa, V. K. (2008). Cortical network dynamics with time delays reveals functional connectivity in the resting brain. Cognitive neurodynamics, 2(2), 115-120. \
[2] Deco, G., Jirsa, V., McIntosh, A. R., Sporns, O., & Kötter, R. (2009). Key role of coupling, delay, and noise in resting brain fluctuations. Proceedings of the National Academy of Sciences, 106(25), 10302-10307. \
[3] Cabral, J., Hugues, E., Sporns, O., & Deco, G. (2011). Role of local network oscillations in resting-state functional connectivity. Neuroimage, 57(1), 130-139.

---

> ### Author Response · Authors · 2025-11-19
>
> We sincerely appreciate the reviewer’s valuable feedback and the depth of their analysis. Detailed responses to each point are provided below.
>
> ----
>
> **1. Comparisons with Brain-JEPA [1] and BNT [2]**
> * We fully agree with the reviewer that, given the rapid evolution of the fMRI foundation model field, a substantial related work section and direct comparisons to recent foundation models are crucial. We thank the reviewer for specifically highlighting the need for comparisons against Brain-JEPA [1] and BNT [2] and would like to clarify that evaluations for both models were, in fact, included in our original submission.
>    - **Brain-JEPA [1]**: A direct and fair comparison with Brain-JEPA was also included in the original submission's Appendix (Section D.11, Table 15). As noted in the appendix, this comparison required careful handling due to performance sensitivity to preprocessing (see the fourth response to Reviewer SCN9 for detailed discussion). Our results demonstrate that BDO, when trained under identical preprocessing outperforms Brain-JEPA on the HCP-A dataset.
>    - **BNT [2]**: We apologize for the confusion. We did include BNT as a primary baseline in all relevant tables (Tables 1, 2, and 3). To maintain consistency with the "BrainNet" series of models such as BrainNetCNN, BrainGNN, we abbreviated it as **BrainNetTF**. As the results show, our pretrained BDO model outperforms this powerful supervised baseline.

---

> ### Author Response · Authors · 2025-11-19
>
> **2. Robustness to irregular sampling intervals**
> * We thank the reviewer for this insightful feedback, as it allows us to clarify the core motivation of our continuous-time framework. We fully agree with the reviewer that transformer architectures are capable of capturing temporal dynamics, as demonstrated by models like POYO [3] and NDT [4].
> * However, our critique specifically targets existing patch-based models in the fMRI domain, such as BrainLM and Brain-JEPA. Our claim is _not_ that these transformers cannot learn temporal dynamics, but rather that they are an unnatural fit for the specific SSL scenario where a model must be pre-trained on multiple, heterogeneous datasets combined. Specifically, these models process fMRI data by tokenizing time-series segments into patches on a fixed discrete grid. This structural rigidity leads to unavoidable _temporal mismatches_ when generalizing across multi-site datasets with varying acquisition protocols.
> * Let us use a concrete example: a model pre-trained on UKB (TR = 0.735s) that is transferred to ABIDE (TR ranges from 1.5s to 3.0s). Existing patch-based models typically employ downsampling to align these datasets. Specifically, they downsample the higher-frequency UKB data by taking every third timepoint (sampling interval = 3). However, even with this adjustment, a single input patch for UKB represents 2.205s ($3 \times 0.735$s), whereas a single patch for an ABIDE subject might represent 2.5s. Consequently, the transformer learns "Patch $k \to$ Patch $k+1$" dependencies, but "Patch $k$" itself contains inconsistent underlying temporal information depending on its source. This inconsistency creates an unnatural burden that hinders learning robust, generalizable temporal dynamics.
> * In contrast, BDO solves this problem naturally by adopting the continuous dynamics.
>    - **Real-time axis**: Instead of discrete steps, we place all observations onto a single, unified real-time axis (Figure 3). A datapoint is processed as $(y_{t_i}, t_i=i \times TR)$ where $t_i$ represents the actual physical time.
>    - **Exact integration**: The SDE solver explicitly uses the exact time interval ($\Delta t_i = t_i - t_{i-1}$) specific to each subject to integrate the dynamics. Therefore, BDO naturally accommodates irregular sampling without distorting the real time.
> * This also clarifies the reviewer's question regarding our transformer architecture (Figure 5). It is not a standard patch-based sequence processor trained on a fixed spacing. Instead, it functions as an encoder that takes a set of observations $(y_{t_i})$ where each token is explicitly combined with its real-valued, continuous time-stamp $t_i$ using continuous positional encodings, similar to the approach used in POYO. This time-aware information is then used to parameterize the control policy $\alpha^\theta$ for the underlying continuous-time SDE. Therefore, by relying on the SDE to model latent dynamics on a continuous real-time axis, BDO ensures robustness to irregular TRs, thereby avoiding temporal mismatches.
> * To empirically validate the impact of this temporal mismatch, we conducted an additional simulation study on the HCP-A dataset. Using linear probing, we compared our BDO (86M) model against a distorted variant where the TR information provided to the model was intentionally distorted to simulate the temporal mismatches inherent in fixed-grid discrete models. Specifically, we tested two distortion scenarios: Compressed TR, where the input TR was artificially decreased and Dilated TR, where the input TR was artificially increased.
> || Age (MSE) ↓ | Age (Pearson) ↑ | Gender (Acc.) ↑ | Gender (F1) ↑ |
> |-|-|-|-|-|
> | BDO(86M)-LP | 0.404 ± .010 | 0.768 ± .008 | 72.00 ± 2.95 | 71.30 ± 2.19 |
> | BDO(86M)-LP-Compressed | 0.532 ± .022 | 0.678 ± .017 | 67.59 ± 2.62 | 66.48 ± 2.84 |
> | BDO(86M)-LP-Dilated | 0.558 ± .018 | 0.660 ± .014 | 67.82 ± 1.99 | 66.98 ± 1.83 |
> * As shown in the table, both types of temporal distortion led to consistent performance degradation across all metrics. These results suggest that deviating from the true physical time scale limits the model’s ability to capture robust representations, thereby highlighting the importance of the exact integration scheme employed in BDO.

---

> ### Author Response · Authors · 2025-11-19
>
> **3. Rationale for adopting the Ito process as prior**
> * The Ito process provides a powerful and robust framework for modeling continuous-time stochastic dynamics. Its integration into modern deep learning architectures has proven highly effective for capturing the behavior of complex, noisy, and irregularly sampled time-series data [5]. Indeed, recent studies in computational neuroscience have successfully employed Ito SDEs to model complex brain dynamics [6,7].
> * This choice is further grounded in biophysics. The membrane potential of individual biological neurons, driven by random synaptic inputs, follows an Ito SDE [8]. When these microscopic stochastic fluctuations are aggregated across neural populations, they manifest as population-level variability and this variability might appear as the stochastic BOLD signals measured in fMRI through neurovascular coupling.
> * In our study, we adopted the ito SDE framework based on the assumption that it is difficult to model the intricate nature of fMRI signals beforehand. Rather than imposing a predefined, rigid model, we initially assume the underlying changes are purely random. This assumption, however, does not mean our model is simplistic. It refers to the foundational Wiener process ($dW_t$) of the SDE. The core of our framework is to then learn the complex, data-driven functions (the drift $f$ and diffusion $g$) that govern and control these stochastic dynamics. This approach, which is analogous to learning a prior on the dynamics, allowed us to successfully model the latent dynamics underlying the fMRI data.
> * We appreciate the valuable suggestion and agree that the FitzHugh–Nagumo, Wilson–Cowan, and Kuramoto models may provide more explicit biophysical descriptions of neuronal dynamics. In future work, we plan to integrate such mechanistic formulations into our framework to develop more biophysically grounded models.
> ----
> **4. Ablation study on ROI parcellation**
> * We thank the reviewer for this valuable suggestion. As requested, we have conducted a new ablation study to investigate the effect of ROI parcellation granularity. We re-trained our BDO (86M) model using the Schaefer atlas with 200, 400, 600, 800, and 1000 parcellations. We then evaluated the linear probing performance on the HCP-A dataset. These new models were trained for 100 epochs (compared to 200 epochs for the models in the main paper).
>
> | # ROIs | Age (MSE) ↓ | Age (Pearson) ↑ | Gender (Acc.) ↑ | Gender (F1) ↑ |
> |-|-|-|-|-|
> |  200 | 0.616 ± .061 | 0.641 ± .042 | 67.82 ± 0.87 | 66.04 ± 2.79 |
> |  400 | 0.563 ± .026 | 0.668 ± .014 | 68.06 ± 1.50 | 67.02 ± 1.31 |
> |  600 | 0.557 ± .038 | 0.678 ± .028 | 69.91 ± 1.43 | 68.76 ± 1.32 |
> |  800 | 0.571 ± .058 | 0.669 ± .039 | 73.38 ± 0.87 | 72.82 ± 1.20 |
> | 1000 | 0.537 ± .068 | 0.695 ± .050 | 74.31 ± 3.00 | 73.75 ± 2.84 |
>
> * The results show a clear trend: increasing the spatial resolution generally improves downstream performance, consistent with prior findings that finer parcellations can capture richer representations of brain activity [7,8]. This analysis confirms that our BDO framework effectively leverages more granular spatial information and scales well with increased feature dimensionality.
> * Furthermore, we wish to highlight the computational efficiency of our BDO framework.
>    - Many existing transformer-based models such as BrainLM, Brain-JEPA, and BrainMass, process fMRI data by tokenizing ROIs into a long sequence. Consequently, increasing the number of ROIs by a certain factor would lengthen the input sequence ($N$) by the same factor, leading to a quadratic increase ($O(N^2)$) by the square of that factor in the computational and memory demands of the transformer's attention mechanism.
>    - In contrast, BDO's architecture is robust to this change. At each time step, we first map the $N$-dimensional ROI space to a fixed d-dimensional latent space using a simple linear projection. Therefore, increasing N only affects the number of parameters in this initial linear layer and the last layer of the decoder. Also, it does not change the sequence length ($T$) or the computational complexity of the transformer encoder ($O(T^2)$).
> * This design makes BDO scalable with respect to spatial resolution, allowing us to easily conduct this 1000-ROI experiment, which would be prohibitively expensive for many other tokenization-based architectures. We will add this new analysis to the appendix in the final version.

---

> ### Author Response · Authors · 2025-11-19
>
> **5. Related Work section**
>
> * We thank the reviewer for pointing out the placement of the Related Work section. Ideally, this discussion belongs in the main text. However, it was initially moved to the appendix due to the strict 9-page limit. We are grateful for the additional page allowance granted during the rebuttal period. In the revised manuscript, we will reintegrate the core components of our literature review into the main text as a new Section 2. This section will specifically include continuous-time approaches in brain dynamics modeling and provide a detailed comparison between our SOC-based framework and recent foundation models. A comprehensive review of the broader literature will remain in Appendix A for further reference.
> ----
>
> We sincerely thank the reviewer for their constructive and detailed feedback. Your comments have guided us to clarify our comparisons with key baselines, validate our core claims on continuous-time modeling, and demonstrate the spatial scalability of our approach. We will carefully address the minor grammatical errors and typos you highlighted, along with these substantive points. If there are any remaining questions or concerns, we would be happy to discuss them further and provide additional clarification.
>
> ----
>     [1] Dong et al., Brain-JEPA: Brain Dynamics Foundation Model with Gradient Positioning and Spatiotemporal Masking
>     [2] Kan et al., Brain Network Transformer
>     [3] Azabou et al., A Unified, Scalable Framework for Neural Population Decoding
>     [4] Ye and Pandarinath, Representation learning for neural population activity with Neural Data Transformers
>     [5] Li et al., Scalable Gradients for Stochastic Differential Equations
>     [6] ElGazzar and van Gerven, Generative Modeling of Neural Dynamics via Latent Stochastic Differential Equations
>     [7] Zhou et al., Uncovering Alzheimer’s Disease Progression via SDE-based Spatio-Temporal Graph Deep Learning on Longitudinal Brain Networks
>     [8] Sacerdote and Giraudo, Stochastic Integrate and Fire Models: a review on mathematical methods and their applications
>     [9] Dadi et al., Fine-grain atlases of functional modes for fMRI analysis
>     [10] Schaefer et al., Local-Global Parcellation of the Human Cerebral Cortex from Intrinsic Functional Connectivity MRI

---

> ### Comment · Reviewer_9gcz · 2025-11-19
> **Re: Brain-JEPA results**
>
> Thank you to the authors for responding so quickly. I will read and respond to the full response at a later time, in the meantime: why do you not compare against the publicly available Brain-JEPA weights? Although comparing to a reproduction is fine, if you are matching pre-processing pipelines anyway, why not just (also) compare against the publicly available pre-trained weights? And why do you only compare against Brain-JEPA's performance on the HCP-A dataset? I believe Brain-JEPA is probably the most important work to compare against given that it is arguably the current state-of-the-art model (behind the newly published BrainHarmonix paper, which you could also compare against). Hence, limiting the comparison in this way and pushing it so far into the back of the Appendix does not instill confidence in the added value of this model over Brain-JEPA, but I am happy to be convinced otherwise.

---

> > ### Author Response · Authors · 2025-11-20
> >
> > We appreciate the reviewer’s insistence on transparency and the rigor of our comparisons. We understand why limiting the scope and not using public weights instilled a lack of confidence. We address these points below, detailing our methodology driven purely by the architectural constraint and the necessity for a fair comparison standard.
> >
> > ----
> > **1. More comprehensive study**
> > * The limitation of our initial comparison to only the HCP-A dataset was fundamentally due to the inherent rigidity of the Brain-JEPA architecture, which requires a minimum input length of 160 timesteps (by dividing the sequence into 10 patches for each ROI). This architectural constraint causes methodological issues for other datasets.
> > * For instance, the ABIDE and ADHD datasets have signal lengths of 78 and 74 timepoints, respectively, rendering the direct application of the pre-trained checkpoint infeasible. In the case of HCP-EP, although the sequence length is 410, its TR is 0.7 seconds compared to the pre-training TR of 2.205 seconds. To align the required temporal resolution, a sampling interval of 3 is necessary, which consequently reduces the resulting sequence length below the minimum 160 timesteps, thus rendering the data insufficient for the fixed-patch model.
> > * Given these architectural constraints, and the fact that the Brain-JEPA paper only included experiments on HCP-A among our four core downstream datasets (HCP-A, ABIDE, ADHD200, HCP-EP), we restricted our initial comparison scope to the HCP-A dataset. However, artificially extending the sequence length by repeating data can be considered as a methodologically questionable approach for comparative analysis. Therefore, we will proceed with the expanded comparative study, utilizing the publicly available pre-trained checkpoint as a supplementary reference.
> >
> > ----
> > **2. Publicly available pre-trained weights**
> > * Our primary objective was to ensure all SSL baselines were compared under identical training conditions. However, the publicly available Brain-JEPA pre-trained weights used 300 epochs of training, whereas our comparative SSL baseline standard was set at 200 epochs. Additionally, we noted a discrepancy in the official repository where L1 loss was implemented in the code, despite the paper stipulating L2 loss.
> > * Regarding these aspects, we considered that re-training the model under uniform conditions represented a more appropriate methodological choice. However, as mentioned above, we will conduct the expanded comparative study utilizing the publicly available pre-trained checkpoint.
> > ----
> >
> > **3. Comparison with concurrent work**
> > * We thank the reviewer for recommending the concurrent work, BrainHarmonix. We confirm that a performance comparison is feasible, although direct use of their public pre-trained weights is not possible, as their model utilizes a specific 400-ROI setting which requires pre-training the model under our unified pipeline for a consistent comparison. We will incorporate this important model into our comparative analysis soon.
> >
> > ----
> > We sincerely appreciate the reviewer for their valuable insights and guidance. We commit to executing the requested experiments immediately and will update our comments with the resulting data as soon as they become available during the discussion period.

---

> > > ### Author Response · Authors · 2025-11-25
> > >
> > > We sincerely thank the reviewer for the constructive suggestion to utilize the publicly available pre-trained weights of Brain-JEPA. We agree that comparing against the official checkpoint provides a more rigorous benchmark. Following your recommendation, we have conducted additional experiments on HCP-A, ABIDE, ADHD200, and HCP-EP using the official Brain-JEPA weights (pre-trained for 300 epochs), compared to our model (pre-trained for 200 epochs) in both LP (linear probing) and FT (fine-tuning) settings.
> > >
> > > ----
> > >
> > > * The results are summarized in Table 1 and Table 2 below. Table 1 presents the performance on HCP-A, which aligns well with Brain-JEPA's pre-training conditions (sufficient sequence length and compatible TR). Table 2 reports the results on ABIDE, ADHD200, and HCP-EP, which represent more challenging scenarios characterized by short sequence lengths and heterogeneous TRs, necessitating preprocessing adjustments like repetition for fixed-patch architectures.
> > >
> > > ----
> > >
> > > Table 1. Results for HCP-A
> > >
> > > | Methods | Age (MSE) ↓ | Age (ρ) ↑ | Gender (ACC) ↑ | Gender (F1) ↑ | Flanker (MSE) ↓ | Flanker (ρ) ↑ |
> > > |---|---|---|---|---|---|---|
> > > | Brain-JEPA-LP (86M) | 0.364 ± .023 | 0.804 ± .004 | 73.15 ± 1.09 | 73.09 ± 1.18 | 0.962 ± .131 | 0.360 ± .064 |
> > > | BDO-LP (86M) | 0.298 ± .022 | 0.839 ± .010 | 74.48 ± 1.82 | 74.52 ± 3.81 | 0.966 ± .073 | 0.343 ± .059 |
> > > | Brain-JEPA-FT (86M) | 0.275 ± .011 | 0.854 ± .008 | 82.42 ± 1.21 | 82.38 ± 1.20 | 0.891 ± .011 | 0.424 ± .017 |
> > > | BDO-FT (86M) | 0.262 ± .009 | 0.862 ± .004 | 82.92 ± 1.92 | 82.88 ± 1.99 | 0.862 ± .043 | 0.452 ± .088 |
> > >
> > > ----
> > >
> > > Table 2. Results for ABIDE, ADHD200, and HCP-EP
> > >
> > > | Methods | ABIDE (Acc) ↑ | ABIDE (F1) ↑ | ADHD200 (Acc) ↑ | ADHD200 (F1) ↑ | HCP-EP (Acc) ↑ | HCP-EP (F1) ↑ |
> > > |---|---|---|---|---|---|---|
> > > | Brain-JEPA-LP (86M) | 59.92 ± 3.74 | 58.85 ± 3.72 | 56.22 ± 1.06 | 56.34 ± 1.11 | 72.38 ± 3.56 | 71.35 ± 3.89 |
> > > | BDO-LP (86M) | 64.09 ± 2.60 | 64.04 ± 2.60 | 58.40 ± 3.03 | 58.54 ± 2.99 | 73.33 ± 3.56 | 72.54 ± 3.41 |
> > > | Brain-JEPA-FT (86M) | 64.55 ± 4.80 | 64.27 ± 4.82 | 62.15 ± 0.61 | 62.32 ± 0.51 | 75.24 ± 1.35 | 74.47 ± 1.46 |
> > > | BDO-FT (86M) | 67.90 ± 2.68 | 67.88 ± 2.67 | 63.15 ± 2.81 | 63.32 ± 2.72 | 79.05 ± 2.69  | 78.65 ± 2.63 |
> > >
> > > ----
> > >
> > > * The HCP-A dataset aligns well with the pre-training conditions of Brain-JEPA. Specifically, the effective TR after downsampling in HCP-A ($0.72\text{s} \times 3 \approx 2.16\text{s}$) is very close to the pre-training TR derived from UK Biobank ($0.735\text{s} \times 3 \approx 2.205\text{s}$), and the sequence length remains sufficient. In this favorable setting, both models demonstrate strong performance, with BDO achieving superior or comparable performance across HCP-A tasks. It is worth noting that BDO achieves these results with 200 epochs of pre-training, compared to the 300 epochs used for the official Brain-JEPA weights.
> > > * As noted in our earlier response, ABIDE and ADHD200 are multi-site datasets characterized by heterogeneous TRs in addition to short sequence lengths. For a fixed-patch architecture like Brain-JEPA, this TR variability implies that input patches correspond to inconsistent physical durations across subjects, complicating the modeling of unified temporal dynamics. This structural mismatch, combined with the need for data repetition to meet input constraints, likely limits performance. In contrast, through continuous-time modeling, BDO natively adapts to subject-specific TRs and variable lengths, maintaining higher robustness in these scenarios.
> > > * This advantage is maintained in HCP-EP (TR=0.7s), where Brain-JEPA necessitates downsampling for TR alignment ($0.7\text{s} \times 3 \approx 2.1\text{s}$) and subsequent data repetition to meet sequence length constraints. In contrast, BDO processes data directly without such adjustments. This difference likely contributes to the observed performance gap, underscoring the benefits of continuous-time modeling.
> > >
> > > ----
> > >
> > > We believe these additional comparisons clarify the distinct strengths of our proposed framework in terms of flexibility and generalization across diverse real-world fMRI datasets. We will incorporate this comprehensive comparative analysis into the final version of the manuscript to further strengthen our empirical evaluation. We are grateful for your insightful feedback, which has significantly strengthened the validation of our framework.

---

### Author Response · Authors · 2025-11-19

We sincerely thank all the reviewers for thoughtful feedback. We are pleased that reviewers recognized the importance of modeling continuous-time latent dynamics for fMRI representation learning, and acknowledged the novelty of our SOC formulation and the computational efficiency achieved by our simulation-free inference scheme. We are also encouraged by the multiple reviewers who found the paper theoretically elegant and well-written.

Please note that newly added figures supporting our rebuttal are temporarily placed in Appendix E for immediate readability, and we will place them into suitable sections in the final version of the paper.

----

In the revision, we will revise the manuscript carefully and include additional clarifications and experiments to address the reviewers' constructive comments. Since there are shared concerns across the reviews, we address these topics collectively below.

 - **Clarification of our framework**: We clarified that our core novelty lies in reframing fMRI representation learning as a SOC problem. We emphasized that our unified objective is not an ad-hoc combination but mathematically derived from the SOC formulation (Reviewer 8VFs). Additionally, we provided theoretical clarifications regarding the control policy formulation to ensure consistency between the SOC theory and our practical implementation (Reviewer SCN9).

 - **Justification of continuous latent dynamics**: We expanded our discussion on the necessity of continuous-time modeling for handling heterogeneous TRs, highlighting the temporal mismatches inherent in patch-based transformers. We further empirically validated this advantage via a new TR distortion experiment (Reviewer 9gcz). Furthermore, we demonstrated that latent dynamics show reduced latent displacement in ADHD participants, implying that the learned continuous dynamics encode ADHD group-specific signature information (Reviewer 8VFs).

 - **Sensitivity of JEPA objectives**: We addressed the discussion regarding the training sensitivity of Brain-JEPA (Reviewers 9gcz, SCN9). In the original submission, Appendix D.11 revealed that the performance of Brain-JEPA is dependent on data preprocessing steps and this observation aligns with the known sensitivity of standard JEPA objectives to input distribution shifts.

----

Notable changes will be highlighted in teal in the revised manuscript. We have made every effort to address all concerns raised, as detailed in the individual responses below. We hope these additional experiments and theoretical clarifications satisfactorily address the reviewers' concerns and further highlight the robustness and significance of our contributions.

---

### Comment · Area_Chair_iqvi · 2025-11-26
**Reminder to Engage!**

Dear Reviewers,

We are one week away from the end of the discussion period and the review responses have been posted. If you have not done so already, please read the response and check if the authors have addressed your concerns. Also please acknowledge the review by responding and stating how the response (and updated manuscript if provided) does or does not change your evaluation of the work. Earlier responses allow for meaningful engagement and potential for further clarification.

-Area Chair

---

### Author Response · Authors · 2025-12-01

We extend our sincere gratitude to reviewers for their dedication. To summarize the review and rebuttal process and outline the key updates in the revised manuscript, we first highlight the consensus on the paper's core value, followed by a concise action-item list detailing how each concern has been systematically addressed in the revised manuscript.

Overall, independent of the initial ratings, _all reviewers rated Soundness and Presentation as 'Good' or 'Excellent’_ and the initial feedback provided by the reviewers was already positive regarding the novelty and theoretical grounding of our work. Although we did not receive final responses to all reviewers, we anticipate a positive reassessment given their highly favorable evaluation of our core contributions:

----

* **Reviewer cuG3 (8)** found our paper _"exceptionally well written"_ and praised the _"innovation to use SOC in SSL context"_ as clear, that the method is _"applicable beyond targeted fMRI applications"_ and _"the experimental setup clearly shows the main benefits"_. In their final comment, the reviewer confirmed to keep their positive score.
* **Reviewer 8VFs (4)** found the SOC formulation to be _"mathematically elegant"_ and noted that our approach offers a _"principled way to model latent neural dynamics"_. They also acknowledged that our results are _"empirically strong"_ across multiple benchmarks. Following the resolution of the raised issues during the rebuttal process, the reviewer raised the score to 6.
* **Reviewer SCN9 (4)** highlighted our work as a _"highly novel and theoretically compelling contribution"_ with a _"novel and elegant formulation"_. They specifically commended the _"principled unification of MAE and JEPA"_ and the _"impressive computational efficiency"_ achieved by our simulation-free inference. Furthermore, they recognized the _"strong and comprehensive empirical validation"_.
* **Reviewer 9gcz (4)** recognized that our model _"outperforms baselines across a variety of tasks and datasets"_ and exhibits _"good scalability"_, describing the proposed SOC approach as an interesting framework for fMRI pre-training.

----

We present the revised manuscript (changes marked in **teal**), addressing key concerns:

----

**1. Theoretical Clarifications (Reviewers SCN9, 8VFs)**
* **Open-Loop Control (Appendix C.1)**: Justified using the Donsker-Varadhan variational principle, ensuring mathematical validity for simulation-free inference.
* **Drift Matrix (Appendix D.3)**: Detailed the parameterization of $D_t = V \Lambda_t V^\top$ with shared eigenbasis and learnable eigenvalues.
* **Core Novelty (Section 1)**: Clarified reframing fMRI representation learning as a SOC problem as the primary contribution.

----

**2. Additional Baselines (Reviewers 9gcz, 8VFs)**
* **Official Brain-JEPA (Appendix D.5.1)**: BDO outperforms official pre-trained Brain-JEPA, especially on heterogeneous TR datasets.
* **Solver-Based Baselines (Appendix D.5.2)**: BDO surpasses Latent ODE and GRU-ODE-Bayes in both accuracy and efficiency.
* **Concurrent Work (Appendix D.5.3)**: Comparison with BrainHarmonix (Sep 2025) suggests the advantage of continuous stochastic modeling over discrete tokenization.

----

**3. Validation of Continuous Dynamics (Reviewers 9gcz, 8VFs)**
* **TR Distortion (Appendix D.13)**: Performance degradation under artificial TR compression/dilation implies the necessity of exact time integration.
* **Qualitative Dynamics (Appendix D.11.2)**: Latent velocity norms reveal greater state-transition fluidity in healthy controls vs. ADHD patients, aligning with neuroscientific findings.

----

**4. Additional Ablation Studies (Reviewers 9gcz, SCN9)**
* **Spatial Scalability (Appendix D.7.2)**: BDO scales efficiently with higher ROI resolution (200-1000).
* **Temporal Aggregation (Appendix D.12)**: Comparison with PMA suggests potential gains from attention-based pooling over mean-pooling.

----

**5. Expanded Related Work (Reviewers 9gcz)**
* **Reintegration (Section 2)**: Moved Related Work to the main text. Highlighted how continuous modeling resolves temporal mismatches in fixed-grid SSL models on heterogeneous datasets.
* **Extended Appendix (Appendix A)**: Included detailed backgrounds on SSL and SOC.

----

We once again appreciate all the reviewers for their rigorous evaluation and helpful insights, which have significantly contributed to strengthening the quality and clarity of our work.


Sincerely,

The Authors of Submission 15175

---

### Meta-Review · Area_Chair_92ac · 2026-01-13

**Summary:**

This paper proposes BDO (Brain Dynamics with Optimal Control), a self-supervised learning framework for fMRI representation learning that formulates the problem as a stochastic optimal control (SOC) problem over continuous-time latent dynamics. The core innovation is modeling brain activity as continuous-time SDEs to handle temporal irregularity across heterogeneous datasets with varying temporal sampling rates, while unifying MAE and JEPA objectives through the SOC formulation. A simulation-free inference strategy enables scaling to large datasets.

The paper initially received largely negative scores and just one positive score. A strong rebuttal with new experiments demonstrating a favorable comparison to more baselines, validation of the continuous time framework, clarification of mathematical details, and a demonstration of computational scaling addressed many concerns and caused one negative reviewer to raise their score to acceptance and would likely have led the others to also raise scores to borderline acceptance.

**Reviewer Concerns:**

The paper initially received largely negative scores and just one positive score. A strong rebuttal with new experiments demonstrating a favorable comparison to more baselines, validation of the continuous time framework, clarification of mathematical details, and a demonstration of computational scaling addressed many concerns and caused one negative reviewer to raise their score to acceptance and would likely have led the others to also raise scores to borderline acceptance.

**Reviewer Scores:**

While two reviewers confirmed their positive scores indicating acceptance (8, 4->6), the other two reviewers would likely have raised their scores to borderline acceptance following the rebuttal response.

---

### Decision · Program_Chairs · 2026-01-26

Accept (Poster)